# The link between European warm temperature extremes and atmospheric persistence

Emma Holmberg[1], Gabriele Messori[1, 2], Rodrigo Caballero[2], and Davide Faranda[3,4,5]

[1]Department of Earth Sciences and Centre of Natural Hazards and Disaster Science (CNDS), Uppsala University, Uppsala, Sweden
[2]Department of Meteorology and Bolin Centre for Climate Research, Stockholm University, Stockholm, Sweden
[3]Laboratoire des Sciences du Climat et de l'Environnement LSCE-IPSL, CEA Saclay l'Orme des Merisiers, UMR 8212 CEA-CNRS-UVSQ, Université Paris-Saclay, 91191 Gif-sur-Yvette, France
[4]London Mathematical Laboratory, London, UK
[5]Laboratoire de Météorologie Dynamique/IPSL, École Normale Supérieure, PSL Research University, Sorbonne Université, École Polytechnique, IP Paris, CNRS, 75005, Paris, France

**Correspondence:** Emma Holmberg (emma.allwright@geo.uu.se)

**Abstract.** We investigate the link between warm temperature extremes in Europe and the persistence of large-scale atmospheric circulation patterns for both winter and summer, along with some possible physical mechanisms connecting the two. We assess atmospheric persistence leveraging concepts from dynamical systems theory, and reconcile this approach with the more conventional meteorological views of persistence. We find that wintertime warm spells are partly associated with persistent zonal advection at the surface level, but display no statistically significant persistence anomaly in the mid troposphere. For summertime heatwaves, we find a weak yet significant link to anomalously persistent circulation patterns in the mid troposphere, while there are few significant persistence anomalies of the surface circulation pattern. We further find no evidence of a strong warm temperature advection signal. This suggests that other radiative and dynamical processes, for example sensible heating and adiabatic warming, as well as local effects, could play a more important role than large-scale warm temperature advection for these events. We thus argue that persistent atmospheric configurations are not a necessary requirement for warm temperature extremes, and that the results depend to a considerable extent on region and tropospheric level.

## 1 Introduction

Heatwaves routinely exact a heavy toll on society, including increased mortality, aggravation of droughts, crop failure, damages to infrastructure and power outages (Muthers et al., 2017; Ouzeau et al., 2016; Adélaïde et al., 2021). A notable example was the 2003 European heatwave, which led to an estimated approximately 40,000 excess deaths, and economic damages of roughly 13 billion EUR (Garcia-Herrera et al., 2010; De Bono et al., 2004). Moreover, in most areas of the world these events are expected to become increasingly severe under anthropogenically-forced climate change (Seneviratne et al., 2012; Perkins-Kirkpatrick and Gibson, 2017; Seneviratne et al., 2014). For example, extremely hot summers, where the temperature exceeds a fixed threshold, are expected to become increasingly likely throughout this century (Christidis et al., 2015). In Europe, the Mediterranean region is projected to experience an increase in the frequency and duration of heatwaves, whilst south-central

Europe is expected to experience a pronounced increase in the amplitude of heatwaves (Fischer and Schär, 2010). Furthermore, heatwaves are expected to be associated with a dramatic rise in damages to infrastructure over the next century (Forzieri et al., 2018). Uncommonly high temperatures can also have widespread impacts during the winter season. Wintertime warm spells can lead to significant disruptions in ecological systems, for example by causing plants to exit dormancy too early in the season, potentially leaving them vulnerable to subsequent cold spells (Ladwig et al., 2019). Some livestock are also highly sensitive to changes in the freeze-thaw cycle (Furberg et al., 2011). This underscores the importance of furthering our understanding of summertime heatwaves and wintertime warm spells.

Wintertime European warm spells are often associated with a large-scale zonal flow (Slonosky and Yiou, 2002), which results in warm and moist air from the North Atlantic being transported to Europe. Summertime heatwaves are instead typically associated with atmospheric blocking over the Euro-Atlantic sector (Rousi et al., 2022; Lupo, 2021; Schaller et al., 2018). A summertime blocked flow over the North Atlantic leads to weakened zonal flow and more pronounced meridional flow, favouring the southerly advection of very hot airmasses to Europe, for example during the 2003 and 2010 heatwaves as discussed in Miralles et al. (2014). Cassou et al. (2005) notes that another atmospheric circulation regime, namely the Atlantic low with an associated ridge over southern continental Europe, can also be correlated with heatwaves. Following the same principle as for blocking, this configuration leads to weakened zonal and strengthened meridional flow over southern Europe. In addition to the advection of warm air, other physical mechanisms can lead to heatwaves, including radiative forcing and subsidence (Zschenderlein et al., 2019; Bieli et al., 2015). Indeed, the latter study finds the effect of temperature advection to be almost negligible for summertime European heatwaves, in contrast with earlier literature. Soil moisture anomalies have also been linked to European heatwaves (e.g. Fischer et al., 2007; Hirschi et al., 2011; Zschenderlein et al., 2019), however, soil moisture has a characteristic time scale which is much longer than that of the atmosphere. Including effects from both the atmosphere and soil moisture in a persistence perspective would thus require two largely distinct analyses and here we choose to focus on the former.

Both 500 hPa geopotential height and sea-level pressure (SLP) fields have been used to analyse circulation patterns associated with temperature extremes. Geopotential height fields were, for example, used by Messori et al. (2017) to analyse wintertime temperature extremes, and by Quesada et al. (2012) to analyse summertime heatwaves, the associated weather regimes and wet or dry winter/spring conditions. Similarly, Jézéquel et al. (2018) argued that geopotential height fields are better suited than SLP fields to investigate temperature anomalies, as the latter can be subject to a blurring of the signal due to heat lows. Nonetheless, a number of studies successfully used SLP fields to investigate heatwaves. For example Alvarez-Castro et al. (2018) and Cassou and Cattiaux (2016) used SLP to analyse the atmospheric dynamics of heatwaves, and the seasonal relationship between temperature and circulation patterns in a warming world, respectively.

Regardless of the variable used for analysis, based on the above literature one may make the hypothesis that the occurrence of persistent blocked (zonal) flows in summer (winter) may favour the occurrence of warm temperature extremes, since short-

lived circulation anomalies are unlikely to lead to the build-up of large and long-lived temperature anomalies. Blocking has been diagnosed both as a localised atmospheric feature (Davini et al., 2012; Tibaldi and Molteni, 1990; Sousa et al., 2021), and as a weather regime representing one of the recurrent states of the North Atlantic atmospheric circulation (e.g. Hannachi et al., 2017; Hochman et al., 2021). In both cases, blocks are often regarded as very persistent atmospheric flow patterns, which can last for several days up to a few weeks (Drouard et al., 2021). Summertime blocks are typically associated with adiabatic warming of air due to subsidence and clear sky conditions allowing for radiative warming near the surface. Intuition would suggest that the probability of a heatwave occurring would be increased with an increased duration of these atmospheric conditions which allow air to be warmed, and many studies imply as much. However, we are not aware of any studies explicitly relating the duration of a blocked flow pattern to the probability of heatwave occurrence.

The persistence of a block as diagnosed using feature-based detection algorithms is often imposed by the algorithm itself (e.g. by requiring a minimum persistence to define a feature as a block (Schwierz et al., 2004; Davini et al., 2012)). Moreover, it focuses on the feature itself as opposed to the larger-scale atmospheric flow pattern, and often allows for the center of the block to move a significant distance over the block's lifetime and for the block's intensity to vary, provided it stays above a given threshold. The persistence of blocking as diagnosed by weather regimes provides an indication of the larger-scale atmospheric persistence, as the whole Euro-Atlantic region is typically considered. However, weather regimes provide a very coarse-grained partitioning of atmospheric variability, most often into four reference states (Michelangeli et al., 1995). Different timesteps assigned to the 'blocked' regime can therefore display relatively large differences in the atmospheric flow pattern.

More recently, the persistence of blocking has been considered from the angles of Rossby wave activity or non-equilibrium quasi-stationary states in a dynamical system. The former approach considers blocking as a manifestation of planetary wave breaking (Pelly and Hoskins, 2003; Masato et al., 2012; Barnes and Hartmann, 2012). The latter approach applies mathematical tools from dynamical systems theory to atmospheric fields, and interprets North Atlantic atmospheric variability as having a number of non-equilibrium quasi-stationary states with specific dynamical characteristics. The conventional view considers two such states, namely blocked and zonal flows (Legras and Ghil, 1985), with this approach being further developed by Mo and Ghil (1987). The dynamical properties and transitions between these two states can then be used to infer their persistence.

A further development stemming from dynamical systems theory leverages the concept of atmospheric recurrences, or analogues – namely the fact that the atmosphere displays configurations similar to ones it has displayed in the past. Unlike weather regimes where a very large number of flow states are assigned to a given regime, analogues are typically defined quite stringently, such that each state only has a limited number of analogues. Leveraging the notion of the extremal index, namely a measure of clustering in extreme value theory (Section 3.2.2 Lucarini et al., 2016; Moloney et al., 2019), it is possible to use analogues to estimate the persistence of a given atmospheric configuration. This definition of persistence can be intuitively understood as the length of time a map is expected to remain similar to itself. In the present case where we use daily data, the persistence corresponds in other words to the expected number of consecutive days for which the atmospheric circulation in a

chosen geographical domain displays a 'similar' structure of high and low pressure systems. A key aspect of this approach is that it does not categorise circulation patterns like a regime-based approach. The persistence of the atmosphere on each day in the data is calculated for that specific day, based on its (in practice) unique set of analogues. When applied to the North Atlantic in previous work, this approach has highlighted zonal flows as being highly persistent – more so than blocked flows. This was shown by Faranda et al. (2017) and Faranda et al. (2016) for the full year, and Messori et al. (2017), for the winter season. A similar conclusion was drawn by Hochman et al. (2021), who, using the extremal index, identified the zonal weather regime as being more persistent than some of the blocked weather regimes in a 7-regime classification (albeit with Greenland blocking being the most persistent amongst all regimes). There is therefore an apparent discrepancy between blocking as a persistent atmospheric feature (Legras and Ghil, 1985; Drouard et al., 2021) and blocking as a transient feature of an atmosphere whose persistent state is one of zonal flow (Faranda et al., 2016, 2017; Messori et al., 2017).

In this study, we adopt this dynamical systems viewpoint to explore whether summertime heatwaves and wintertime warm spells in Europe are related to circulation patterns which display above-average persistence. In doing so, we will attempt to reconcile the different views highlighted above on the persistence of blocked and zonal flows, and explore the strengths and limitations of the dynamical systems view of atmospheric persistence. The remainder of the paper is structured as follows: Section 2 details the datasets and methods used for the analysis, Section 3 shows composites of the atmospheric configurations associated with the warm temperature extremes, and an analysis of the persistence of these configurations and of possible driving mechanisms for heatwaves, namely temperature advection and radiative forcing. Section 4 discusses these results and finally Section 5 presents the conclusion of this paper.

## 2 Data and methods

### 2.1 Data

We use the ERA5 reanalysis dataset from the European Centre for Medium Range Weather Forecasts (ECMWF) (Hersbach et al., 2020) over 1979–2018. The specific variables used were temperature at 2m (t2m), meridional and zonal wind components at 10m ($v_{surf}$ and $u_{surf}$ respectively), geopotential at 500 hPa (Z500), temperature at 500 hPa ($T_{500}$), meridional and zonal wind components at 500 hPa ($v_{500}$ and $u_{500}$ respectively), surface sensible heat flux (SSHF), surface pressure ($p_{surf}$) and SLP, all at a horizontal spatial resolution of 0.5°. Daily means were calculated from hourly data for surface variables, and from 6-hourly data for variables considered at 500 hPa, due to the much larger size of pressure-level variables. Our Euro-Atlantic domain spans 25–75 °N, 345–50 °E, similar to that of Messori et al. (2017) but with a reduced westerly extension so as to ensure the signal is directly relevant to Europe. The anomalies for each variable were computed relative to a calendar day climatology smoothed with a 31 day running mean.

### 2.2 Definitions

We define heatwaves at single gridboxes following Alvarez-Castro et al. (2018), that is 5 or more consecutive days with temperature anomalies above the 90th percentile of the local distribution, namely the distribution at a given grid box. We also define regional heatwaves over the British Isles, Scandinavia, central Europe (Germany), western Russia (Russia), Iberia and the central Mediterranean (Mediterranean) (following Zschenderlein et al., 2019), by requiring that at least five percent of the land grid boxes in the region must experience a heatwave, as defined above, for a given day. The regions are shown by black boxes in Fig. 1. The summer season is defined as June–September (JJAS) and the winter season is defined as December–March (DJFM). For the remainder of this paper we will use the term warm spell to refer to a set of days which satisfy the above definition for a heatwave and which occur during the winter season. The surface potential temperature advection, $T^{potential}_{surf,adv}$ was defined as follows:

$$T^{potential}_{surf,adv} = -v_{surf} \times \nabla T^{potential},$$ (1)

with units $[K(day)^{-1}]$ and where $v_{surf} = (u_{surf}, v_{surf})$ is the vector surface wind, and the potential temperature, $T^{potential}$ is defined as:

$$T^{potential} = t2m \times (p_{surf}/p_0)^{-R/C_p},$$ (2)

where $p_{surf}$ is surface pressure, $p_0 = 1000$ hPa and $R/C_p \approx 2/7$. Temperature advection at 500 hPa, $T_{500,adv}$, was defined as follows:

$$T_{500,adv} = v_{500} \times \nabla T_{500},$$ (3)

with units $[K(day)^{-1}]$ and where $v_{500}$ denotes the vector form of $u_{500}$ and $v_{500}$. The function gradient from the python package NumPy (Harris et al., 2020) was used to compute the gradients in Equations (1) and (3).

Statistical significance at a level of $p = 0.05$ was tested for each grid box in the composite plots in Sections 3.1 and 3.2 using a two-sided test to assess the uncertainty of the location of the true mean using a cluster bootstrap method (Cameron et al., 2008). This method was used because we want to study the entire duration of the heatwave, yet the days within a given heatwave are highly correlated. The same cluster bootstrap method was used to assess significance for the box plots in Section 3, however this time a one-sided test was used to test for negative anomalies. Sensitivity tests for the definition of heatwaves and warm

spells can be found in the Appendix A. Furthermore, the method detailed in Wilks (2016) has been applied in all figures where significance testing was performed to control for the False Discovery Rate due to the multiple simultaneous significance tests.

## 2.3 Atmospheric persistence computation

We calculate atmospheric persistence separately for the SLP and Z500 fields over the full Euro-Atlantic domain considered

here. This enables us to capture both surface and mid-level flow patterns. The computation of persistence relies on the dataset being analysed including good analogues for all its timesteps. We show in Fig. A4 in the Appendix A analogue quality tests, which demonstrate that the analogues we find in our data are of sufficient quality to perform this analysis, and are not severely impacted by the non-stationarity of the atmosphere due to the atmospheric warming trend. We define persistence as $\theta^{-1}$, where $\theta$ is the extremal index (Moloney et al., 2019) and persistence is in units of timesteps. Persistence is an estimate of the average

number of consecutive recurrences, or analogues, for a given atmospheric state. In other words, it quantifies how long a given state of the atmospheric circulation is expected to remain similar to itself. The computation of $\theta$ depends on the definition of recurrences, which we identify here as follows:

1) The Euclidean distance between the SLP or Z500 maps for each day and all others in the data set is computed. Thus, for any given day we can quantify how 'similar' it is to any other day.

2) Of these distances, the smallest 5 percent ($q = 0.95$), as recommended by Süveges (2007) are selected as analogues, that is the days that are most similar to the given day.

3) A generalised Pareto distribution is fitted to the negative log of the analogue distances to ensure that the distribution of negative log distances converges (Ferro and Segers, 2003). By applying this requirement we ensure that we have a sufficiently large data set and analogues of sufficient quality.

After this requirement is satisfied, $\theta$ can be estimated using the methodology of Süveges (2007). Specifically, we apply Equation 2 from Süveges (2007), where they have derived the maximum likelihood estimator for $\theta$. Indeed, the naïve estimator involving simply counting the average number of consecutive analogues for a given day provides biased results. The inverse of $\theta$ is the expected length of sequences, or clusters, of similar maps. In our case, this corresponds to the expected number of temporally consecutive atmospheric analogues in days. Since we consider a fixed number of analogues for each day, the quantity is also

related to the average time between clusters of analogues. This connects to our intuitive perception of persistence, as a low $\theta$ value (high persistence) would correspond to a map which typically remains similar to itself for more days than average, and has a longer time between clusters of similar days. In practice, the calculation of $\theta$ is performed using the package CDSK

by Robin (2021). Sensitivity tests for both the geographical domain on which $\theta$ is computed and the threshold for analogue selection (q) can be found in the Appendix A.

## 3  Results

### 3.1  Characterising Atmospheric Persistence during Warm Temperature Extremes

We first examine the atmospheric configurations associated with regional European warm temperature extremes in the extended winter and summer seasons (Fig. 1).

In winter (panels a – f), the warm spell regions are typically located on the northern edges of areas of positive Z500 anomalies, with anticyclonic structures located so as to favour advection from the warm/wet southern flank of a zonal flow towards the regions of interest. During summer, in the more north western regions, namely Scandinavia and the British Isles (panels g, j), the Z500 anomalies are reminiscent of canonical blocked configurations, with an anticyclonic structure located so as to block zonal flow towards the areas of interest. The other four regions (panels h, i, k, and l) show a positive Z500 anomaly core in the vicinity of the respective region, albeit with a smaller magnitude than in the two north western regions. These configurations are consistent with those discussed in Zschenderlein et al. (2019) for the summer season. We show composites of SLP in the Appendix A (Fig. A10), which show structures well in agreement with Fig. 1, except for the Mediterranean region (panel j) during summer, which shows a very weak signal relative to the other regions (panels g–i).

Next, we examine the persistence of the surface and mid-level atmospheric configurations associated with heatwaves by considering composites of $\theta$ anomalies for heatwaves on a grid box by grid box basis, followed by the anomalies of $\theta$ during heatwaves on a regional scale (Figs. 2 and 3 respectively).

Significant negative $\theta$ anomalies (recall the inverse relationship between $\theta$ and persistence, meaning that smaller $\theta$ values correspond to larger persistence values) are present across much of Europe during winter (Fig. 2a) at the surface level. At the mid level, negative anomalies are mostly limited to northern Scandinavia, whilst a region of weak positive $\theta$ anomalies is present over southern Scandinavia and Russia (Fig. 2b). In summer, negative surface $\theta$ anomalies are present over Scandinavia and the northern part of the British Isles, albeit with limited significance (Fig. 2c,d). There is a clearer signal for Z500, with significant negative anomalies over parts of western and northern Europe. In winter, SLP shows in general $\theta$ anomalies of a larger magnitude than Z500, whilst in summer we see the opposite. Focusing on regional heatwaves, Fig. 3a shows that wintertime warm spells appear to be associated with significantly enhanced surface level atmospheric persistence in Scandinavia and Germany, and non-significant negative $\theta$ anomalies in all other regions. No anomalous persistence signal is seen at the mid-level (Fig. 3b). In summer, heatwaves in the British Isles and Scandinavia display weak yet significant negative $\theta$ anomalies at the surface level whilst all other regions show near-zero or weakly positive $\theta$ anomalies (Fig. 3c). At the mid-level in Summer, comparatively small yet significant negative $\theta$ anomalies can be seen for Germany, Scandinavia, the British Isles, Iberia and the Mediterranean (Fig. 3d), suggesting a heightened persistence of the associated atmospheric flow configurations.

### 3.2  Characterising summertime heatwaves and wintertime warm spells in Europe

The above results for wintertime warm spells (Section 3.1) are partly consistent with previous literature which has found zonal flow configurations in SLP, in turn connected to warm wintertime temperatures, to be highly persistent. The results for summertime heatwaves highlight a link with mid tropospheric persistence, consistent with the widely discussed link with atmospheric

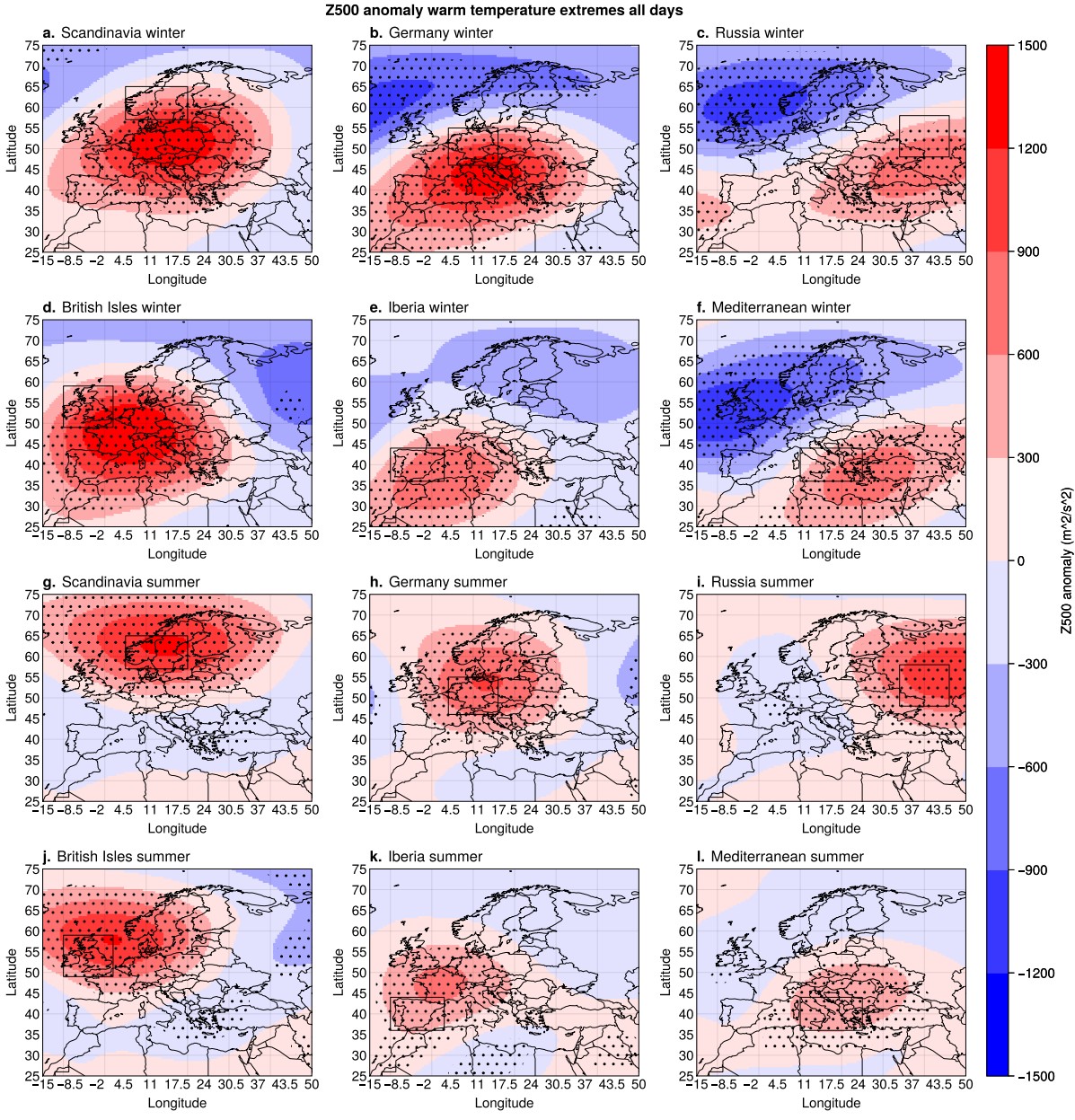

**Figure 1.** Composite Z500 $[m^2 s^{-2}]$ anomalies during warm spell (top)/ heatwave (bottom) days in (a,g) Scandinavia, (b, h) Germany, (c, i) Russia, (d, j) British Isles, (e, k) Iberia, (f, l) Mediterranean, during winter (a–f) and summer (g–l). Statistical significance is assessed as described in Section 2 and shown with grey stippling.

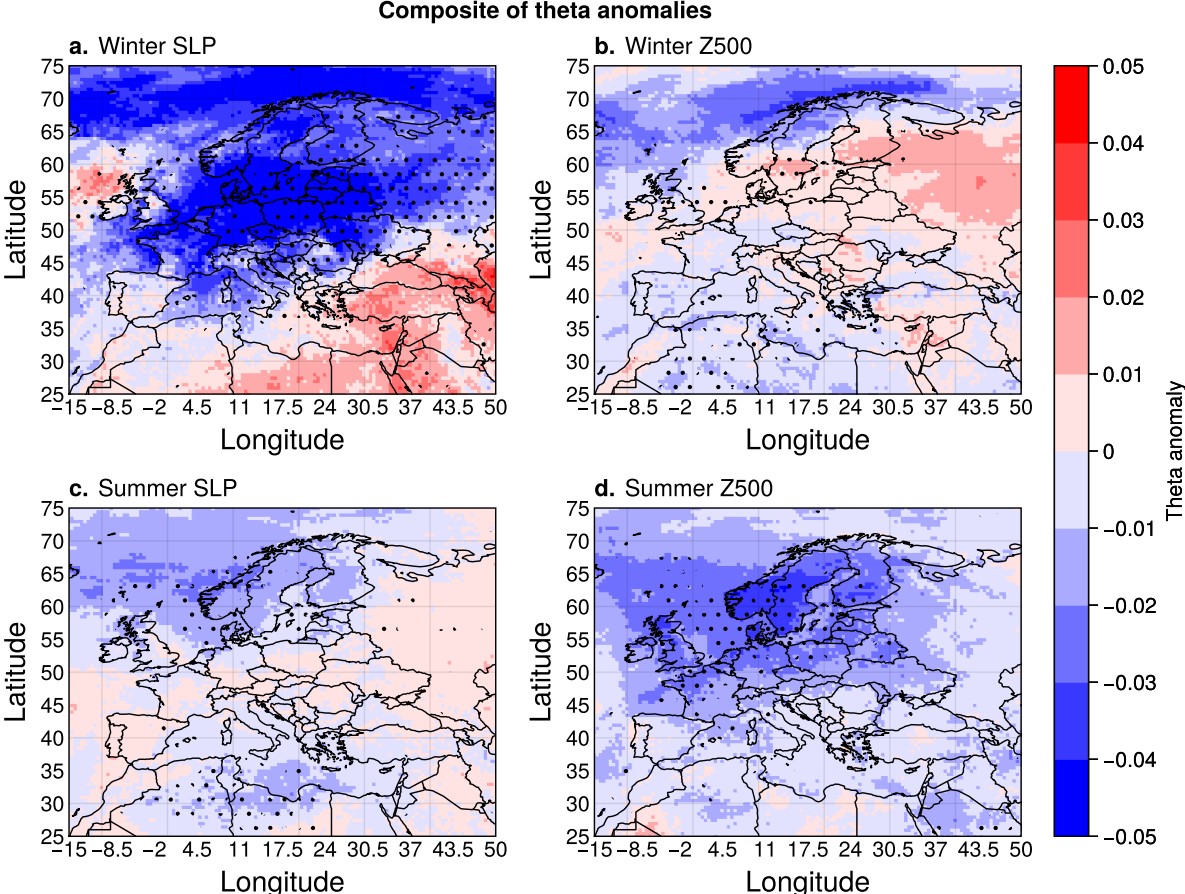

**Figure 2.** $\theta$ $[days^{-1}]$ anomalies for all warm spell or heatwave days for each grid box, during winter (a,b) and summer (c,d), where $\theta$ has been calculated using SLP fields in (a,c) and Z500 fields in (b,d). Statistical significance is assessed as described in Section 2 and shown with stippling.

blocks. We do however note that the mid tropospheric persistence anomalies are relatively weak. To better understand these features, we investigate further the physical characteristics of warm temperature extremes.

Panels a–f in Fig. 4 show composites of surface potential temperature advection during wintertime warm spells, generally pointing to a dominant role of maritime warm air advection towards the region of interest. This is consistent with the zonal flows highlighted in Fig. 1, which correspond to advection of comparatively warm oceanic air towards the continent. Panels g–l in Fig. 4 show a weaker signal for surface potential temperature advection during summertime heatwaves in the regions of interest, suggesting that in many cases the advective component can only explain part of the observed large temperature

anomalies.

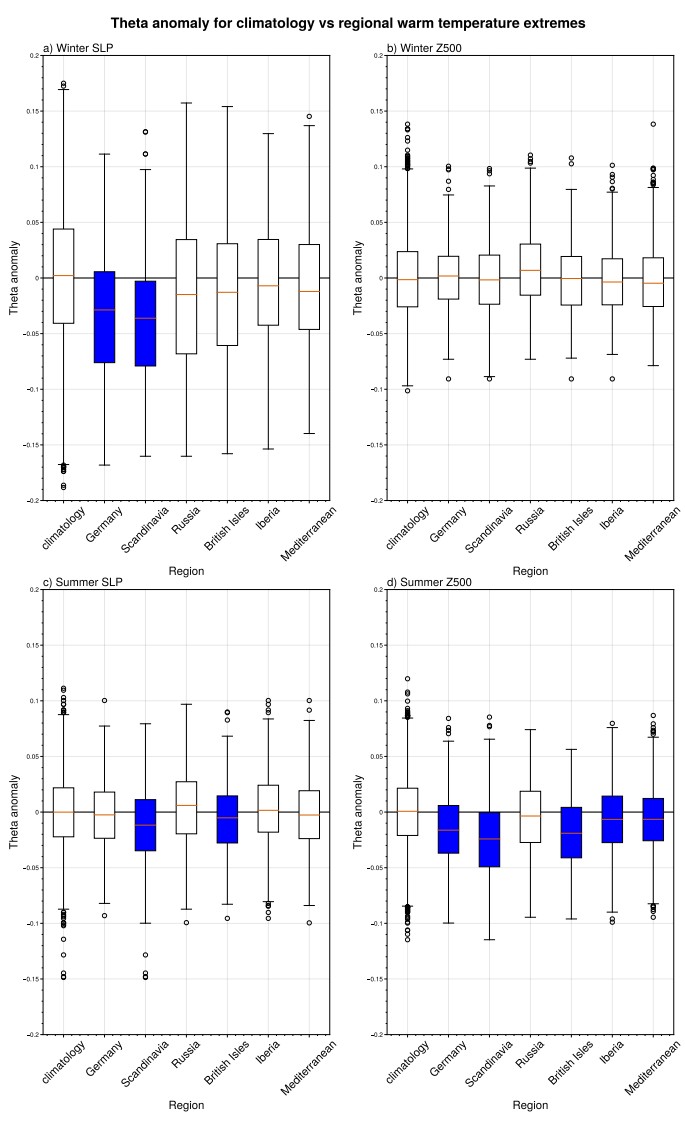

**Figure 3.** Box plots of $\theta$ $[days^{-1}]$ anomalies for all heatwave (c,d) or warm spell (a,b) days in the considered regions, at both surface (a,c) and mid (b,d) levels. Blue boxes indicate statistical significance, assessed as described in Section 2.

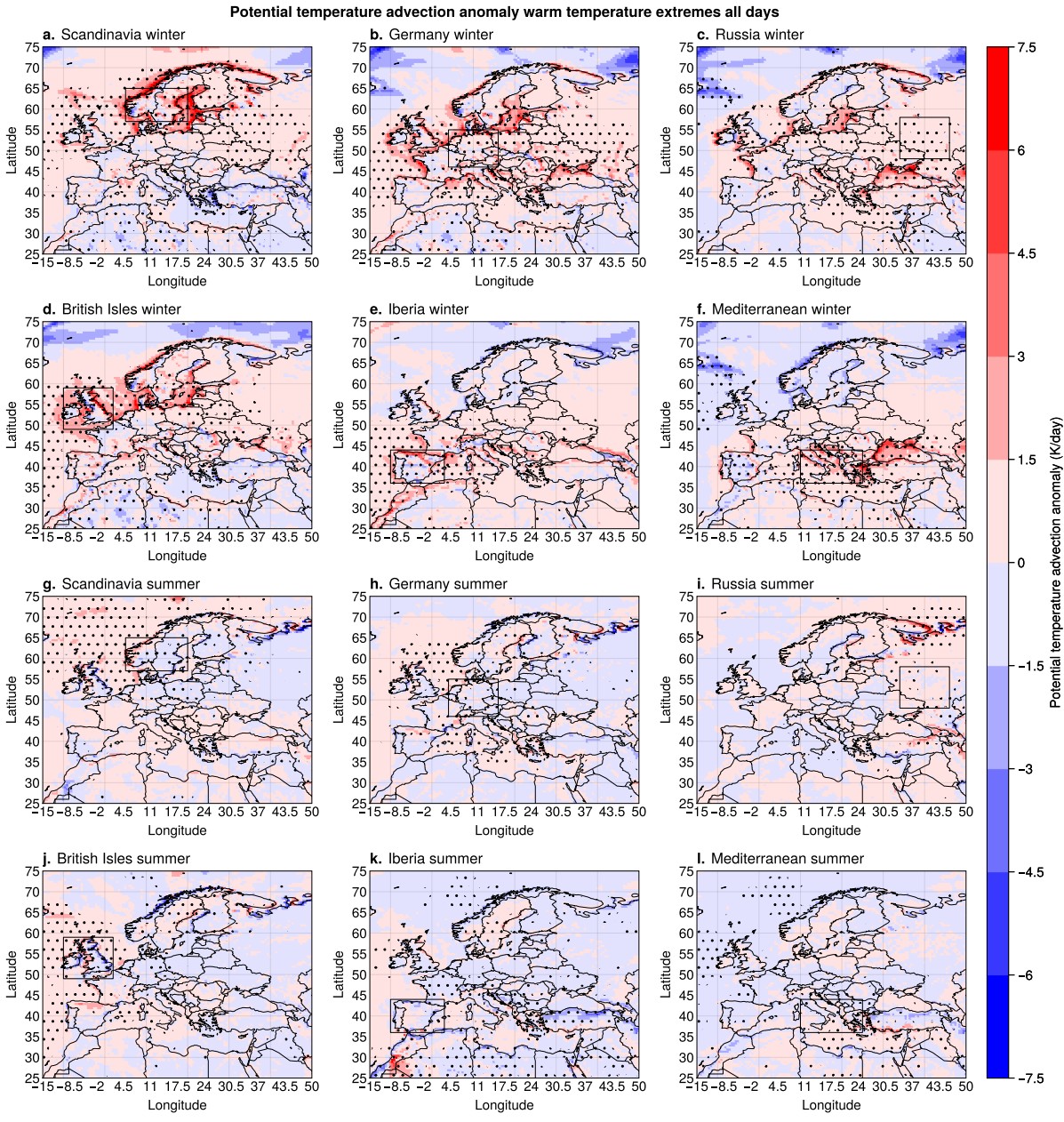

**Figure 4.** *Potential temperature advection* [$K/day$] composite anomalies at the surface level, during warm spell and heatwave days in (a, g) Scandinavia, (b, h) Germany, (c, i) Russia, (d, j) British Isles, (e, k) Iberia, (f, l) Mediterranean, during winter (a–f) and summer (g–l). Positive anomalies, denoted by the colour red, represent warm advection, whilst the colour blue corresponds to cold air advection. Statistical significance is denoted with stippling and assessed as described in Section 2.

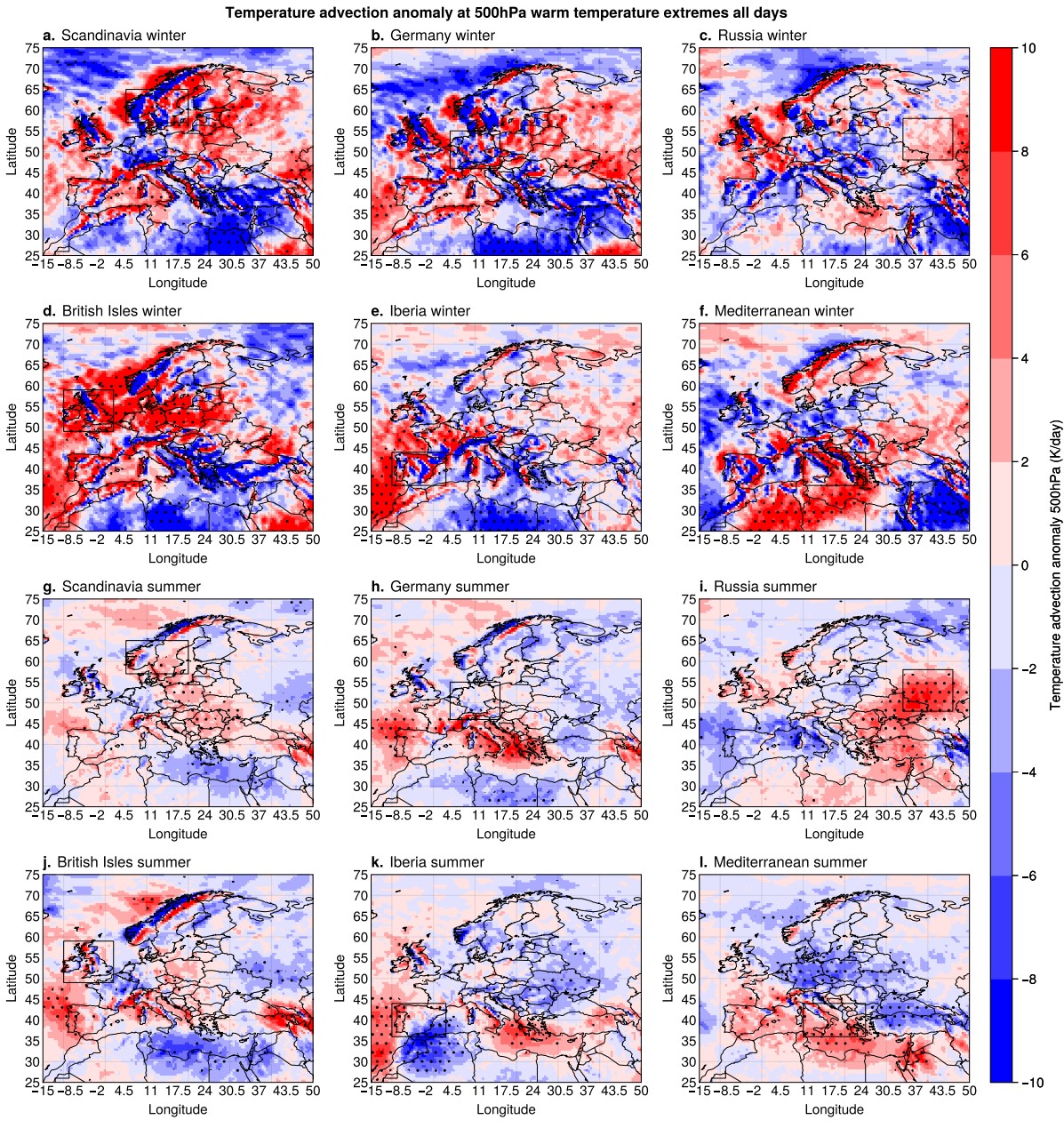

**Figure 5.** *Temperature advection* [$K/day$] composite anomalies at 500 hPa, during warm spell and heatwave days in (a, g) Scandinavia, (b, h) Germany, (c, i) Russia, (d, j) British Isles, (e, k) Iberia, (f, l) Mediterranean, during winter (a–f) and summer (g–l). Positive anomalies, denoted by the colour red, represent warm advection, whilst the colour blue corresponds to cold air advection. Statistical significance is denoted with stippling and assessed as described in Section 2.

Panels a–f in Fig. 5 show composites of mid level temperature advection during wintertime warm spells, again pointing to a significant role of warm air advection for many of the regions. Panels g–l in Fig. 5 show a weaker but significant signal for mid-level temperature advection during summertime heatwaves.

Fig. 6 shows composite SSHF anomalies during heatwaves. Positive anomalies imply an upward flux – in other words the surface heats the air above it whilst negative anomalies imply the air above the surface is warmer than the surface itself. During wintertime warm spells (Fig. 6a–f), the widespread negative anomalies indicate that a relatively warm atmosphere helps warm the surface, consistent with a large role for temperature advection. Iberia, the Mediterranean and Russia show less clear pictures, with weak anomalies over land and noisier significance stippling in the Iberia and the Mediterranean regions. During summertime heatwaves (Fig. 6g–l), there are widespread positive anomalies in Scandinavia, Germany and the British Isles, while weaker yet positive signals are found over the Mediterranean and the western part of Iberia, and no clear positive signal is found over Russia. To summarise, we observe negative winter and positive summer SSHF signals (Fig. 6), in combination with a moderate advective signal at the surface in winter and comparatively weak signal in summer (Fig. 4). We interpret this as evidence that warm temperature advection is critical for wintertime warm spells, whilst radiative forcing and other dynamical or local factors play an important role in European summertime heatwaves.

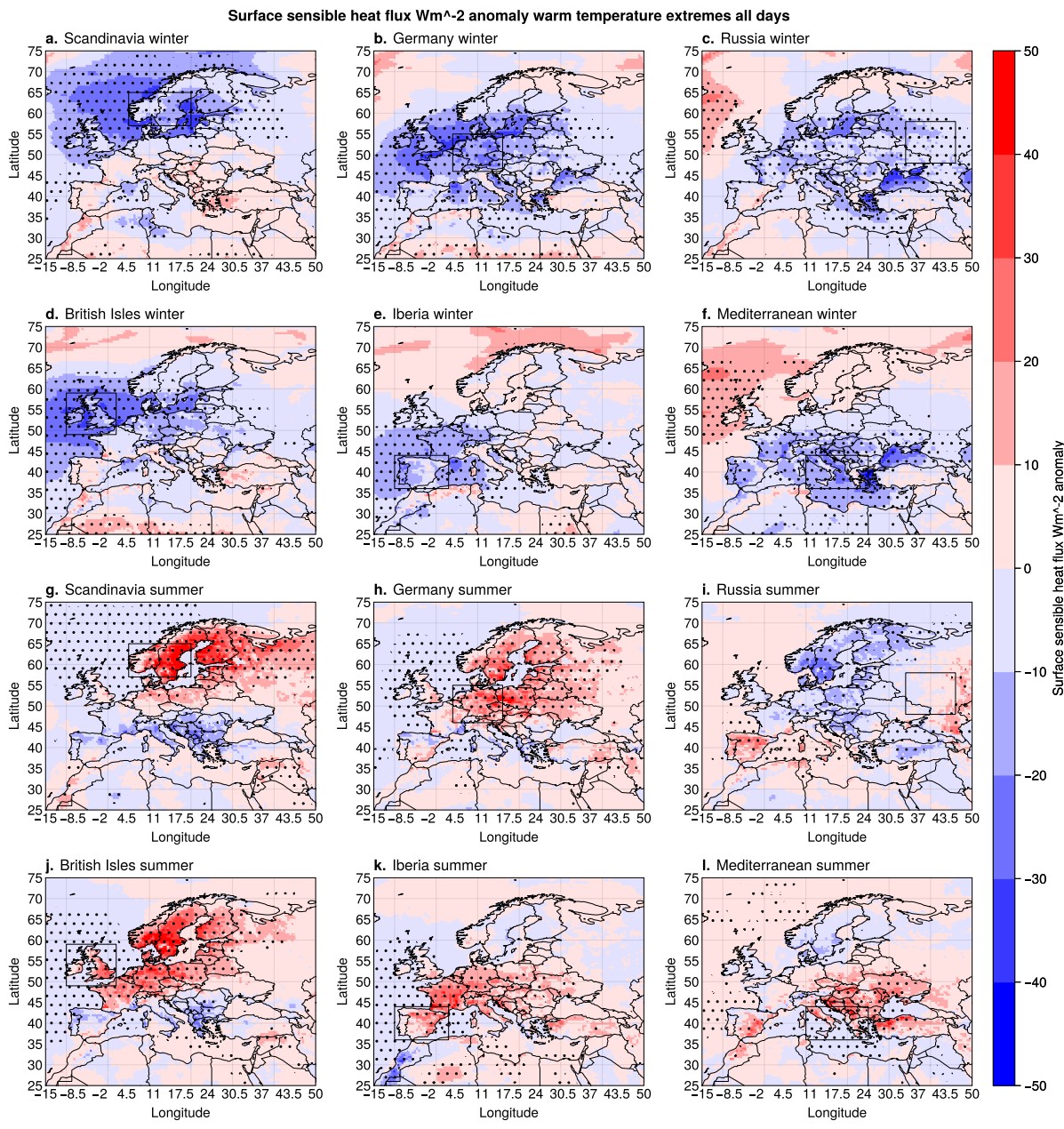

**Figure 6.** Surface sensible heat flux $[Wm^{-2}]$ composite anomalies during during warm spell and heatwave days in (a, g) Scandinavia, (b, h) Germany, (c, i) Russia, (d, j) British Isles, (e, k) Iberia, (f, l) Mediterranean, during winter (a–f) and summer (g–l). Statistical significance is denoted with stippling and assessed as described in Section 2.

## 4   Discussion

Both SLP and Z500 have been widely used to analyse circulation patterns associated with temperature extremes. Using a quantification of atmospheric persistence based on dynamical systems theory (Faranda et al., 2017) we investigate the regional persistence of these fields during European warm temperature extremes.

We consider both summertime heatwaves and wintertime warm spells. Conventionally, summertime European heatwaves have been associated with persistent blocking (Legras and Ghil, 1985; Mo and Ghil, 1987), typically lasting several days to a few weeks (e.g. Drouard et al., 2021). Our results show a link between summertime heatwaves and persistent atmospheric configurations primarily in the mid-troposphere. Only heatwaves in Scandinavia and the British Isles also show a clear link to anomalously high persistence in SLP. This is consistent with the findings of Faranda et al. (2017), who presented evidence that blocking is not associated with high persistence when considering the full year and surface-level circulation patterns. Whilst five of the six considered regions appear to show at least some link between heatwaves and mid level persistent atmospheric configurations, we note that the median negative $\theta$ anomalies for regional heatwaves attain at most magnitudes around 0.02. An anomaly in $\theta$ of -0.02 would correspond to an anomaly in persistence of less than five hours, relative to a mean summertime value of $\theta$ of approximately 0.33, when calculating $\theta$ using the Z500 field. Thus, we argue that heatwaves are to some extent favoured by persistent atmospheric conditions, but this is far from a necessary condition. Our work also connects to that of Lucarini and Gritsun (2020), who show that the onset and decay of blocking corresponds to structurally unstable states. In the context of dynamical systems, structural instability means that the entire system is highly sensitive to perturbations, and a small perturbation can lead to instantaneous qualitative changes between states. In the current analysis, the system would be the atmosphere and the mathematical equations which govern it, and a state would correspond to an instantaneous map of some representative variable. This means that the persistence of a state, which is analogous to the expected time between qualitative changes in atmospheric states, is also highly sensitive to perturbations, and thus a clear link between blocking, heatwaves and anomalously persistent atmospheric configurations from a dynamical systems perspective is not necessarily expected.

Wintertime warm spells are often associated with large-scale zonal flows (Slonosky and Yiou, 2002), which have been previously highlighted as persistent atmospheric configurations (e.g. Faranda et al., 2016; Messori et al., 2017). We find partial support for the link between wintertime warm spells and atmospheric persistence. When looking at Z500, atmospheric persistence during warm spells is close to climatology in all the regions we consider here, with a weak, non-significant signal of increased persistence in Iberia and the Mediterranean. These are indeed two of the regions that Messori et al. (2017) found to show the closest link between persistent zonal flows and high wintertime temperatures. A clearer signal of increased persistence is visible at the surface, albeit being significant only for Germany and Scandinavia. This is consistent with the notion of zonal positive North Atlantic Oscillation (NAO)–type flows being more persistent than average, as previously noted by Faranda et al. (2017), and of the NAO+ influencing warm wintertime temperatures in Northern Europe (Vihma et al., 2020). An investigation of atmospheric persistence during local temperature extremes for both warm spells and heatwaves provides consistent results,

with SLP for wintertime warm spells and Z500 for summertime heatwaves being the two cases showing the most widespread signal of increased persistence.

We thus find qualitatively different results depending on whether SLP or Z500 is used as a proxy for atmospheric flow patterns. We further find a seemingly only partial agreement with conventional notions of persistence in the meteorological literature, and in particular with the connection between highly persistent blocking, on timescales of several days to weeks, and European heatwaves. Concerning the first point, we argue that this is because the two fields we analyse provide different information, which has seldom been analysed jointly in the context of atmospheric persistence in the literature. Mid-tropospheric flows reflect closely the dynamics of blocking, while the surface field may reflect features such as lows upstream of blocking (e.g. Shutts, 1983; Pfahl et al., 2015) or heat lows (e.g. Fischer et al., 2007). The dynamical systems based persistence metric reflects these differences and allows for a nuanced analysis of the persistence of the atmospheric circulation at different atmospheric levels.

A limitation of this study is that the dynamical systems analysis approach assumes a stationary system. Whilst our analysis of the analogue quality in Fig. A4 suggests that atmospheric data for the recent decades provides high-quality analogues, a recent study by Faranda et al. (2023) found that the frequency of some circulation patterns has changed over the time period 1950–2021, although the study did not separate anthropogenic effects and those from natural variability. We could thus conjecture that we may soon approaching a point where anthropogenic warming will have a sufficiently strong signal such that atmospheric analogues no longer follow the theoretical limit distribution, and analyses based on atmospheric analogues will explicitly need to account for the climate system's non-stationarity.

One can also consider the criterion of persistence and its link or lack thereof to heatwaves and warm spells in terms of the underlying dynamics and their relevant spatial scales. Wintertime warm spells have previously been associated with oceanic warm air advection (Slonosky and Yiou, 2002; Messori et al., 2017). Figs. 3 and 4 indeed show anomalously persistent surface level configurations and advection being linked to warm spells in Scandinavia and Germany, whilst in the Mediterranean, Iberia and the British Isles we detect anomalous surface level advection, but no significant signal for anomalously persistent configurations being linked with warm spells. We suggest that this could be because warm oceanic air does not need to travel far to lead to warm advection in these regions. Our results further provide no evidence that warm spells in Russia are linked with persistent surface level atmospheric configurations, and show only weak surface advection signals, suggesting that other mechanisms, such as subsidence, could contribute to wintertime warm spells in this region.

Literature discussing the drivers of heatwaves during summer mentions radiative forcing, subsidence and soil moisture as key factors (Pfahl, 2014; Zschenderlein et al., 2019). Our results show a clear radiative signal (implied by upward sensible heat flux anomalies) associated with summertime heatwaves, while surface temperature advection appears to play a smaller role than during wintertime warm spells. This is supported by Zschenderlein et al. (2019), who also claim that warm temperature

advection plays a weak role in summertime heatwaves. We see warm temperature advection at the mid-level in Fig. 5, suggesting that warm air is possibly being transported higher up in the atmosphere before descending to the location of the heatwave while undergoing adiabatic warming. The effect of subsidence in generating large positive temperature anomalies was investigated by de Villiers (2020), who focused on on the late July 2019 European heatwave. Domeisen et al. (2023) also note the contributing role of subsidence in driving heatwaves, with Röthlisberger and Papritz (2023) highlighting the dominating role of adiabatic warming of air due to subsidence in driving heatwaves in southern Europe and the Mediterranean. This is consistent with our results showing a closer link of heatwaves to the persistence of mid-level rather than surface circulation patterns. We thus suggest that heatwaves do not require highly persistent large-scale surface level atmospheric configurations, while there appears to be a significant association at the mid level. We contrast this to warm spells driven by surface advection, such as in Germany or Scandinavia, for which the large-scale structure of the atmosphere must remain qualitatively similar for longer periods of time. The investigation of the relationship with soil moisture, whilst interesting, may prove to be a rather complex undertaking in a dynamical systems framework due to the different time scale on which it operates relative to atmospheric motions, and is left to future studies.

We conclude our discussion by considering two case studies, namely the July 1991 and July 1992 heatwaves in Germany. These were selected as they displayed the single days with some of the most negative and positive $\theta$ anomalies for both Z500 and SLP respectively, for heatwaves in the region, and thus make for visually immediate example. Figs. 7 and 8 show the Z500 and SLP anomaly configurations, respectively, for the two identified days in the top row and one day later in the bottom row. For the persistent case, the large-scale positive Z500 anomaly and smaller negative cores remain stable over the period considered. For the non-persistent case, there is a considerable spatial evolution of the negative Z500 anomaly features in the southern part of the domain. A similar visual appraisal holds for SLP anomalies. To support this quantitatively, we note that the Euclidean norm for each pair of days is lower for the high persistence cases than for the low persistence ones, for both Z500 and SLP. Notwithstanding the differences in persistence, a visual appraisal suggests that for both heatwaves a blocking algorithm would likely detect a blocked flow pattern, namely a reversal of the meridional Z500 gradient Fig. A12.

These two case studies highlight that it is possible for heatwaves to occur during either high or low persistence atmospheric configurations, and that the persistence as quantified using dynamical systems theory can be partly related to the intuitive concept of 'similarity' of spatial atmospheric configurations. This supports our conclusion that highly persistent configurations are not a necessary criterion for European summertime heatwaves to occur.

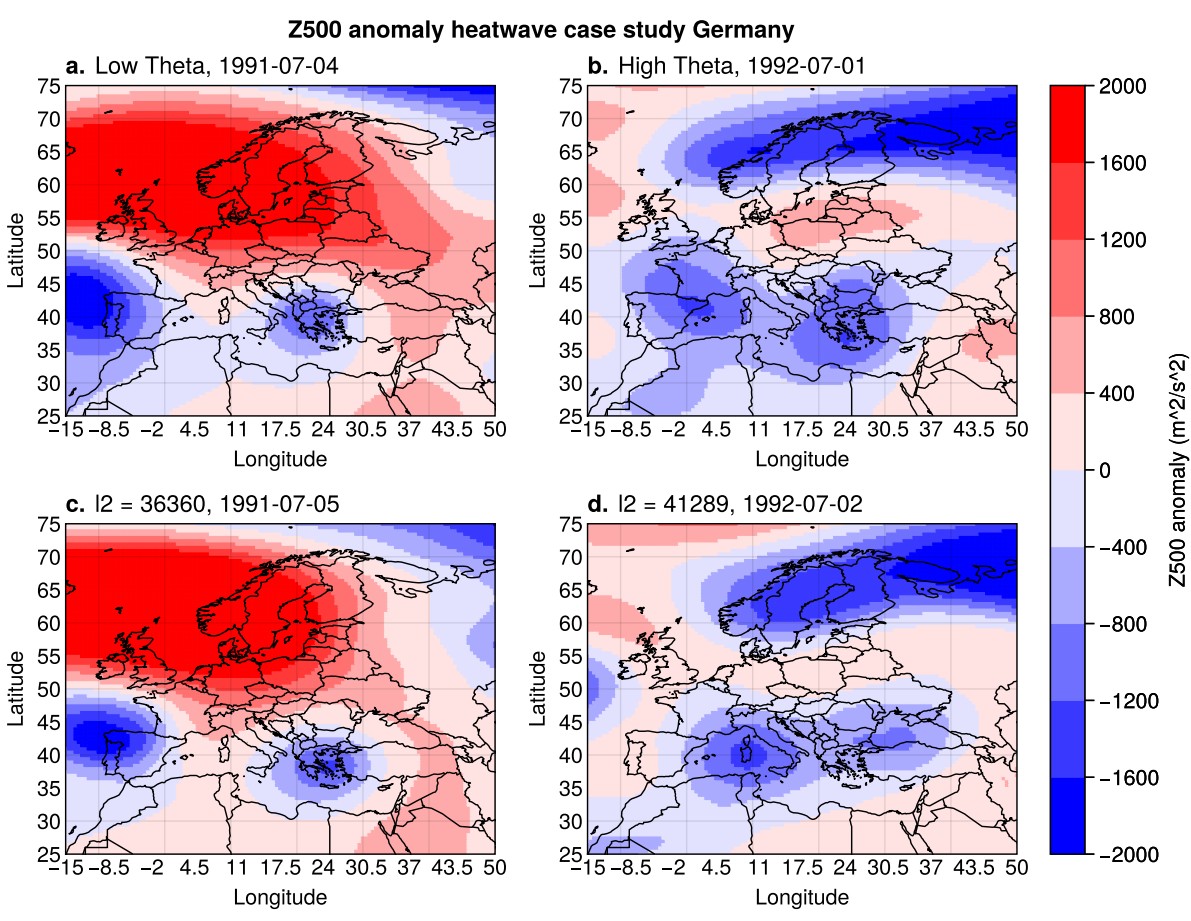

**Figure 7.** Z500 $[m^2/s^2]$ anomaly maps corresponding to the days with some of the highest and lowest $\theta$ anomalies within all days which are classified as days where there is a heatwave in the region Germany. Panel titles in (c), (d) show a lead time of 1 day to the respective high and low $\theta$ heatwave days, with the values corresponding to the euclidean distance between the initial and 1 day lead Z500 maps for each case.

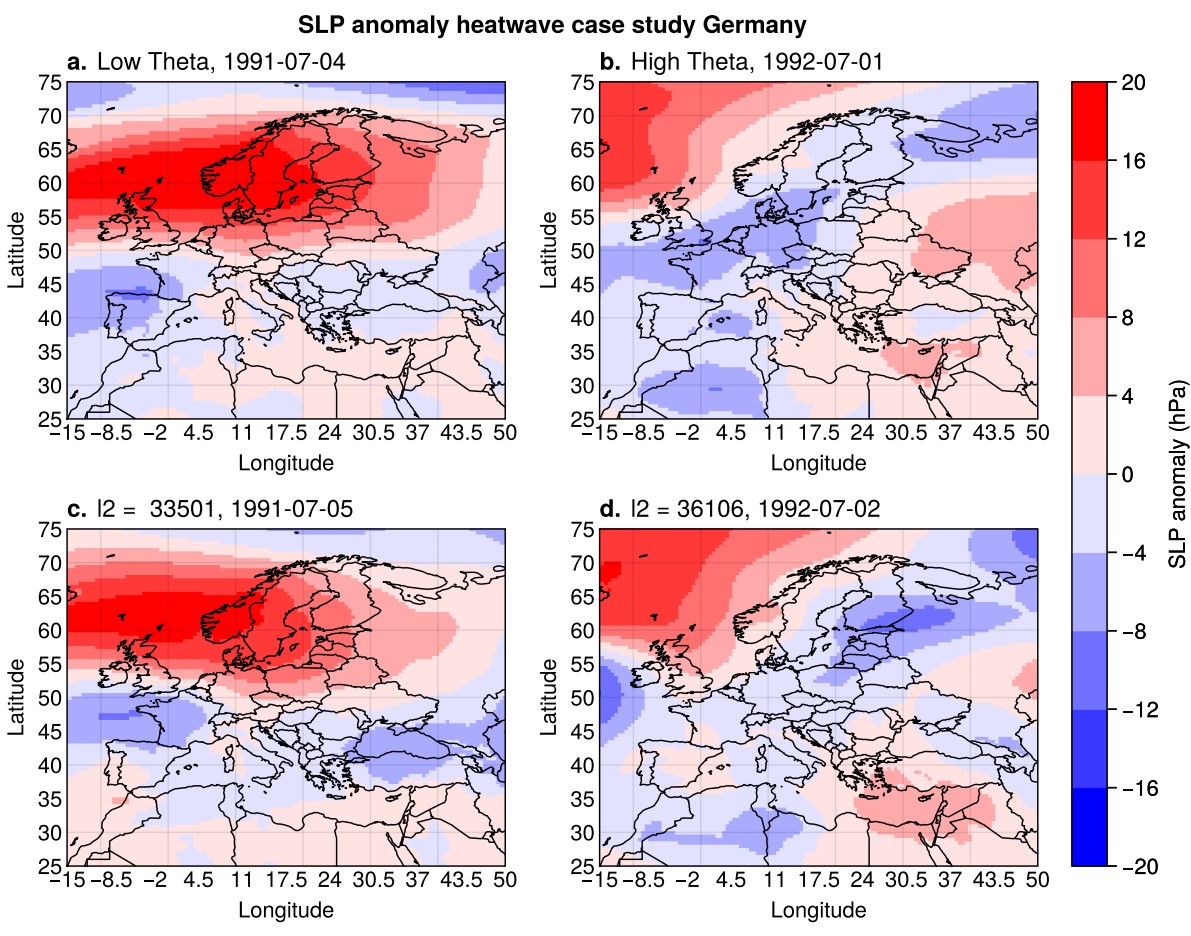

**Figure 8.** SLP $[hPa]$ anomaly maps corresponding to the days with some of the highest and lowest $\theta$ anomalies within all days which are classified as days where there is a heatwave in the region Germany. Panel titles in (c), (d) show a lead time of 1 day to the respective high and low $\theta$ heatwave days, with the values corresponding to the euclidean distance between the initial and 1 day lead SLP maps for each case.

# 5 Conclusions

We have analysed the persistence of the atmospheric circulation patterns at surface and mid-levels associated with European summertime heatwaves and wintertime warm spells. We define atmospheric persistence using a robust quantitative approach issued from dynamical systems theory. Our results point to wintertime warm spells being associated with anomalously persistent surface level atmospheric configurations, although the anomalies are mostly not statistically significant when aggregated at regional level. This partly supports a previous investigation on the topic (Messori et al., 2017). During summer, we find that heatwaves, on average, show almost no link to surface persistence, while most regions show a significant link to enhanced mid-level atmospheric persistence. Nonetheless, the magnitude of the mid-level persistence anomalies is weak and appears at odds with the notion of heatwaves driven by atmospheric blocks persisting over several days to weeks. Our findings are consistent with recent work by Lucarini and Gritsun (2020) who also identified blocking as not necessarily being a persistent configuration.

We ascribe the different atmospheric persistence characteristics of summertime versus wintertime warm temperature extremes to their different driving mechanisms. We found a far weaker signal for surface temperature advection in summer than in winter. This is consistent with the results of Zschenderlein et al. (2019) and Bieli et al. (2015), who investigated summer and the full year respectively, and suggested that advection is not a dominating driver of heatwaves in summer. We indeed found a strong radiative signal for summertime heatwaves.

We have further attempted to clarify the discrepancy between the dynamical systems and conventional views of atmospheric persistence associated with heatwaves. The dynamical systems persistence considers a fixed spatial domain, and has a relatively stringent definition of what 'similar' means in terms of atmospheric configurations. Such a dynamical systems definition is thus more sensitive than alternative definitions to changes in the regional atmospheric configuration.

## Appendix A: Sensitivity Analysis

We report here sensitivity tests of our results to the choice of definition of warm temperature extremes, geographical domain, and definition of atmospheric analogues.

Sensitivity to heatwave/ warm spell duration was tested by considering 3 and 7 day minimum duration requirements, see Figs. A1, A2. For the three day minimum duration we find no notable change in persistence at the surface level during summer and the mid level during winter. At the surface level during winter the Iberia and British Isles regions show negative significant anomalies in $\theta$, whilst Germany, Iberia and the Mediterranean no longer show significant negative anomalies at the mid level in Summer. For the seven day minimum duration we see no change at the mid level during winter, whilst Russia and the British Isles show significant negative $\theta$ anomalies at the surface level. During summer, Russia shows a significant negative anomaly at the mid level whilst the British Isles no longer show a significant anomaly, although the quantitative difference is small. The sensitivity to the temperature threshold was tested by considering temperatures above the 95th percentile, see Fig. A3. The only qualitative persistence differences were found for Iberia at the mid level and the British Isles at the surface level in winter, where negative $\theta$ anomalies could be seen, and Iberia and Russia at the mid level in summer, where significant negative $\theta$ anomalies could and could no longer be seen respectively. We note that the quantitative differences appear quite small also in this case.

The sensitivity of the results to the geographical domain used to calculate $\theta$ was tested. For a westward-shifted domain (Fig. A5) significant negative $\theta$ anomalies emerged for Iberia during winter at both surface and mid levels and for the Mediterranean at the surface level. Moreover, during summer the British Isles and Mediterranean no longer displayed significant negative anomalies at surface and mid levels respectively. When considering a smaller domain (Fig. A6) Russia and the Mediterranean showed significant $\theta$ anomalies at surface level during winter. When considering a larger domain (Fig. A7) Russia and the Mediterranean showed significant anomalies at surface level and Iberia displayed a significant anomaly at the mid level, during winter.

A time series of the shape factor of the fit for the Z500 analogues can be seen in Fig. A4. This shows values very close to 0, suggesting that trends due to warming do not have a major effect on the overall quality of the analogues. We further investigate sensitivity to the definition of analogues – or in other words to the number of recurrences selected during the computation of $\theta$. This was tested by considering $q = 0.97$ and $q = 0.99$, shown in Figs. A8 and A9 respectively. For $q = 0.97$, Russia shows a statistically significant negative $\theta$ anomaly during wintertime warm spells at the surface level, whilst Iberia no longer shows a significant anomaly at the mid level during Summer. For $q = 0.99$, Germany no longer shows a significant anomaly at the surface level during winter, although it does show a significant negative anomaly during summer at the surface level. In addition, Iberia and the Mediterranean no longer show significant anomalies at the mid level.

In summary, we find some sensitivity of our results to our methodological choices. None of the sensitivity tests, however, affect our qualitative conclusion that anomalously high atmospheric persistence is not a necessary requirement for summertime heatwaves. We show a composite of the variance of Z500 anomalies for days during heatwaves/ warm spells in Fig. A11. This provides some indication of local persistence, although we highlight that this figure should be interpreted with care due to the local nature of this analysis in contrast to the analysis conducted in the body of this paper, which considers atmospheric variability across the entire domain.

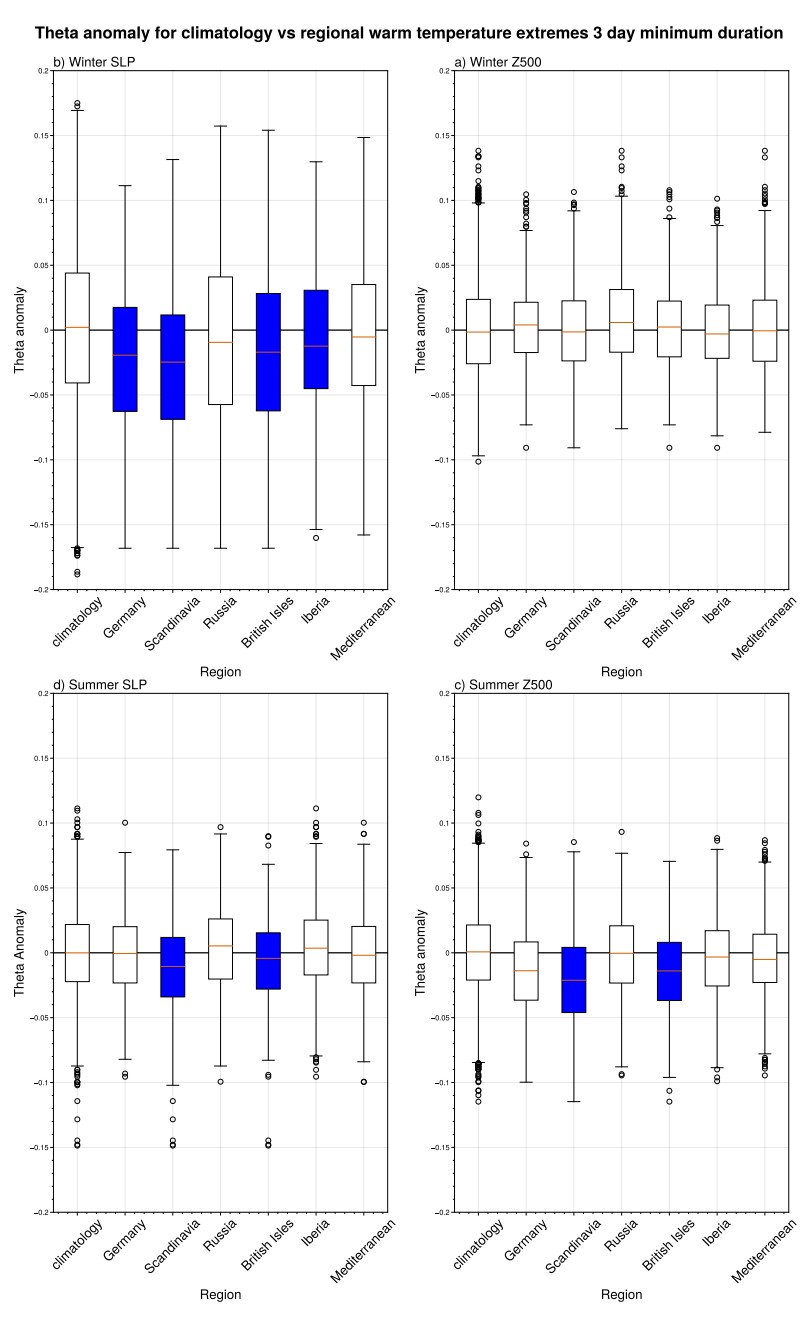

**Figure A1.** $\theta$ [$days^{-1}$] anomaly box plots as in Fig. 3 but for a heat wave or warm spell of at least 3 days in duration.

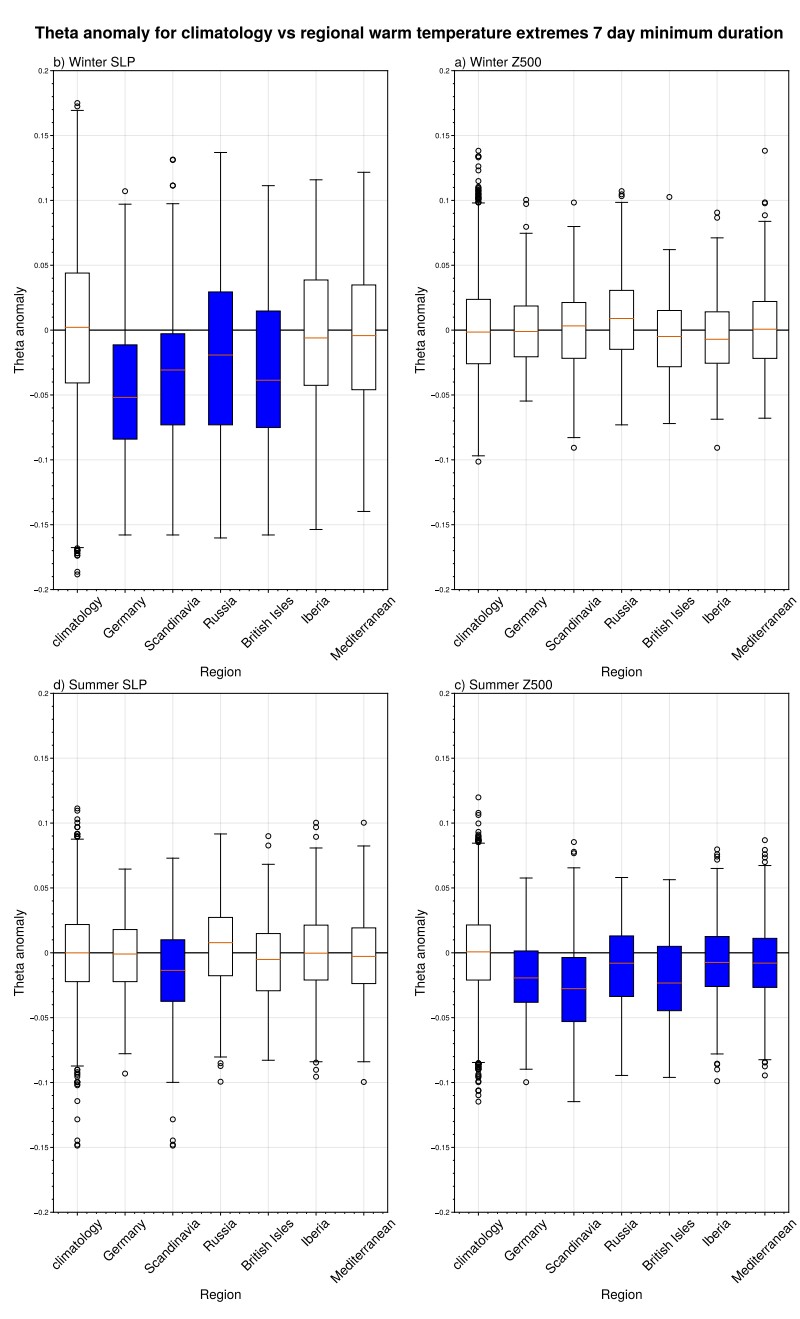

**Figure A2.** $\theta$ $[days^{-1}]$ anomaly box plots as in Fig. 3 but for a heat wave or warm spell of at least 7 days in duration.

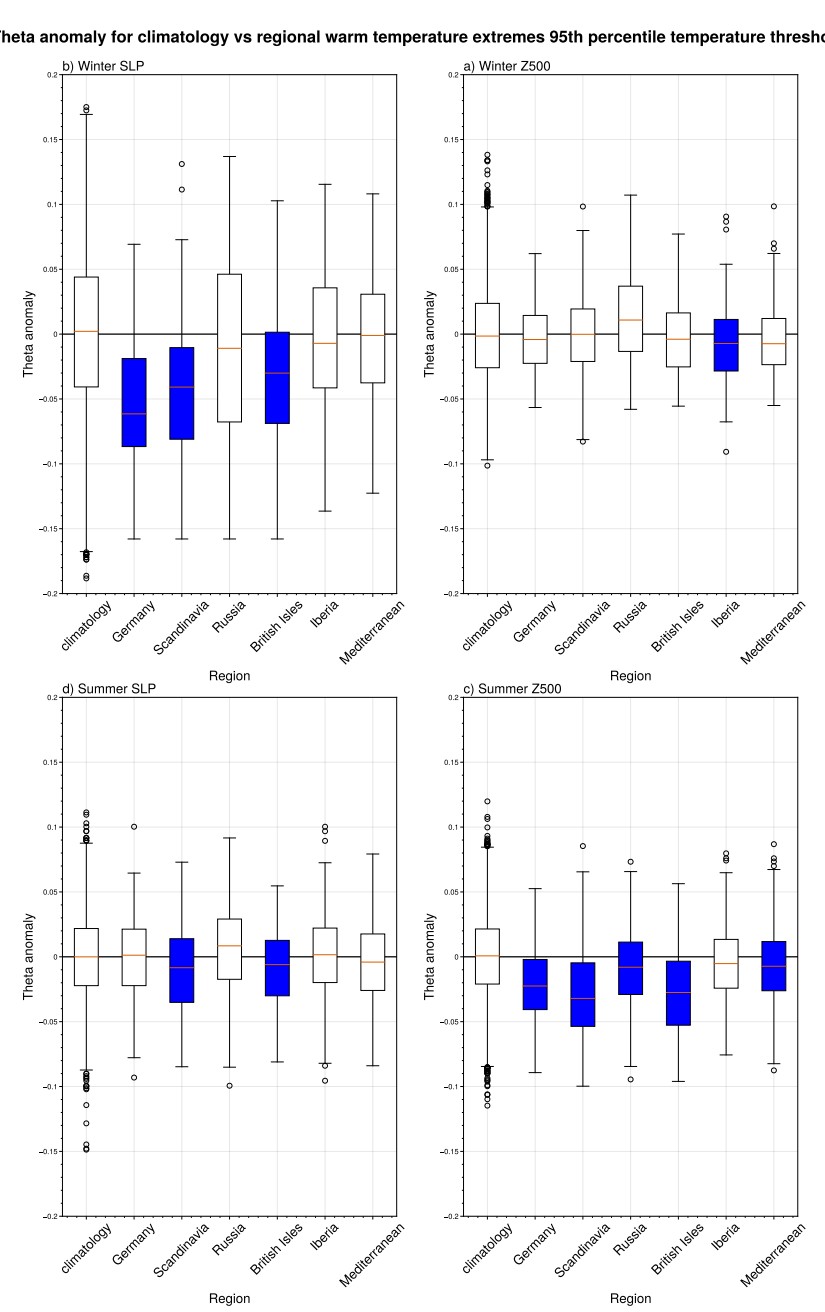

**Figure A3.** $\theta$ $[days^{-1}]$ anomaly box plots as in Fig. 3 but for warm temperature extremes defined using temperatures above the local $95^{th}$ percentile.

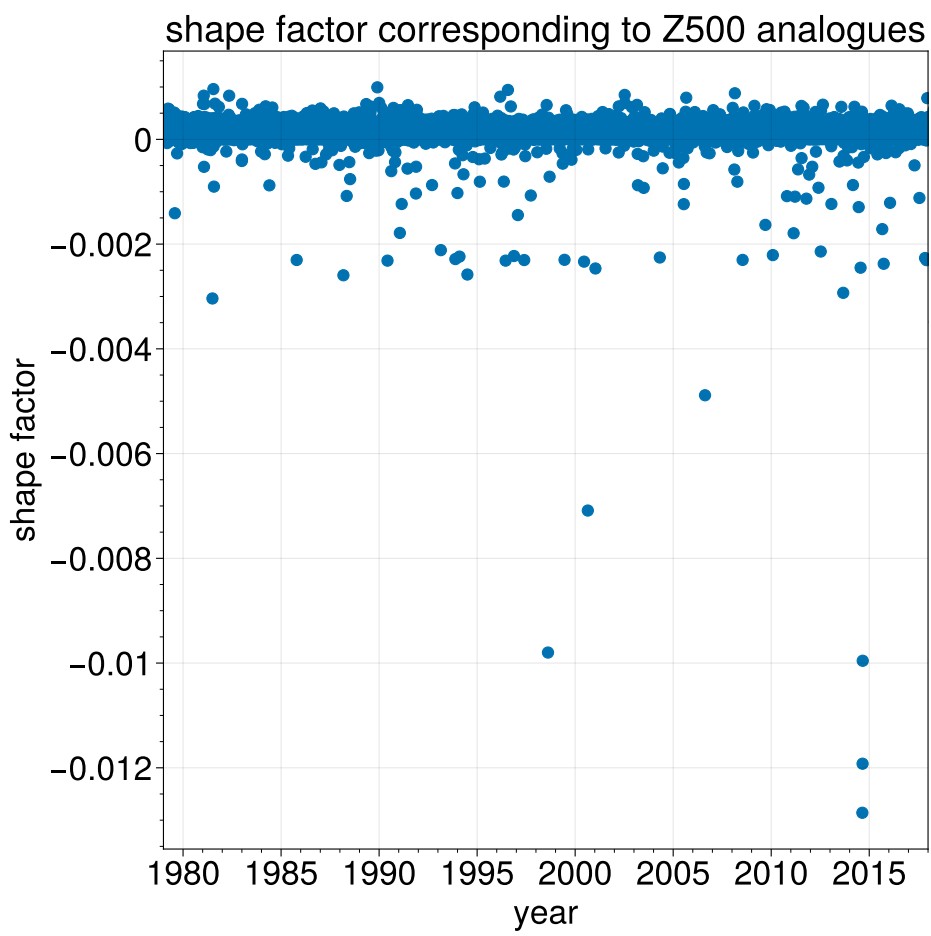

**Figure A4.** Shape factor of the distribution fitted to the analogues. Values approximately equal to 0 indicate that the analogues are distributed according to the statistical assumptions of our analysis method or, in other words, that the analogues are of good quality.

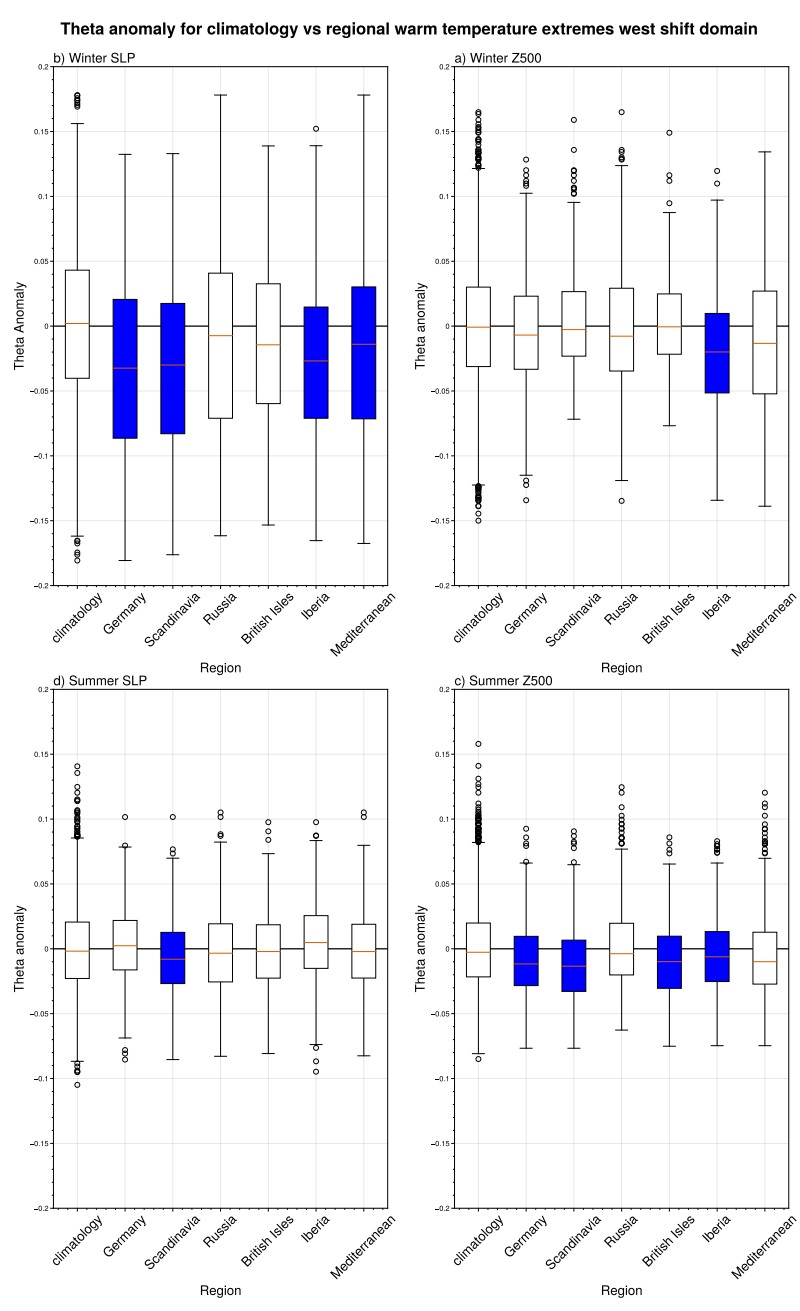

**Figure A5.** $\theta$ [$days^{-1}$] anomaly box plots as in Fig. 3 but with a westward shifted domain covering -50°–35°E.

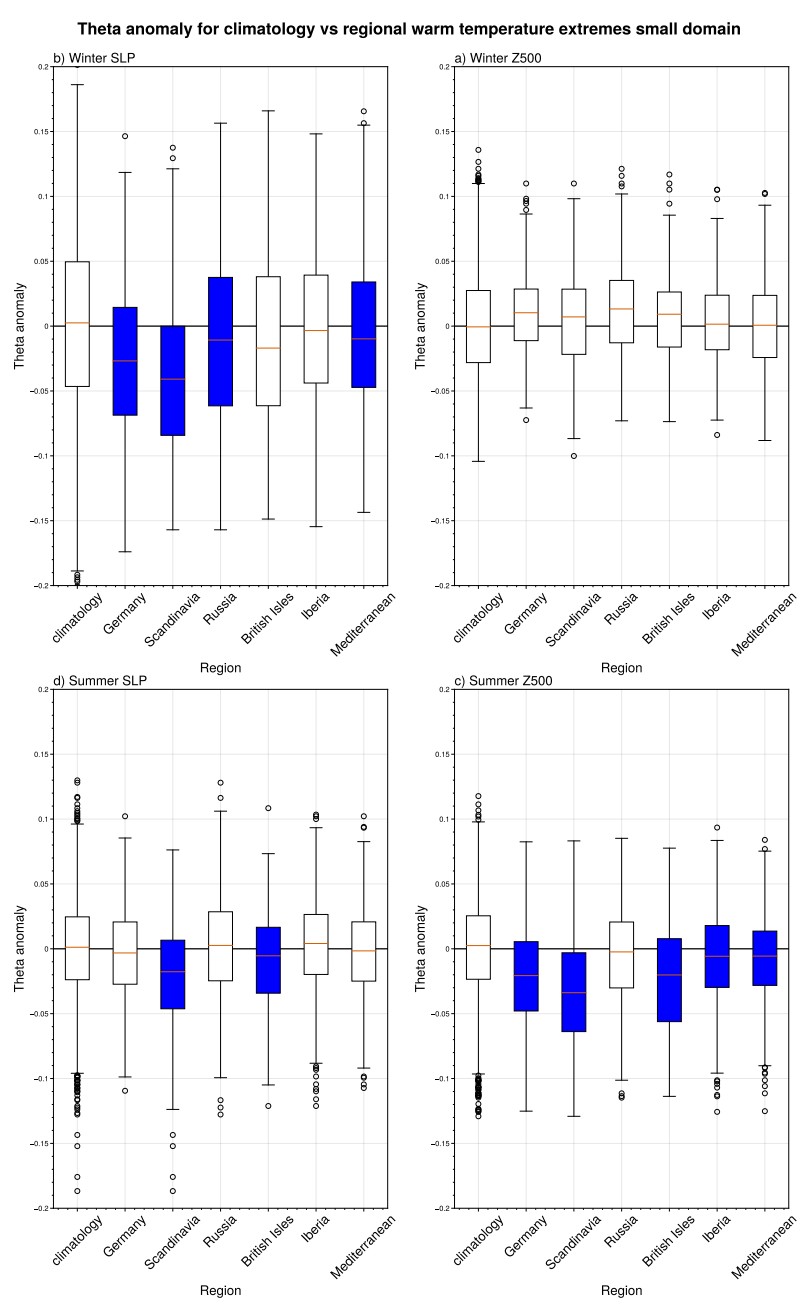

**Figure A6.** $\theta$ [$days^{-1}$] anomaly box plots as in Fig. 3 but with a smaller domain covering -10°–45°E.

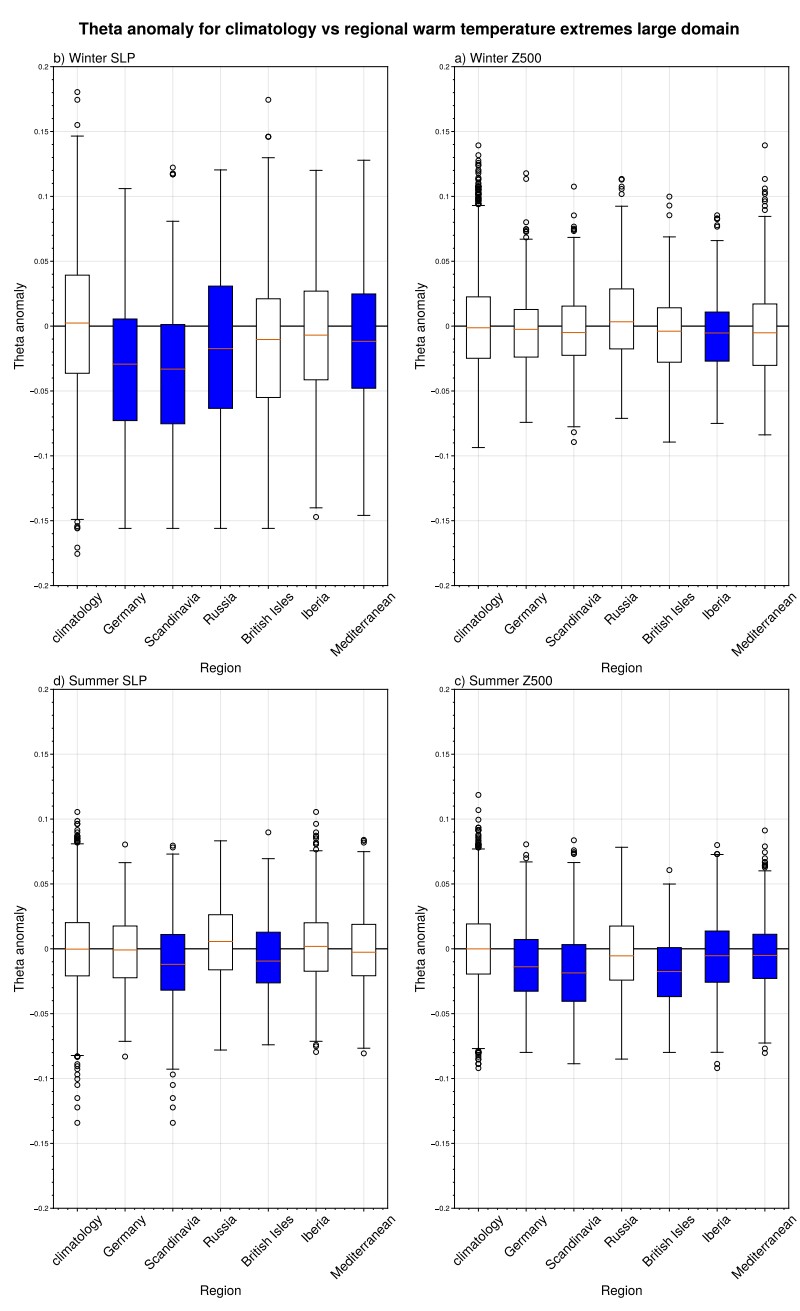

**Figure A7.** $\theta$ [$days^{-1}$] anomaly box plots as in Fig. 3 but with a larger domain covering -20°E–55°E.

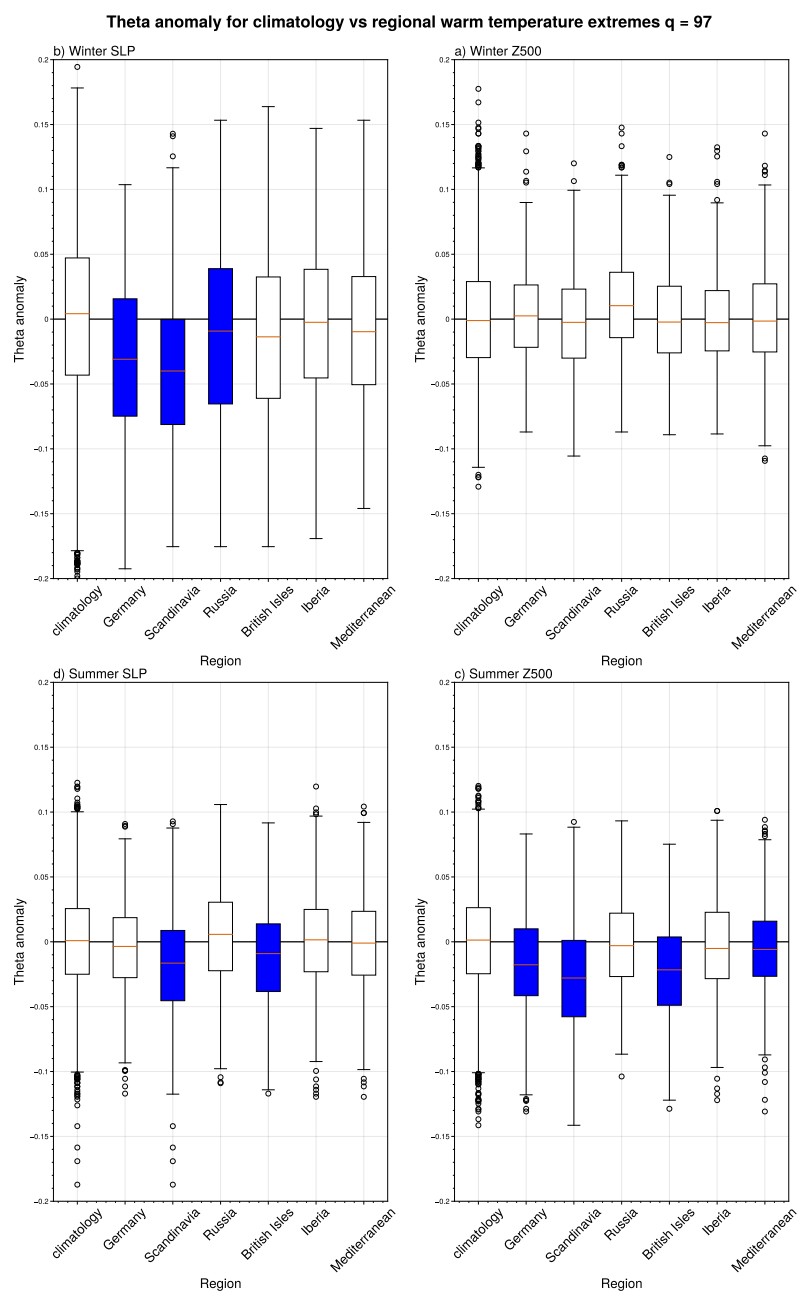

**Figure A8.** $\theta$ $[days^{-1}]$ anomaly box plots as in Fig. 3 but using the $97^{th}$ percentile of the distribution of $-log$ distances to define analogues.

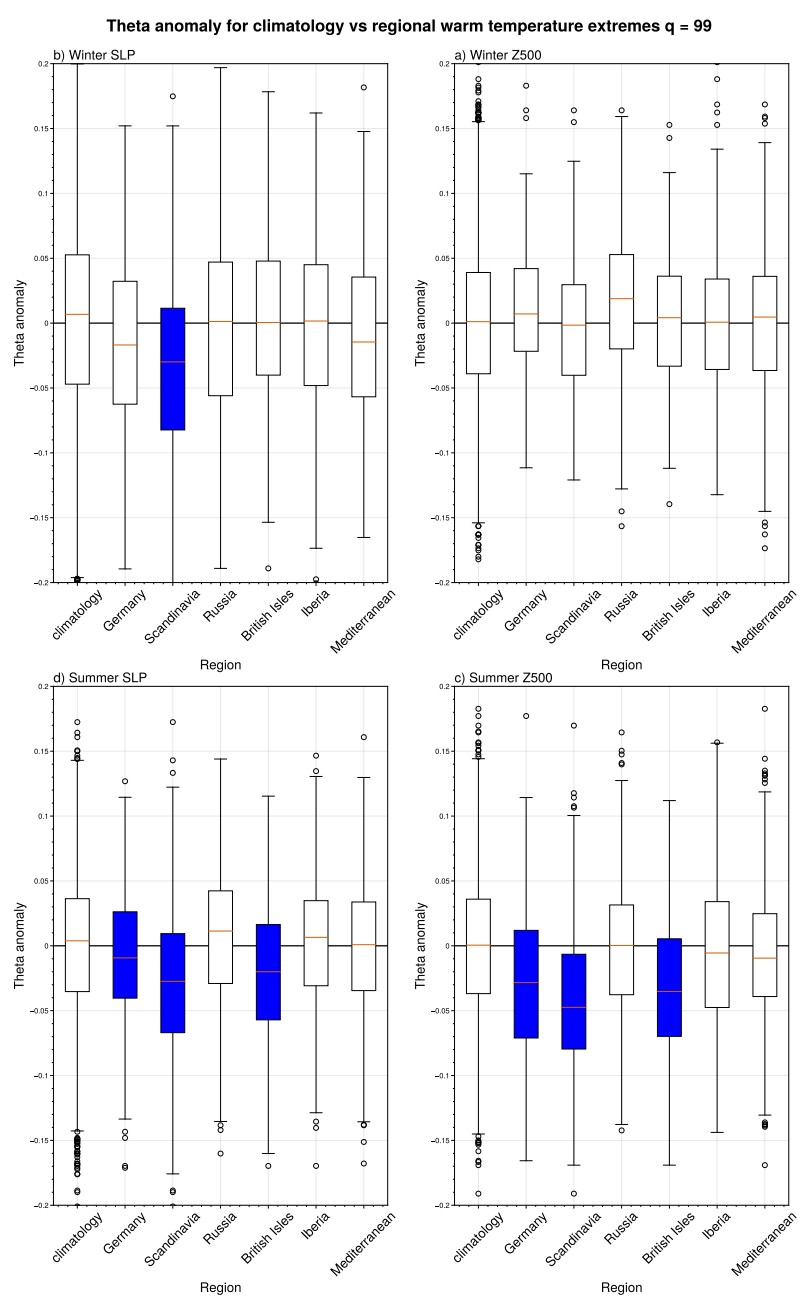

**Figure A9.** $\theta$ $[days^{-1}]$ anomaly box plots as in Fig. 3 but using the $99^{th}$ percentile of the distribution of $-log$ distances to define analogues.

**SLP anomaly warm temperature extremes all days**

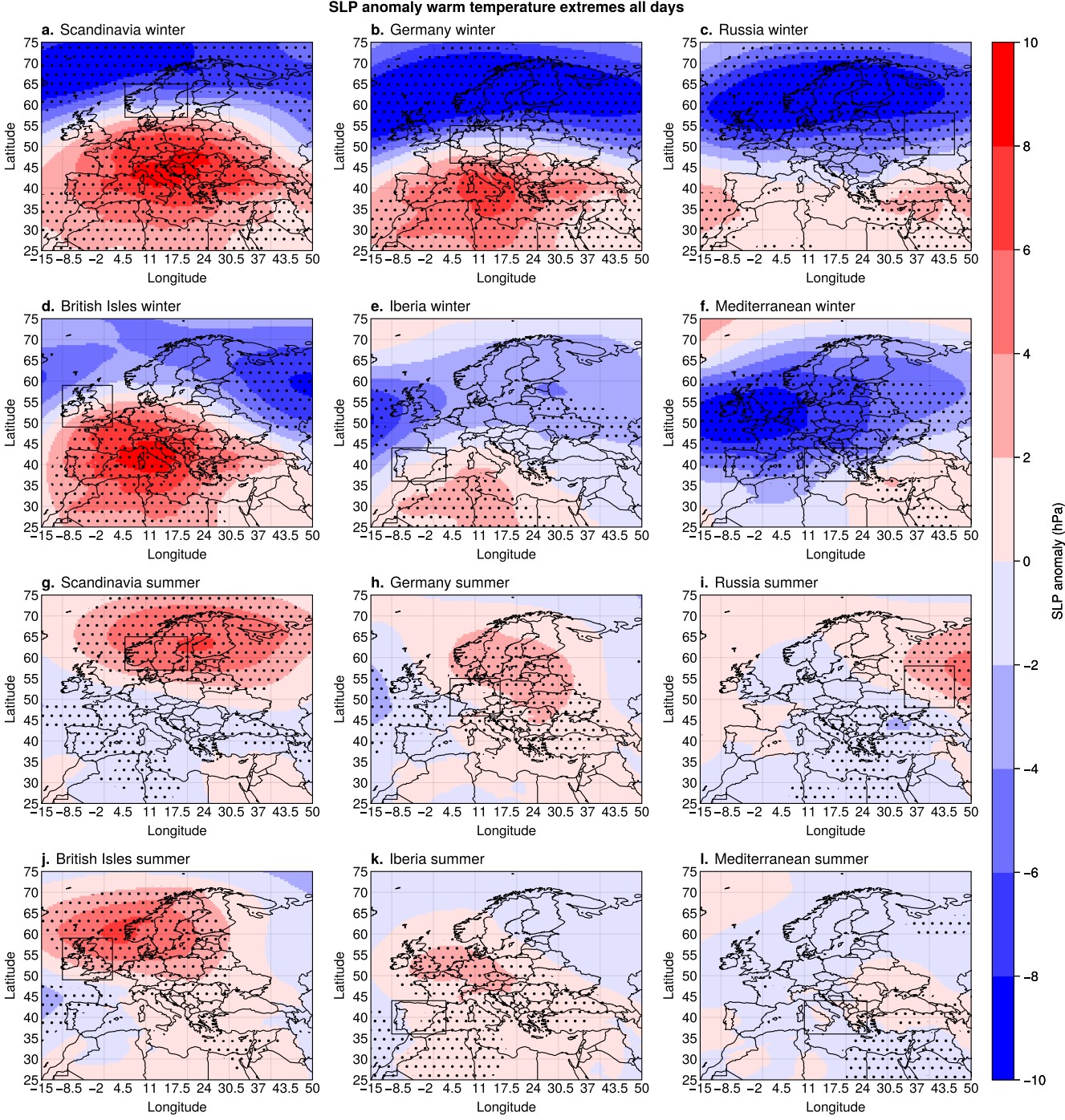

**Figure A10.** Composite SLP [$hPa$] anomalies during warm spell/ heatwave days in (a,g) Scandinavia, (b, h) Germany, (c, i) Russia, (d, j) British Isles, (e, k) Iberia, (f, l) Mediterranean, during winter (a–f) and summer (g–l). Statistical significance is assessed as described in Section 2 and shown with stippling.

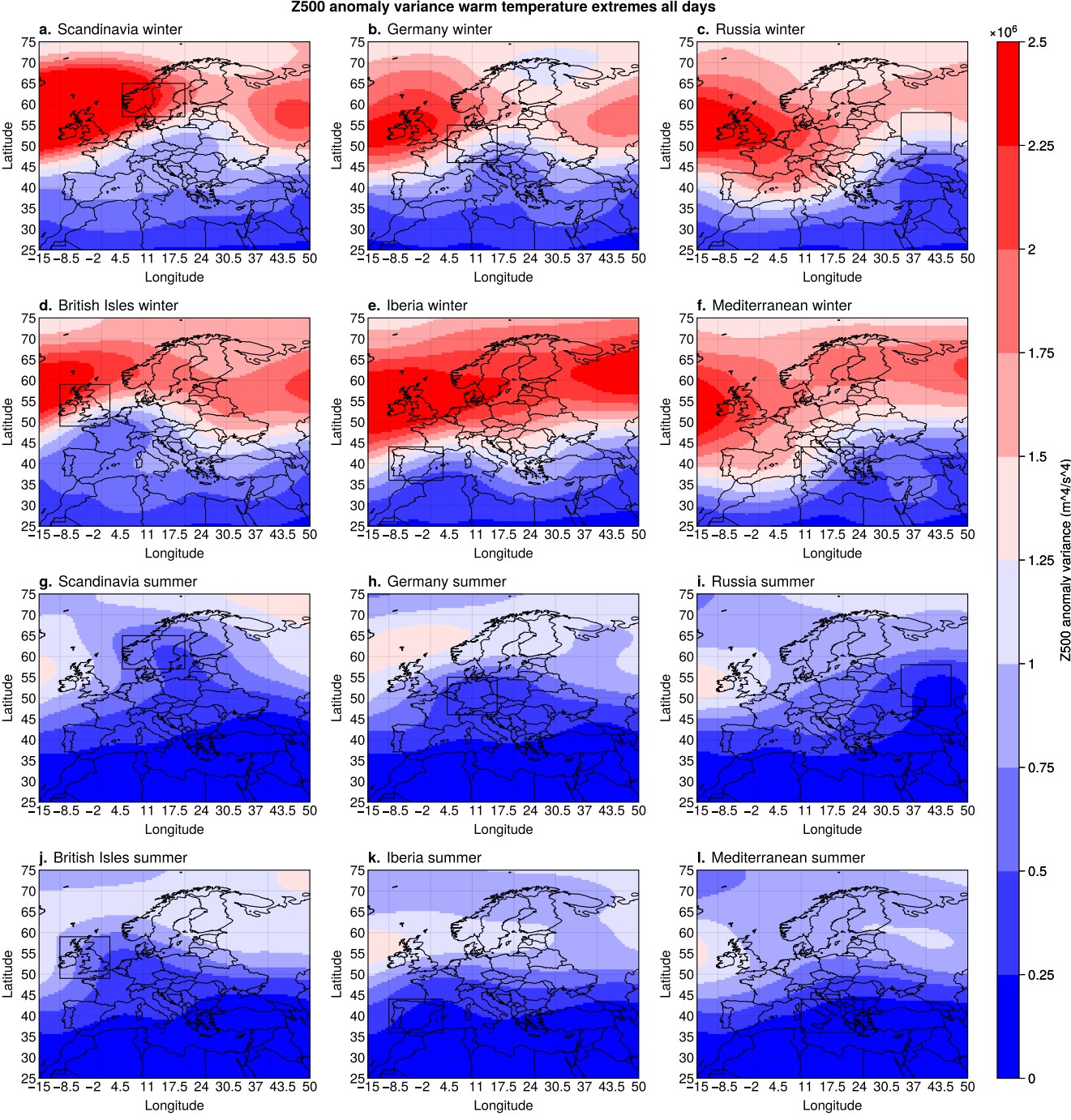

**Figure A11.** Composite Z500 $[m^4 s^{-4}]$ variance during warm spell/ heatwave days in (a,g) Scandinavia, (b, h) Germany, (c, i) Russia, (d, j) British Isles, (e, k) Iberia, (f, l) Mediterranean, during winter (a–f) and summer (g–l).

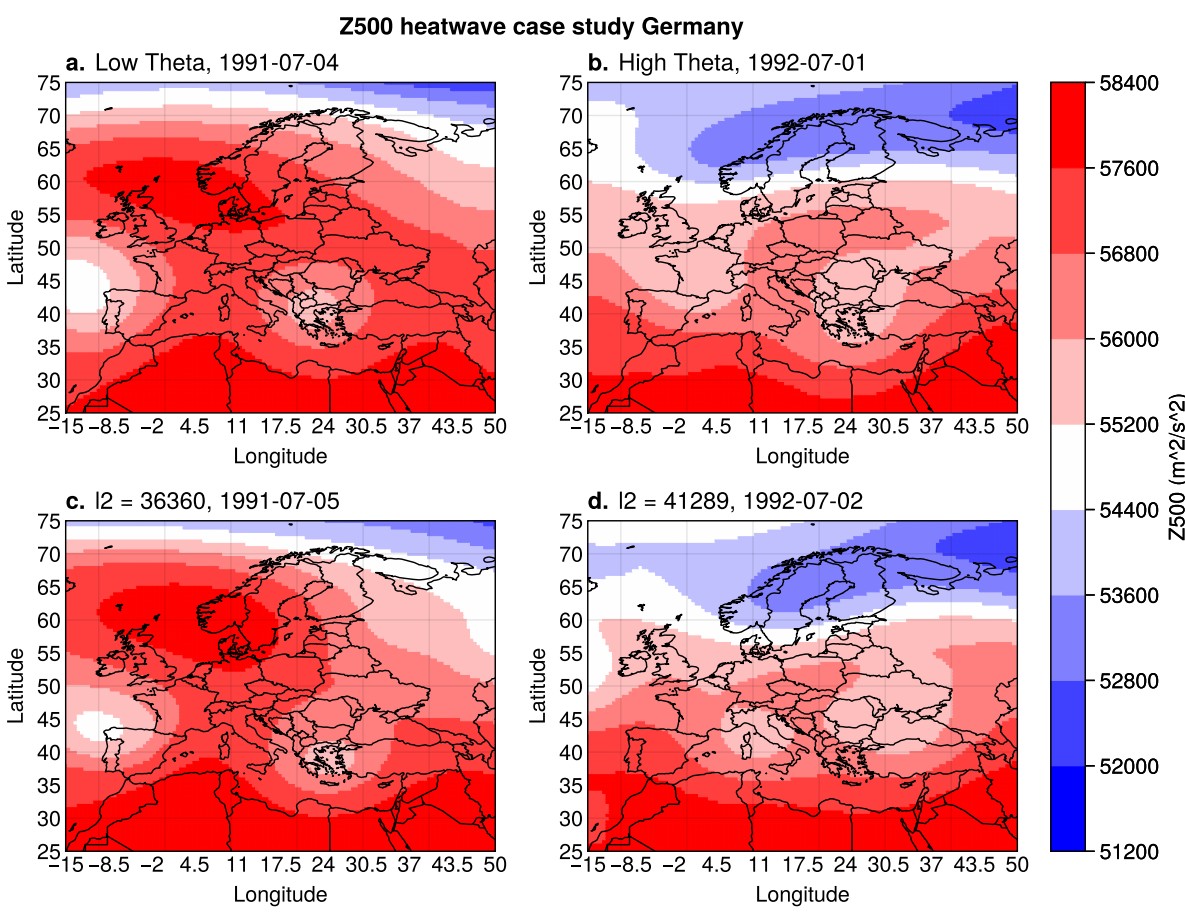

**Figure A12.** Z500 [$m^2/s^2$] maps corresponding to the days with some of the highest and lowest $\theta$ anomalies within all days which are classified as days where there is a heatwave in the region Germany. Panel titles in (c), (d) show a lead time of 1 days to the respective high and low $\theta$ heatwave days, with the values corresponding to the euclidean distance between the initial and 1 day lead SLP maps for each case.

*Author contributions.* EH and GM designed the study. EH carried out the analysis and drafted the first version of the manuscript. All authors contributed to structuring the analysis and reviewing the manuscript.

*Competing interests.* One author is a member of the editorial board of this journal, Earth System Dynamics. The peer-review process was guided by an independent editor, and the authors have no other competing interests to declare.

*Acknowledgements.* This work was supported by the European Union's Horizon 2020 research and innovation programme under the Marie Skłodowska-Curie grant agreement No. 956396 (ITN project EDIPI – European weather extremes: drivers, predictability and impacts). The authors would like to thank the anonymous reviewers and editor for their support in improving the quality of this study. EH would also like
to thank Jacopo Riboldi for helpful discussions.

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
