# Peer review of "The link between European warm temperature extremes and atmospheric persistence"

_Earth System Dynamics, 2022_

## Referee Comment (RC2)

Review of paper

**The counter-intuitive link between European heatwaves and atmospheric persistence**

by Emma Holmberg et al.

submitted to *Earth System Dynamics*

Holmberg et al. examine the persistence of the European-wide SLP field during summer heat waves and winter warm spells using a previously developed measure of persistence that is based on well-established concepts in dynamical systems theory. The authors find that winter warm spells are associated with increased persistence (compared to climatological persistence values derived from all days) while for summer heat waves persistence anomalies are found to be moderately positive during heat waves in Scandinavia and even negative during heat waves in other regions. The authors identify in particular the absence of positive persistence anomalies for summer heat waves as "counter intuitive". The study is overall well written and the figures are, for the most part, clear. Also, I appreciate that authors' attempt to leverage concepts from dynamical systems theory to put the notion of "circulation persistence" onto more solid theoretical foundations. In that sense, I would like to encourage the authors to continue exploring their avenue of research.

However, despite the good intention and interesting overarching research goal of this study, there are a considerable number of issues that significantly compromise the value of this paper in its current form. I therefore suggest this study undergoes major revisions before it can be re-evaluated again. Some of my major comments below relate to the presentation of the methodology and I expect the authors will be able to address those rather easily. However, I also see more fundamental issues related to the authors choice of meteorological analysis techniques as well as to their interpretation of their findings.

**Major comments**

A) The description of how "persistence" is defined and computed in Section 2.3 is not detailed enough to convey a physical intuition of what the authors exactly mean by persistence. I appreciate that the authors employ a rigorous and mathematical definition of persistence and I am aware that previous studies already introduced this definition of persistence. However, in some instances the authors challenge extremely well accepted reasoning within the atmospheric dynamics community (Blocking patterns during European heat waves are persistent flow features) and thus they should provide sufficient detail and explanations such that peers from this and other fields (i.e., scientists outside dynamical systems research community) can understand what exactly is meant by "persistence" without first consulting other papers. For instance, why exactly is the unit of theta days$^{-1}$? The frequency of what is it exactly? Moreover, is its inverse something like an e-folding

time of some quantity? Or on lines 127–128: Why exactly is a generalized Pareto distribution fitted to the negative log of the analogue distances? Or on lines 129–131: What is the method of Süveges (2007)? And after all: How does a measure of clustering in extreme value theory (L69–70) inform about "persistence"? Note that I am not questioning your approach, based on the current Section 2.3 it just appears overly opaque to me.

B) Related to A): On line 125 you state that you choose the 5% of days with the smallest distances as analogues to any day of interest. However, presumably not all of these analogues occurred during heat waves/warm spells. It is unclear to me how that fact affects the interpretation of your results. I read in your reply to the Community Comment by Dr. Alexandre Tuel that your methodology characterizes "local properties of the attractor". I don't fully understand the exact meaning of that statement, but it nevertheless appears plausible to me that a certain large-scale circulation (defined by its European-wide SLP pattern) can be persistent in some cases, but much less so in others. That is, it is quite plausible that heat waves occur in a unusually persistent manifestations of a given large-scale circulation pattern, that not always exhibits this level of persistence. For instance, previous studies have shown that soil moisture anomalies associated with heat waves can have an "anchoring effect" on the associated anticyclonic circulation (e.g., Martius et al., 2021), which only affects the persistence of a given circulation pattern when a heat wave and the associated soil moisture anomaly occurs, but not in situations in which a similar circulation pattern occurs without a heat wave. How do the authors ensure that the persistence they quantify from sets of days including both heat wave and non-heat wave days is representative for heat wave days only? In case this comment simply results from a misunderstanding of your approach, I strongly recommend rewording the description of your approach.

C) A key finding of this study is that during summer heat waves in many parts of Europe, the circulation is not anomalously persistent and the authors claim that this is at odds to the well accepted notion that European summer heat waves (at least in Central and Northern Europe) often occur in association with "persistent blocking" (e.g., lines 157–158, or 180–182). However, a contradiction is not particularly apparent to me here. It is well possible that zonal flows (which often lead to wet and thus not anomalously hot summer conditions across central Europe) are more persistent by the author's metric than blocked flows. Nevertheless it may still be true that severe summer heat waves occur preferentially during the most persistent blocking episodes (e.g., lines 42–44 and references cited there). I therefore suggest that the authors more clearly frame their research question and better explain what exactly is counter-intuitive about their findings.

D) I believe Fig. 4 and the quantity B does not help to understand the contribution of large-scale warm air advection to heat waves/warm spells, for two reasons: Firstly, it identifies essentially just regions of gradients in the surface elevation. The vector **B** is computed from

SLP (which everywhere is indicating the pressure *at sea-level*), while two-meter temperature is dependent on the height of the terrain and thus large horizontal (i.e., terrain following) gradients of two-meter temperature are first and foremost indicating steep terrain. Second, the implicit assumption motivating the definition of $B$ is presumably that by the geostrophic balance the gradient of SLP is indicative of the surface winds. However, especially near the surface the geostrophic balance is often quite substantially disturbed by surface drag and the near-surface flow is not as close to geostrophic as the upper-level flow. Therefore, I believe the authors should repeat their analysis and either consider horizontal temperature advection (i.e., $-\mathbf{v}\nabla T$) on a near-surface pressure level explicitly or, alternatively, consider the advection of potential temperature by the 10m winds.

E) The choice of SLP as variable to quantify the persistence of the large-scale circulation appears peculiar and suboptimal to me, in particular in the context of this study, which often makes reference to atmospheric blocking. Firstly, it is well known that during heat waves so-called heat lows can develop and thus the evolution (and hence potentially also the persistence) of the mid- to upper-tropospheric flow may differ substantially from the evolution of the near-surface flow. Secondly, atmospheric blocks often feature rapidly amplifying disturbances on their upstream side, which feed into the blocks and thereby enhance their persistence (Shutts, 1983; Pfahl et al., 2015). In the SLP field these upstream disturbances manifest themselves as extratropical cyclones (i.e., with strong signals in the SLP field). Thus, one does not a priori expect a "persistent" European-wide SLP pattern during persistent atmospheric blocks. Thirdly, it is anyways quite a stretch to argue about the persistence of blocking flows based on SLP, as blocking is predominantly a manifestation of the upper-level flow (Kautz et al., 2022; Woollings et al., 2018). The authors could easily exclude these issues, e.g., by performing their persistence quantification with (de-trended) Z500 fields or, even better, isentropic PV fields rather than SLP fields, as these quantities much more directly inform about the large-scale circulation.

**Minor comments**
1. L5: I do not think your study indeed "reconciles" the dynamical systems and traditional views on persistence. It rather demonstrates that your approach leads to results that diverge from traditional views on persistence.
2. L16–17: The authors should clarify what exactly they mean with the word "extreme". Do they mean "rare", e.g., as on line 20 where the authors refer to "Uncommonly high temperatures", i.e., "extreme" in the sense of a large return period? Or do the authors perhaps mean "extreme" in the sense of "hot enough to cause impact"? If the authors mean "extreme" in the sense of "rare" then the sentence on lines 16–17 is contradictory. If extremely hot summers are defined via their rareness then, by definition, they cannot become more likely. Rather, the temperature corresponding to a certain return period will increase, i.e., events of equal extremeness will become more intense.

3. L43–44: The authors could explain here in more detail how the "current understanding of the link between blocks and summertime heat waves" works physically. How exactly are long-lasting blocks supposed to increase the odds of intense heat waves? Perhaps mention here the adiabatic warming in subsiding air, the clear-sky conditions, drying of soils (due to the persistent local weather within the block), etc.

4. L57–60: The authors refer to a series of papers that developed the Quasi-resonant amplification (QRA) hypothesis. The theoretical basis of this hypothesis has been severely challenged by a number of recent publications (e.g., Wirth and Polster, 2021; Wirth, 2020). So far the authors of the QRA papers have not been able to respond to or address this critique. Therefore, if the authors of this paper decide to refer to the QRA papers they should at least indicate that the validity of the reasoning in these QRA papers is severely contested.

5. L69–71: Here you introduce the concept of persistence from a dynamical systems perspective. I'd find it very helpful if you could add here what "persistence" means in that context, i.e., if you could convey as physical intuition for what that term means in the dynamical systems context and, perhaps also to what extent it is comparable to the persistence of an individual blocking anticyclone identified with the feature-based perspective.

6. L89: Is it really 1978–2018 or 1979–2018? Did you use the ERA5 back extension? If so, why only back to 1978?

7. L91: Maybe 12 UTC instead of "noon"?

8. L93: Something is wrong with the longitudes of your domain.

9. L94: Is "daily climatology" the same as "calendar day climatology"? If so, consider using the latter term.

10. L97: What exactly is "the local distribution"? Do you take any forced trend into account when identifying your events of interest? If not, why is that choice justified in this case?

11. L109–114 and Figs 1, 2, 4 and 5: Given that several of the authors are excellent statisticians I am surprised that you did not apply the False-Discovery-Rate test of Benjamini and Hochberg, (1995) as detailed in Wilks, (2016). Why did you not follow the procedure suggested by Wilks, (2016) even though you clearly perform multiple test simultaneously?

12. L123: Did you include any latitude weighting when computing the Euclidean distances?

13. Figures 1, 2, 4 and 5: The stippling indicating significance is difficult to see. Can you make it better visible? Also, please add the boxes indicating the regions to Figs. 4 and 5, so that the reader always knows what the areas of interest are.

14. Figure 1: I'd be curious to see an analogous figure to Fig. 1 but displaying the Z500 anomaly variance. This would show where (in space) the circulation is (or is not) variable (i.e., not persistent) during your events.

15. L154–155: I wonder to what extent your European wide circulation persistence compares with the persistence of the "local" weather during your events (e.g, measured by the average/median duration of hot spells during your events or by the T2m variance during your events). If you could indeed bring together these two views on persistence then I think the word "reconciling" in your abstract would be justified.

16. Figure 3: The axis labels are difficult to read. Please enlarge them.
17. L165–166: The orographic signatures in Fig. 4 are just a consequence of the definition of B. I do not think we learn anything from these signals about the importance of downslope winds for hot extremes.
18. L171–172: I do not understand this sentence. How exactly is "Russia an exception"?
19. L201–203: The authors refer to "other atmospheric structures". What exactly is meant here?
20. L225–228: I find this statement on the importance of persistence for local (i.e., diabatic) and remote drivers (i.e., advection of air from climatologically warmer regions) of heat waves misleading and suggest to reword this statement. Firstly, in particular for heat waves driven by radiative effects the persistence of (local?) weather (i.e., dry conditions, clear skies) is well known to be important because the amplification of land-atmosphere feedbacks, i.e., soil drying and associated increase in sensible at the expense of latent heat fluxes, take time (e.g., Miralles et al., 2019). Furthermore it is unclear why warm air advection would require persistence. Even if the circulation changes over time, long-range transport of air from climatologically warm regions may occur. Moreover, at least at synoptic scales, near-surface warm air advection tends to be largest ahead of cold fronts which I would not characterize as persistent weather situations. I understand that the authors here refer to the persistence of "the large-scale atmospheric configuration", but as it is written now the statement (i.e., the authors interpretation of their results) is at odds to existing literature and, in my opinion, not sufficiently well substantiated by their results. The issue might be resolved if a distinction could be made between the persistence in the large-scale circulation pattern and the persistence of local weather.
21. L237–239: The "visual appraisal" does not suffice to claim that "a blocking algorithm would detect a blocked flow persisting for several days". Firstly, the large family of blocking identification algorithms derived from the original approach of Tibaldi and Molteni (1990) usually consider gradient reversals in the absolute (not anomaly) Z500 fields, which is not apparent from your Fig. 6. Secondly, the "visual appraisal" in Fig. 6 does not convince me that, e.g., the 5-day persistence criterion for negative upper-level PV anomalies in the Schwierz et al. (2004) algorithm would be fulfilled. I suggest to tone down the interpretation of Fig. 6 in this regard.

**References**

Benjamini, Y. and Hochberg, Y.: Controlling the False Discovery Rate: A practical and powerful approach to multiple testing, J. R. Stat. Soc., 57, 289–300, https://doi.org/10.1111/j.2517-6161.1995.tb02031.x, 1995.

Kautz, L.-A., Martius, O., Pfahl, S., Pinto, J. G., Ramos, A. M., Sousa, P. M., and Woollings, T.: Atmospheric blocking and weather extremes over the Euro-Atlantic sector – a review, Weather Clim. Dyn., 3, 305–336, https://doi.org/10.5194/WCD-3-305-2022, 2022.

Martius, O., Wehrli, K., and Rohrer, M.: Local and Remote Atmospheric Responses to Soil Moisture Anomalies in Australia, J. Clim., 34, 9115–9131, https://doi.org/10.1175/JCLI-D-21-0130.1, 2021.

Masato, G., Hoskins, B. J., and Woollings, T.: Winter and summer Northern Hemisphere blocking in CMIP5 models, J. Clim., 26, 7044–7059, https://doi.org/10.1175/JCLI-D-12-00466.1, 2013.

Miralles, D. G., Gentine, P., Seneviratne, S. I., and Teuling, A. J.: Land–atmospheric feedbacks during droughts and heatwaves: state of the science and current challenges, Ann. N. Y. Acad. Sci., 1436, 19–35, https://doi.org/10.1111/NYAS.13912, 2019.

Pfahl, S., Schwierz, C., Croci-Maspoli, M., Grams, C. M., and Wernli, H.: Importance of latent heat release in ascending air streams for atmospheric blocking, Nat. Geosci., 8, 610–614, https://doi.org/10.1038/ngeo2487, 2015.

Scherrer, S. C., Croci-Maspoli, M., Schwierz, C., and Appenzeller, C.: Two-dimensional indices of atmospheric blocking and their statistical relationship with winter climate patterns in the Euro-Atlantic region, Int. J. Climatol., 26, 233–249, https://doi.org/10.1002/joc.1250, 2006.

Schwierz, C., Croci-Maspoli, M., and Davies, H. C.: Perspicacious indicators of atmospheric blocking, Geophys. Res. Lett., 31, L06125, https://doi.org/10.1029/2003GL019341, 2004.

Shutts, G. J.: The propagation of eddies in diffluent jetstreams: Eddy vorticity forcing of "blocking" flow fields, Q. J. R. Meteorol. Soc., 109, 737–761, https://doi.org/10.1002/qj.49710946204, 1983.

Tibaldi, S. and Molteni, F.: On the operational predictability of blocking, Tellus A, 42, 343–365, https://doi.org/10.1034/J.1600-0870.1990.T01-2-00003.X, 1990.

Wilks, D. S.: "The stippling shows statistically significant grid points": How research results are routinely overstated and overinterpreted, and what to do about it, Bull. Am. Meteorol. Soc., 97, 2263–2273, https://doi.org/10.1175/BAMS-D-15-00267.1, 2016.

Wirth, V.: Waveguidability of idealized midlatitude jets and the limitations of ray tracing theory, Weather Clim. Dyn., 1, 111–125, https://doi.org/10.5194/WCD-1-111-2020, 2020.

Wirth, V. and Polster, C.: The Problem of Diagnosing Jet Waveguidability in the Presence of Large-Amplitude Eddies, J. Atmos. Sci., 78, 3137–3151, https://doi.org/10.1175/JAS-D-20-0292.1, 2021.

Woollings, T., Barriopedro, D., Methven, J., Son, S.-W., Martius, O., Harvey, B., Sillmann, J., Lupo, A. R., and Seneviratne, S.: Blocking and its response to climate change, Curr. Clim. Chang. Reports, 4, 287–300, https://doi.org/10.1007/s40641-018-0108-z, 2018.

---

## Author Comment (AC2)

Dear Reviewer 1,

Thank you for your helpful comments. We have provided our responses below in blue.

The authors present an analysis of atmospheric circulation persistence with a focus on summer heat waves and winter warm spells in Europe. Their method is based on atmospheric circulation analogues that are used to estimate the persistence of a atmospheric circulation configuration. The study is overall very interesting and the approach appears to be promising. In the current version there is a lack of clarity in the interpretation of results and the conclusions drawn from the analysis.

We thank the reviewer for their valuable time and helpful comments, and detail after each point the planned changes we would make, if we are invited to submit a revised manuscript.

Major comments:

In the current state, the manuscript lacks some clarity on the interpretation of the main findings. Some passages indicate, that the presented analysis that is based on a dynamical systems viewpoint contradicts main findings coming from the atmospheric blocking community (line 158-160, line 180-183, line 196-197). In the discussion the authors explain the methodological differences between their approach and the blocking approach (line 197-...). I would say that the main difference is the definition of "persistence". The authors use a definition of persistence that analyses the atmospheric circulation over a larger region. They can therefore quantify persistence for any day in the observations and compare the persistence of heat-wave days to other days. When analyzing the persistence of blocking there is a focus on a specific atmospheric circulation pattern and the persistence of blocking is not analyzed relative to the persistence of other flow patterns. It seems as if two approaches that are useful for different research questions are compared to each other which makes some of the interpretations of the paper misleading.

We thank the reviewer for highlighting that this section of the manuscript requires further clarification, and would edit the text so as to improve the clarity of this section. We do indeed use a different definition of persistence, and do agree in part that different definitions of persistence may have more or less applicability in various situations. We agree that the persistence of blocking not being analysed with respect to other regimes is true for blocking algorithms e.g. Davini et al. (2012), however, this is not true when blocking is considered as one of the four North Atlantic weather regimes (Vautard 1990). In the latter case, the persistence of blocking is directly compared to that of the other three regimes. In a revised manuscript we would specifically highlight the difference in definitions and clarify when we refer to specific atmospheric features and when to the persistence of the regional circulation. We want to emphasize that it was never our intention to mislead the reader, and we are committed to being transparent and providing a clear and accurate representation of our research findings.

I would suggest to describe the research question more precisely and frame the interpretation of the results and the discussion along this research question. Is the research question "Is the atmospheric circulation observed during heat waves more persistent than the average persistence?" or is it "Does longer persistence of a atmospheric circulation pattern that favors heat waves lead to more intense heat waves?" or is the research question "would it be more appropriate to describe the persistence of heat waves with a dynamical systems approach".

Thank you for your feedback. Based on your first suggestion, we plan to revise our manuscript to place a stronger emphasis on a clear research question. Specifically, we aim to explore whether the circulation patterns observed during heatwaves are related to circulation patterns that display above-average persistence. In doing so, we will also explain why we have chosen to use a dynamical systems approach, as suggested in your third comment. We want to clarify that our intention is not to focus on how atmospheric persistence may modulate the intensity of individual heatwaves. Rather, our primary objective is to investigate the connection between heatwave circulation patterns and above-average persistent circulation patterns.

If the main focus of the paper is a comparison with statements from the blocking literature I would also recommend to explain these statements in a bit more detail in the introduction to allow for more clarity in the discussion.

We will add more detail in the introduction according to the reviewer's suggestion. Whilst one aspect of this manuscript is to discuss different understandings of persistence, we also aim to explain how the dynamical systems approach works to readers from a more conventional atmospheric dynamics background. We wish to highlight the differences between these approaches and familiarize a broader audience with the technique we propose, so that it may be applied in research questions where it could be useful, and not remain on the mathematical or theoretical fringes of the climate science community.

One example of an interpretation that I would question:

As the authors explain, the most persistent atmospheric flow is zonal flow (see line 71-72). If summertime heat waves occur when the zonal flow is blocked, one would expect, that the atmospheric circulation during heat waves is not anomalously persistent. To me everything seems to be as expected so far. Therefore, I would write the sentence in line 157-158 differently. To me this seems to be a misunderstanding: It might be true, that more persistent blocking leads to more severe heat waves. And this can be true irrespective of whether zonal flow is generally more persistent than blocking. I therefore also disagree with line 180-182.

In a revised manuscript we would adjust lines 157-158 to reflect that the work presented in the manuscript to that point does not necessarily contrast the existing literature on blocked flow patterns. We would like to highlight that this manuscript is only considering the occurrence of heatwaves, rather than how individual heatwave severity may be modulated by persistence. Consequently, we would revise lines 180-

182 to reflect this refining of the argument. Furthermore, we would revise lines 42-44, as upon reading the review comments it has become apparent that we have inadvertently placed weight on the link between the persistence of blocks and the intensity of heatwaves, as opposed to only their occurrence.

Advection analysis and figure 4:

Looking at figure 4 it seems as if the advection that is analyzed here shows rather small scale features. Do these small scale features really represent the large scale flow that is shown in figure 1? Due to this (potential?) inconsistency I do not find the lines 160-166 convincing. In my view, more analysis would be needed to really interpret the role of warm air advection.

Based also on other review comments, we plan to significantly revise the section on temperature advection and would investigate an alternate metric for warm temperature advection. Specifically, we plan to use the advection of potential temperature by 10m winds, and have included a revised figure below, see Figure 1. Ultimately this does not change our qualitative conclusion that winter time warm spells appear to be associated with warm temperature advection in all regions except Russia, whilst there is a comparatively weak signal during summertime heatwaves. We approach Russian warm spells with caution because the potential temperature advection signal is small and noisy, and the significance stippling also appears to be noisy.

**Potential temperature Advection Anomaly Heatwave All Days**

[Figure]

Figure 1: Potential temperature advection (K/day) anomaly during warm spell/ heatwave days in (a,g) Scandinavia, (b, h) Germany, (c, i) Russia, (d, j) British Isles, (e, k) Iberia, (f, l) Mediterranean, during winter (a–f) and summer (g–l). Statistical significance is assessed as described in Section 2 of the manuscript and shown with grey stippling.

I would assume that the analysis is sensitive to the domain over which the atmospheric circulation is analyzed. A justification or an explanation of the choice of the Europe wide domain is lacking in section 2.1. There is one sensitivity test with a shifted domain which is great, but the interpretation of this sensitivity analysis is lacking in the main part of the manuscript. Furthermore, I think it would be more interesting to test the sensitivity to the size of the domain.

The chosen domain roughly encompasses the regions over which we define the heatwaves. This will be clarified in the revised text. In a revised manuscript we would also elaborate further on the domain sensitivity tests. We have run two additional sensitivity tests for a smaller (32.5-67.5N and -12.5-42.5E) and a larger domain (28.5-72.5N and -17.5-47.5E), which we include below in Figures 2 and 3 respectively. We see that the further sensitivity studies appear to show no qualitative effect for summer. For winter we do see a qualitative change with only Germany and Scandinavia, and Germany, Scandinavia and Russia for the small and large domains respectively, showing significant negative anomalies in theta. The sensitivity studies in the previous manuscript, in particular for heatwave/ warm spell duration, also highlighted that our wintertime results appear to be less robust than those for summer. While we do see some evidence of a link between atmospheric circulation persistence and wintertime warm spells, we interpret this with caution due to the apparent sensitivity of these results. This analysis does not change our qualitative conclusion that atmospheric persistence measured using SLP fields does not appear to be a necessary requirement for summertime heatwaves.

[Figure]

Figure 2: Box plots of θ [days⁻¹] anomalies for all heatwave or warm spell days in the considered regions. Theta has been calculated on a reduced domain 32.5-67.5N and -12.5-42.5E. Blue boxes indicate statistical significance, assessed as described in Section 2 of the manuscript.

[Figure]

Figure 3: Box plots of θ [days⁻¹ ] anomalies for all heatwave or warm spell days in the considered regions. Theta has been calculated on an expanded domain 28.5-72.5N and -17.5-47.5E. Blue boxes indicate statistical significance, assessed as described in Section 2 of the manuscript.

Minor comments:

L11: exact?
Exact is meant as a verb in this sentence.

L129: Is it relevant that the reader understands the method of Süveges? If yes, please explain it in more detail. If no, I'm not sure if you need to compare it to another method (Ferro and Segers) which is not expleained either?
While the details of the estimator of theta are not directly relevant to our results, we do agree that it is important to ensure the self-contained reproducibility of our article. Based on this, and other review comments, we therefore plan to significantly expand and elucidate the method section of this manuscript, specifically adding further detail on the method of Süveges, with the goal of making the methods section as self-contained as possible.

L130: Please write the package name here (I assume it is not "Robin")
What we meant was the package "by Robin (2020)". This will be fixed in a revised manuscript, with the package name (CDSK) included.

L184-185: This formulation could lead to a misinterpretation of the results (see my major comments above). The presented analysis does not study the link between more persistent anti-cyclonic configurations and the intensity and persistence of heat-waves.
Thank you for highlighting this and we agree with your second sentence, thus we will clarify the manuscript accordingly as this is not the message we wish to convey.

Figure A1: This sensitivity test is helpful and important. Would it also be possible to do a sensitivity test where the domain over which atmospheric circulation is analyzed is smaller, for example only the Mediterranean?
Following one of the previous comments by the Reviewer, we have run additional sensitivity tests on the domain size. Concerning shrinking the domain to a regional scale, this is in theory possible but may lead to noisy results due to the small number of gridboxes which would then be used to determine analogues. It would likely work for a higher resolution dataset, but would then require a separate analysis for each heatwave domain. We would leave this sort of analysis for future work.

References:

Davini, P., Cagnazzo, C., Gualdi, S., and Navarra, A.: Bidimensional diagnostics, variability, and trends of northern hemisphere blocking, Journal of Climate, 25, 6496–6509, https://doi.org/10.1175/JCLI-D-12-00032.1, 2012

R. Vautard, "Multiple Weather Regimes over the North Atlantic: Analysis of Precursors and Successors-Mon," Weather Reviews, Vol. 118, 1990, pp. 2056-2081. http://dx.doi.org/10.1175/1520-0493(1990)118

---

## Author Comment (AC3)

Dear Reviewer 2,
Thank you for your valuable suggestions and feedback on our manuscript. We provide our responses below in blue.

Review of paper

**The counter-intuitive link between European heatwaves and atmospheric persistence**

by Emma Holmberg et al.

submitted to *Earth System Dynamics*

Holmberg et al. examine the persistence of the European-wide SLP field during summer heat waves and winter warm spells using a previously developed measure of persistence that is based on wellestablished concepts in dynamical systems theory. The authors find that winter warm spells are associated with increased persistence (compared to climatological persistence values derived from all days) while for summer heat waves persistence anomalies are found to be moderately positive during heat waves in Scandinavia and even negative during heat waves in other regions. The authors identify in particular the absence of positive persistence anomalies for summer heat waves as "counter intuitive". The study is overall well written and the figures are, for the most part, clear. Also, I appreciate that authors' attempt to leverage concepts from dynamical systems theory to put the notion of "circulation persistence" onto more solid theoretical foundations. In that sense, I would like to encourage the authors to continue exploring their avenue of research.

However, despite the good intention and interesting overarching research goal of this study, there are a considerable number of issues that significantly compromise the value of this paper in its current form. I therefore suggest this study undergoes major revisions before it can be re-evaluated again. Some of my major comments below relate to the presentation of the methodology and I expect the authors will be able to address those rather easily. However, I also see more fundamental issues related to the authors choice of meteorological analysis techniques as well as to their interpretation of their findings.

We thank the Reviewer for their time and helpful suggestions, and have addressed below how we would edit the manuscript, if invited to provide a revised manuscript.

**Major comments**

A) The description of how "persistence" is defined and computed in Section 2.3 is not detailed enough to convey a physical intuition of what the authors exactly mean by persistence. I appreciate that the authors employ a rigorous and mathematical definition of persistence and I am aware that previous studies already introduced this definition of persistence. However, in some instances the authors challenge extremely well accepted reasoning within the atmospheric dynamics community (Blocking patterns during European heat waves are persistent flow features) and thus they should provide sufficient detail and explanations such that peers from this and other fields (i.e., scientists outside dynamical systems research community) can understand what exactly is meant by "persistence" without first consulting other papers. For instance, why exactly is the unit of theta days$^{-1}$? The frequency of what is it exactly? Moreover, is its inverse something like an e-folding time of some quantity? Or on lines 127–128: Why exactly is a generalized Pareto distribution fitted to the negative log of the analogue distances? Or on lines 129–131: What is the method of Süveges (2007)? And after all: How does a measure of clustering in extreme value theory (L69–70) inform about "persistence"? Note that I am not questioning your approach, based on the current Section 2.3 it just appears overly opaque to me.

Thank you for your comments, and detailing exactly which sections need further clarification. A revised manuscript would contain a significantly more detailed methods section, with the aim of providing a more self-contained explanation of the methodology. Specific answers to these questions, which would also be included in an extended form in a revised manuscript, are as follows:

The unit days$^{-1}$ is used because 1/theta is a measure of the amount of timesteps a given atmospheric state (or "map") remains qualitatively similar to itself, or in other words the expected number of timesteps for a cluster of similar maps. A cluster here is simply a continuous succession of similar maps, namely a succession of several days with a similar atmospheric state. An estimate of cluster length is therefore an estimate of persistence. We use daily data, and our theta is thus in units of days$^{-1}$, giving a persistence time in days. Theta is not derived as inverse e-folding time.

We fit a generalized pareto distribution (GPD) to the negative log of the distances because the mathematical theories we use are asymptotic theories which hold only in the limit of a sufficiently large data set, thus we do this to ensure that we have a sufficient sample size. Furthermore, the theory requires that the negative log of the distances converge to a certain family of distributions, of which the GPD is one, and convenient to work with. In a revised manuscript, we will clarify that the method of Süveges (2007) pertains to the algorithm used to estimate theta.

B) Related to A): On line 125 you state that you choose the 5% of days with the smallest distances as analogues to any day of interest. However, presumably not all of these analogues occurred during heat waves/warm spells. It is unclear to me how that fact affects the interpretation of your results. I read in your reply to the Community Comment by Dr. Alexandre Tuel that your methodology characterizes "local properties of the attractor". I don't fully understand the exact meaning of that statement, but it nevertheless appears plausible to me that a certain large-scale circulation (defined by its European-wide SLP pattern) can be persistent in some cases, but much less so in others. That is, it is quite plausible that heat waves occur in a unusually persistent manifestations of a given largescale circulation pattern, that not always exhibits this level of persistence.

We believe that there is a misunderstanding of what "local" means, which we will make every effort to clarify in the revised manuscript. When we estimate the persistence of a given atmospheric pattern, this is not the same as the persistence of its analogues. Say that we have a heatwave on e.g. the 5$^{th}$ July 2020, and one of its analogues happens to be the 20$^{th}$ August 2018, which was not a heatwave day. The persistence of the atmospheric pattern of the 5$^{th}$ July 2020 will not be the same as that of the 20$^{th}$ August 2018. Or better said: it could in theory be the same and one could build a synthetic dataset where this is the case, but in practice for "real" data it will not be. That is because the calculation of theta rests on the distribution in time of the closest 5% of analogues of each day, and the 5% of closest analogues of the 5$^{th}$ July 2020 will not be the same as the 5% of closest analogues of the 20$^{th}$ August 2018, even if the two are analogues of each other. Our method enables a heatwave to be attributed to an unusually persistent configuration compared to its analogues. The term 'local' in this context indicates that the measure of persistence is specific to a single map, which refers to a day in our dataset.

For instance, previous studies have shown that soil moisture anomalies associated with heat waves can have an "anchoring effect" on the associated anticyclonic circulation (e.g., Martius et al., 2021), which only affects the persistence of a given circulation pattern when a heat wave and the associated soil moisture anomaly occurs, but not in situations in which a similar circulation pattern occurs without a heat wave. How do the authors ensure that the persistence they quantify from sets of days including both heat wave and non-heat wave days is representative for heat wave days only? In case this comment simply results from a misunderstanding of your approach, I strongly recommend rewording the description of your approach.

As suggested by the Reviewer, we will better highlight the role of non-atmospheric drivers of heatwaves in our introduction and discussion sections in the revised

manuscript. Concerning the methodological part of the question, we refer to our above answer, namely the fact that our method assigns different persistence values to a given day versus its analogues. That is, it allows for the fact that one map in a set of similar maps can be more persistent than the others. To clarify this point, we show below (Figure 1) a box-and-whiskers plot of theta for a heatwave day over Germany (the cross) and for its 5% of closest analogues. As can be seen, this specific heatwave day has an anomalously low theta (high persistence) compared to its analogues. Other heatwave days may instead have an average or unusually low persistence relative to their analogues. We will endeavor to clarify this aspect of our methodology in the manuscript in order to reduce future misunderstandings. In this respect, we trust that the expanded explanation of the methodology we will introduce in response to the Reviewer's comment A will be of help.

[Figure]

Figure 1: Box plot of $\theta$ [days$^{-1}$] anomalies calculated for all analogues of September 12$^{\text{th}}$, 2002. The red cross denotes the theta anomaly for September 12$^{\text{th}}$, 2002.

C) A key finding of this study is that during summer heat waves in many parts of Europe, the circulation is not anomalously persistent and the authors claim that this is at odds to the well accepted notion that European summer heat waves (at least in Central and Northern Europe) often occur in association with "persistent blocking" (e.g., lines 157–158, or 180– 182). However, a contradiction is not particularly apparent to me here. It is well possible that zonal flows (which often lead to wet and thus not anomalously hot summer conditions across central Europe) are more persistent by the author's metric than blocked flows. Nevertheless it may still be true that severe summer heat waves occur preferentially during the most persistent blocking episodes (e.g., lines 42–44 and references cited there). I therefore suggest that the authors more clearly frame their research question and better explain what exactly is counter-intuitive about their findings.

We agree with the Reviewer's point. In a revised manuscript we would clarify that our approach gives a single number quantifying the atmospheric persistence of a specific map. As mentioned in our reply to comment B, when considering a given heatwave associated with a blocked-like configuration, we are not providing the persistence of all blocked configurations, we are providing the persistence of that one specific blocked configuration associated with the heatwave we have identified. Thus, the Reviewer is correct that our results do not contradict the notion of a link between blocking persistence and heatwave persistence. We agree that we need to make the distinction between looking at the persistence of individual heatwaves and looking at the average persistence of blocked versus zonal flows. In a revised manuscript, we would clarify this point.

D) I believe Fig. 4 and the quantity B does not help to understand the contribution of largescale warm air advection to heat waves/warm spells, for two reasons: Firstly, it identifies essentially just regions of gradients in the surface elevation. The vector **B** is computed from SLP (which everywhere is indicating the pressure *at sea-level*), while two-meter temperature is dependent on the height of the terrain and thus large horizontal (i.e., terrain following) gradients of two-meter temperature are first and foremost indicating steep terrain. Second, the implicit assumption motivating the definition of **B** is presumably that by the geostrophic balance the gradient of SLP is indicative of the surface winds. However, especially near the surface the geostrophic balance is often quite substantially disturbed by surface drag and the near-surface flow is not as close to geostrophic as the upper-level flow. Therefore, I believe the authors should repeat their analysis and either consider horizontal temperature advection (i.e., $-v\nabla T$) on a near-surface pressure level explicitly or, alternatively, consider the advection of potential temperature by the 10m winds.

We thank the Reviewer for this helpful suggestion and would now consider the advection of potential temperature by the 10m winds. Ultimately this does not change our qualitative

conclusion that winter time warm spells appear to be associated with warm temperature advection in all regions except Russia, whilst there is a comparatively weak signal during summertime heatwaves. We treat Russian warm spells with caution due to the small, noisy potential temperature advection signal, and because the significance stippling also appears noisier. We have included the revised figure below, see Figure 2.

[Figure]

Figure 2: Potential temperature advection (K/day) anomaly during warm spell/ heatwave days in (a,g) Scandinavia, (b, h) Germany, (c, i) Russia, (d, j) British Isles, (e, k) Iberia, (f, l) Mediterranean, during winter (a–f) and summer (g–l). Statistical significance is assessed as described in Section 2 of the manuscript and shown with grey stippling.

E) The choice of SLP as variable to quantify the persistence of the large-scale circulation appears peculiar and suboptimal to me, in particular in the context of this study, which often makes reference to atmospheric blocking. Firstly, it is well known that during heat waves so-called heat lows can develop and thus the evolution (and hence potentially also the persistence) of the mid- to upper-tropospheric flow may differ substantially from the evolution of the near-surface flow. Secondly, atmospheric blocks often feature rapidly amplifying disturbances on their upstream side, which feed into the blocks and thereby enhance their persistence (Shutts, 1983; Pfahl et al., 2015). In the SLP field these upstream disturbances manifest themselves as extratropical cyclones (i.e., with strong signals in the SLP field). Thus, one does not a priori expect a "persistent" European-wide SLP pattern during persistent atmospheric blocks. Thirdly, it is anyways quite a stretch to argue about the persistence of blocking flows based on SLP, as blocking is predominantly a manifestation of the upper-level flow (Kautz et al., 2022; Woollings et al., 2018). The authors could easily exclude these issues, e.g., by performing their persistence quantification with (de-trended) Z500 fields or, even better, isentropic PV fields rather than SLP fields, as these quantities much more directly inform about the large-scale circulation.

Thank you for this valuable comment. Indeed, your arguments on the dynamical relevance of SLP have highlighted to us that this aspect of the manuscript requires further, detailed analysis. Isentropic PV is a relatively noisy field, yet may be worth attempting to study using our persistence indicator, perhaps after some spatial smoothing. Our original motivation for not using Z500 was that trends in the variable one computes the analogues on may lead to bias results (since one risks finding analogues concentrated in the years around the reference day, due to the trend). Detrending a variable prior to analysis may itself be problematic, since the trend is typically subtracted at each gridpoint, leading to complex distortions of the system's phase-space and analogue distances.

We will consider testing isentropic PV, and alternatively perform additional tests on how well the analogues from de-trended and non de-trended Z500 fields satisfy the assumptions underlying our computation of theta. In reasoning on your comment, an additional thought we had was to extend our computation of theta to consider two variables at the same time, which in this case could be both SLP and Z500 (subject to our additional tests convincing us that Z500 can be used to compute theta). This would reflect both the surface and upper level structures, while retaining the simplicity of having a single persistence value for each atmospheric map. As there is a significant amount of work required, we do not have any detailed results to present on these points at the time of writing this response.

**Minor comments**

1. L5: I do not think your study indeed "reconciles" the dynamical systems and traditional views on persistence. It rather demonstrates that your approach leads to results that diverge from traditional views on persistence.

   We will reword this accordingly, however, in a revised manuscript we would still endeavor to contribute a piece of literature which helps clarify this approach in terms of the traditional views on persistence.

2. L16–17: The authors should clarify what exactly they mean with the word "extreme". Do they mean "rare", e.g., as on line 20 where the authors refer to "Uncommonly high temperatures", i.e., "extreme" in the sense of a large return period? Or do the authors perhaps mean "extreme" in the sense of "hot enough to cause impact"? If the authors mean "extreme" in the sense of "rare" then the sentence on lines 16–17 is contradictory. If extremely hot summers are defined via their rareness then, by definition, they cannot become more likely. Rather, the temperature corresponding to a certain return period will increase, i.e., events of equal extremeness will become more intense.

   Here our definition of extreme is based on a percentile threshold calculated for a given time period. We will clarify that our intention was to highlight that if one defines the threshold for hot extremes based on the value of a fixed percentile of the historical distribution, in the future one will have more hot extremes. As the Reviewer points out, it is natural to assume that recomputing the value of the same fixed percentile for the new temperature distribution would result in events of equal extremeness becoming more intense.

3. L43–44: The authors could explain here in more detail how the "current understanding of the link between blocks and summertime heat waves" works physically. How exactly are long-lasting blocks supposed to increase the odds of intense heat waves? Perhaps mention here the adiabatic warming in subsiding air, the clear-sky conditions, drying of soils (due to the persistent local weather within the block), etc.

   Thank you for your suggestions, in a revised manuscript we will add further detail to a physical explanation of the link between blocking and heatwaves, incorporating your suggestions. As mentioned in other review comments, we are planning to reword our comments about heatwave intensity, as we believe we have inadvertently placed emphasis on heatwave intensity as opposed to occurrence.

4. L57–60: The authors refer to a series of papers that developed the Quasi-resonant amplification (QRA) hypothesis. The theoretical basis of this hypothesis has been severely challenged by a number of recent publications (e.g., Wirth and Polster, 2021; Wirth, 2020). So far the authors of the QRA papers have not been able to

respond to or address this critique. Therefore, if the authors of this paper decide to refer to the QRA papers they should at least indicate that the validity of the reasoning in these QRA papers is severely contested.

Thank you for highlighting this, we will ensure that this point is made in the paper, with the appropriate references cited. Alternatively, we may consider removing these references entirely, as they are not central to any of the conclusions we draw in our study.

5. L69–71: Here you introduce the concept of persistence from a dynamical systems perspective. I'd find it very helpful if you could add here what "persistence" means in that context, i.e., if you could convey as physical intuition for what that term means in the dynamical systems context and, perhaps also to what extent it is comparable to the persistence of an individual blocking anticyclone identified with the feature-based perspective.

We will endeavor to provide more physical intuition; however, this is a mathematical definition at its core so the explanation will likely always remain somewhat technical. One example which may provide a more physical understanding of persistence in the context of dynamical systems is the following: consider persistence to be a number quantifying how many days in a row a given map is expected to remain similar to itself. We interpret this as how many days in a row the atmosphere remains, in terms of the underlying mathematical system, similar to itself. This could be akin to how many days in a row do we have a similar structure of high and low pressure systems driving the atmospheric circulation pattern. We also plan to either add a graphical schematic or move the case study example to a location earlier on in the manuscript.

6. L89: Is it really 1978–2018 or 1979–2018? Did you use the ERA5 back extension? If so, why only back to 1978?

This typo will be corrected from 1978 to 1979.

7. L91: Maybe 12 UTC instead of "noon"?

We will edit the manuscript accordingly.

8. L93: Something is wrong with the longitudes of your domain.

Thank you for spotting this, we will edit the manuscript accordingly.

9. L94: Is "daily climatology" the same as "calendar day climatology"? If so, consider using the latter term.

We are using a calendar day climatology and will clarify in a revised manuscript.

10. L97: What exactly is "the local distribution"? Do you take any forced trend into account when identifying your events of interest? If not, why is that choice justified in this case?

The local distribution refers to the distribution for a given grid box. We do not take forced trends in temperature into account. Whilst there is certainly a trend, our analysis should not be sensitive to how the heatwaves are distributed in time, since we look for atmospheric analogues of each heatwave over the full time period considered. What could cause issues is if the variable we are computing the analogues on has a strong trend. This is why we originally select SLP as opposed to temperature itself, or Z500.

11. L109–114 and Figs 1, 2, 4 and 5: Given that several of the authors are excellent statisticians I am surprised that you did not apply the False-Discovery-Rate test of Benjamini and Hochberg, (1995) as detailed in Wilks, (2016). Why did you not follow the procedure suggested by Wilks, (2016) even though you clearly perform multiple test simultaneously?

The Reviewer is correct in pointing this out. We apologise for missing this important step and will include this in further analysis.

12. 12. L123: Did you include any latitude weighting when computing the Euclidean distances?

No, this was neglected due to the limited latitudinal extent of the domain. We have however verified that applying this weighting does not affect our results, and indeed the correlation between the thetas computed with and without the weighting is 0.9. We have added latitudinal weighting to the updated figures in our responses, and we will do the same for the figures in the revised manuscript.

13. Figures 1, 2, 4 and 5: The stippling indicating significance is difficult to see. Can you make it better visible? Also, please add the boxes indicating the regions to Figs. 4 and 5, so that the reader always knows what the areas of interest are.

We will amend these figures as suggested.

14. Figure 1: I'd be curious to see an analogous figure to Fig. 1 but displaying the Z500 anomaly variance. This would show where (in space) the circulation is (or is not) variable (i.e., not persistent) during your events.

We will include this figure in the supplementary material, and have included the figure below, see Figure 3. This should however be interpreted with care as it provides a local as opposed to domain-wide view of persistence.

[Figure]

Figure 3: Z500($m^2/s^2$) anomaly variance during warm spell/ heatwave days in (a,g) Scandinavia, (b, h) Germany, (c, i) Russia, (d, j) British Isles, (e, k) Iberia, (f, l) Mediterranean, during winter (a–f) and summer (g–l).

15. L154–155: I wonder to what extent your European wide circulation persistence compares with the persistence of the "local" weather during your events (e.g, measured by the average/median duration of hot spells during your events or by the T2m variance during your events). If you could indeed bring together these two views on persistence then I think the word "reconciling" in your abstract would be justified.

This is a good suggestion but we believe it hides considerable complexity. Indeed, it is similar to the question of how the persistence of e.g. a block is related to the persistence of a coinciding heatwave, something which is currently somewhat of a knowledge gap. Indeed, we were unable to find studies explicitly relating duration of blocks with duration of the associated heatwaves (while many studies implicitly look at this by imposing a minimum duration to both detect blocking and heatwaves). We intend to investigate the correlation between persistence and the duration of a heatwave, although we do not anticipate a definitive outcome. We will assess whether to incorporate this analysis into the revised paper once we have examined some initial findings.

16. Figure 3: The axis labels are difficult to read. Please enlarge them.
We will amend these.

17. L165–166: The orographic signatures in Fig. 4 are just a consequence of the definition of B. I do not think we learn anything from these signals about the importance of downslope winds for hot extremes.

We are planning to significantly revise this section following from other review comments as well, and will consider the advection of potential temperature by 10m winds, as discussed in response to comment D, where we include an updated figure.

18. L171–172: I do not understand this sentence. How exactly is "Russia an exception"?

We find that Russia appears to have a less clear case, however we will revise the manuscript to word this less strongly.

19. L201–203: The authors refer to "other atmospheric structures". What exactly is meant here?

This could be for example small low pressure system or other features in the atmospheric map which may be distinct from the most 'dominant' large scale structure.

20. L225–228: I find this statement on the importance of persistence for local (i.e., diabatic) and remote drivers (i.e., advection of air from climatologically warmer

regions) of heat waves misleading and suggest to reword this statement. Firstly, in particular for heat waves driven by radiative effects the persistence of (local?) weather (i.e., dry conditions, clear skies) is well known to be important because the amplification of land-atmosphere feedbacks, i.e., soil drying and associated increase in sensible at the expense of latent heat fluxes, take time (e.g., Miralles et al., 2019).

We will reword this to clarify that this is a key point we wish to make, namely differentiating between local conditions and domain-scale persistence, which corresponds to the large-scale structure of the atmosphere.

21. Furthermore it is unclear why warm air advection would require persistence. Even if the circulation changes over time, long-range transport of air from climatologically warm regions may occur. Moreover, at least at synoptic scales, near-surface warm air advection tends to be largest ahead of cold fronts which I would not characterize as persistent weather situations. I understand that the authors here refer to the persistence of "the large-scale atmospheric configuration", but as it is written now the statement (i.e., the authors interpretation of their results) is at odds to existing literature and, in my opinion, not sufficiently well substantiated by their results. The issue might be resolved if a distinction could be made between the persistence in the large-scale circulation pattern and the persistence of local weather.

We will endeavor to revise the manuscript to highlight where we make a distinction between local and large-scale persistence. In our interpretation we actually make a point that the UK does not show a persistent signal during wintertime warm spells, despite showing a signal for temperature advection. We hypothesise that a persistent signal may be expected for regions where the advection of warm air is expected to come from a more specific direction, as this would likely have a stronger requirement for the large-scale circulation pattern to remain similar. The Reviewer is accurate in pointing out that our manuscript does not utilize a Lagrangian-style tracking algorithm, and our examination of advection is founded on composites. We will elaborate on this aspect further in the revised discussion section.

22. L237–239: The "visual appraisal" does not suffice to claim that "a blocking algorithm would detect a blocked flow persisting for several days". Firstly, the large family of blocking identification algorithms derived from the original approach of Tibaldi and Molteni (1990) usually consider gradient reversals in the absolute (not anomaly) Z500 fields, which is not apparent from your Fig. 6. Secondly, the "visual appraisal" in Fig. 6 does not convince me that, e.g., the 5-day persistence criterion for negative upper-level PV anomalies in the Schwierz et al. (2004) algorithm would be fulfilled. I suggest to tone down the interpretation of Fig. 6 in this regard.

We will tone down the interpretation accordingly, however we would like to note that the first quote of your comment has dropped the word "suggests", which in itself tones the statement down somewhat.

**References**

Benjamini, Y. and Hochberg, Y.: Controlling the False Discovery Rate: A practical and powerful approach to multiple testing, J. R. Stat. Soc., 57, 289–300, https://doi.org/10.1111/j.25176161.1995.tb02031.x, 1995.

Kautz, L.-A., Martius, O., Pfahl, S., Pinto, J. G., Ramos, A. M., Sousa, P. M., and Woollings, T.: Atmospheric blocking and weather extremes over the Euro-Atlantic sector – a review, Weather Clim. Dyn., 3, 305–336, https://doi.org/10.5194/WCD-3-305-2022, 2022.

Martius, O., Wehrli, K., and Rohrer, M.: Local and Remote Atmospheric Responses to Soil Moisture Anomalies in Australia, J. Clim., 34, 9115–9131, https://doi.org/10.1175/JCLI-D-210130.1, 2021.

Masato, G., Hoskins, B. J., and Woollings, T.: Winter and summer Northern Hemisphere blocking in CMIP5 models, J. Clim., 26, 7044–7059, https://doi.org/10.1175/JCLI-D-12-00466.1, 2013.

Miralles, D. G., Gentine, P., Seneviratne, S. I., and Teuling, A. J.: Land–atmospheric feedbacks during droughts and heatwaves: state of the science and current challenges, Ann. N. Y. Acad.

Sci., 1436, 19–35, https://doi.org/10.1111/NYAS.13912, 2019.

Pfahl, S., Schwierz, C., Croci-Maspoli, M., Grams, C. M., and Wernli, H.: Importance of latent heat release in ascending air streams for atmospheric blocking, Nat. Geosci., 8, 610–614, https://doi.org/10.1038/ngeo2487, 2015.

Scherrer, S. C., Croci-Maspoli, M., Schwierz, C., and Appenzeller, C.: Two-dimensional indices of atmospheric blocking and their statistical relationship with winter climate patterns in the EuroAtlantic region, Int. J. Climatol., 26, 233–249, https://doi.org/10.1002/joc.1250, 2006. Schwierz, C., Croci-Maspoli, M., and Davies, H. C.: Perspicacious indicators of atmospheric blocking, Geophys. Res. Lett., 31, L06125, https://doi.org/10.1029/2003GL019341, 2004.

Shutts, G. J.: The propagation of eddies in diffluent jetstreams: Eddy vorticity forcing of "blocking" flow fields, Q. J. R. Meteorol. Soc., 109, 737–761, https://doi.org/10.1002/qj.49710946204, 1983.

Tibaldi, S. and Molteni, F.: On the operational predictability of blocking, Tellus A, 42, 343–365, https://doi.org/10.1034/J.1600-0870.1990.T01-2-00003.X, 1990.

Wilks, D. S.: "The stippling shows statistically significant grid points": How research results are routinely overstated and overinterpreted, and what to do about it, Bull. Am. Meteorol. Soc., 97, 2263–2273, https://doi.org/10.1175/BAMS-D-15-00267.1, 2016.

Wirth, V.: Waveguidability of idealized midlatitude jets and the limitations of ray tracing theory, Weather Clim. Dyn., 1, 111–125, https://doi.org/10.5194/WCD-1-111-2020, 2020.

Wirth, V. and Polster, C.: The Problem of Diagnosing Jet Waveguidability in the Presence of Large-Amplitude Eddies, J. Atmos. Sci., 78, 3137–3151, https://doi.org/10.1175/JAS-D-200292.1, 2021.

Woollings, T., Barriopedro, D., Methven, J., Son, S.-W., Martius, O., Harvey, B., Sillmann, J., Lupo, A. R., and Seneviratne, S.: Blocking and its response to climate change, Curr. Clim. Chang. Reports, 4, 287–300, https://doi.org/10.1007/s40641-018-0108-z, 2018.

Süveges, M.: Likelihood estimation of the extremal index, Extremes, 10, 41–55, https://doi.org/10.1007/s10687-007-0034-2, 2007

---

## Author Comment (AC4)

Dear Reviewer 3,

Thank you for your helpful comments. We provide our responses below in blue.

This paper examines the links between heat extremes and a metric of atmospheric persistence from dynamical systems theory. Using this metric, it is argued that there is no evidence of a link between heat extremes and anomalously persistent circulation patterns. Some assessment of the role of different terms in driving temperature anomalies is also provided and it is argued that there is an important role for zonal temperature advection in driving wintertime heat extremes, while in the summertime temperature advection does not play an important role. I can see that it's useful to quantify atmospheric persistence by this dynamical systems measure, although I have some questions about the methodology. My primary concern is about the interpretation and the conclusions when using this measure. I think there are two potential interpretations. One is that persistent circulation patterns are not closely linked to heat extremes, which is the interpretation the authors have provided. The other is that this dynamical systems metric is actually not a very good measure of persistent circulation patterns. I'm not totally convinced by the authors argument that this second interpretation can be ruled out, so my more major suggestion is that either a clearer demonstration of this being a preferred metric for atmospheric circulation persistence should be provided, or the discussion should be changed to be a bit more balanced on whether this metric is accurately measuring persistence of atmospheric circulation anomalies.

We thank the Reviewer for their valuable time and helpful comments, and detail below the edits we would make, if we are invited to provide a revised manuscript.

General comments:

As mentioned above, my primary concern is whether the interpretation that "atmospheric persistence is not a necessary requirement for summertime heatwaves" (l9) is actually correct, or whether an alternative interpretation is that this dynamical systems theory metric of atmospheric persistence does not adequately capture persistent atmospheric circulation anomalies. I feel like the demonstration in Figure 6 supports that this metric might not be very good at capturing persistent atmospheric circulation anomalies. It is clear that Figure 6b and d demonstrate a persistent block, as indeed the authors state. But it seems like it could be argued that this dynamical systems metric is, therefore, just not a very good metric for persistent atmospheric flow anomalies. I'd question whether it can really be used to argue that "highly persistent configurations are not a necessary criterion for European summertime heatwaves to occur" (l242) when clearly there is a case in Figure 6 that does have a persistant atmospheric flow configuration but is considered not persistent by this metric. I'm left a bit confused about what argument is exactly being made. If I read the paper in a cursory manner, I might think that the conclusion is that you don't need persistent atmospheric circulation patterns to produce heatwaves, but if readers think more about it and pay attention to Figure 6, it seems the conclusion should actually be that this particular dynamical systems metric of persistence doesn't really capture

persistence in atmospheric flow patterns of relevance to heatwaves.  I suggest the authors either need to make a clearer case for why this metric is preferable to those based on persistent flow regimes or alter the wording in places to make clear that actually this dynamical systems metric of persistence doesn't do that good a job of picking out persistant blocking highs or other flow regimes that are relevant for heatwaves.   I think either conclusions is worthy of publication, but I'm confused about which one is being drawn.  Another example of a confusing conclusion is lines 247-249 where it's stated that "our results appear to contrast the conventional view of heatwaves being associated with very persistent blocked configurations" when above, in reference to Fig 6, it's stated that a blocking algorithm would detect a block in the low persistence case for several days (l238).  It seems, then, that this metric is not a very good metric of blocking, so how can it be used to contrast the conventional view of very persistent blocked configurations being connected to heatwaves?

Thank you for this detailed comment. We would like to argue that the choice of a 'good' metric contains a certain amount of perspective. We believe that this is a very important discussion to be had, and possibly the crux of bringing the mathematical and dynamics perspectives of persistence together so as to be reconciled and mutually understood. We argue that a 'good' metric requires a sound, non-subjective definition, and would significantly revise our manuscript so as to clarify this. As part of this discussion, we would argue that the reviewer's evaluation that Fig. 6 "does have a persistent atmospheric configuration" is a subjective statement, while our chosen distance metric allows to make a quantitative statement. In the study we very carefully state that in Fig. 6b and d a blocking algorithm would detect a persistent block, not that there **is** a persistent block. Our whole argument is that the block in those figures should **not** be considered as persistent, at least from an SLP persistence perspective. We are not arguing that blocking algorithms are wrong, but rather that they do not necessarily match an objective and quantitative definition of continental-scale atmospheric persistence. It seems to us that the two conceptualisations of persistence are useful for different research questions, which we would clarify in a revised discussion section.

(2) One aspect of the methodology that I wondered about was is it easier to have higher persistence when there are less anomalies overall in the spatial field.  I'm imagining that if you have a really large amplitude anomaly but that moves slightly, the spatial Euclidian distance between days that have a relatively small movement may end up being a lot larger than the Euclidian distance between days where there's much less going on in the spatial field.  If so, then maybe this isn't a particularly good measure of the persistence of relevance to heat extremes.  Persistence of nothing much going on over the region wouldn't be very meaningful, whereas having a persistant blocking high that stays around for a long time in the region, even if it moves slightly, would likely be more impactful for heat extremes.  Or perhaps there's something in the methodology that prevents this from happening. I recommendt this be discussed or assessed.

This method is indeed dependent on the distance metric chosen. However, note that a given day will not necessarily be more persistent simply because it has closer analogues – as in your example of a day with a large atmospheric feature versus a

day with very small anomalies over the whole domain. The estimate of theta is based on the 5% of closest analogues of each day, and the "closeness" of these analogues to the day itself does not determine the persistence of that day. Rather, it is the distribution of the analogues in time that is important. We will make sure to clarify this very important point in the revised manuscript.

Minor comments by line number:

l94: I don't think "345-45W" is correct.  Maybe 15W-45E?

Thank you, this will be corrected.

Figure 1 caption: I think this could be a bit clearer if it stated "warm spells during winter (top) and heatwaves during summer (bottom)".  (It took me a while to notice the winter and summer in the titles).

Thank you, this will be corrected.

l160: In the discussion of the role of advection starting here and referring to Figure 4, "the role of warm air advection toward the region of interest during warm spells" is not entirely obvious to me in a causal sense.  Couldn't there also potentially be a role for the temperature anomaly itself being produced by some other cause actually leading to temperature advection anomalies.  I feel like the reds next to blues in this figure may be indicative of that i.e., you get some warm anomaly set up and if the zonal flow is westerly then you end up with cold advection to the west of the warm anomaly and warm advection to the east.  Even if that's not the case, the dominant role of warm advection isn't totally clear to me from the figure since it's very noisy.  Is the intention that readers should be paying attention to the larger spatial scales, as oposed to the small scale noise?  If so, maybe some filtering to retain only larger spatial scales could be performed?

Based also on other review comments, we are planning to significantly revise this figure and use an alternate metric, in order to clarify the arguments around warm temperature advection. Specifically, we plan to use the advection of potential temperature by 10m winds, and have included a revised figure below, see Figure 1. Ultimately this does not change our qualitative conclusion that winter time warm spells appear to be associated with warm temperature advection in all regions except Russia, whilst there is a comparatively weak signal during summertime heatwaves. We treat Russian warm spells with caution due to the small, noisy potential temperature advection signal, and because the significance stippling also appears noisier. We will also discuss in the revised text the possibility raised by the Reviewer that the temperature anomaly itself may be produced by some other cause actually leading to temperature advection anomalies.

[Figure]

Figure 1: Potential temperature advection (K/day) anomaly during warm spell/ heatwave days in (a,g) Scandinavia, (b, h) Germany, (c, i) Russia, (d, j) British Isles, (e, k) Iberia, (f, l) Mediterranean, during winter (a–f) and summer (g–l). Statistical significance is assessed as described in Section 2 of the manuscript and shown with grey stippling.

l165-166: You mention the potential role of advection over topography for engendering large temperature anomalies here. But couldn't this also potentially be an artefact of using SLP? SLP will involve some extrapolation below the surface making an assumption about the lapse rate, I think. So SLP will be an approximation over topography and will be affected by the temperature at the surface, so is it possible that this could be playing a role here?

In light of both this comment and others, as mentioned above, we are planning on revising to use the potential temperature advection by 10m winds as a metric instead, which should not be subject to this same possible issue.

---

## Author Comment (AC5)

Dear Alex,

Thank you for your follow up comment. We provide our responses below in blue.

Thank you Emma for these detailed comments which helped me better understand your work. It is nice to see that the results remain the same when based on summer data only. Out of curiosity, did you use the raw SLP data or did you remove an annual cycle?

We used raw SLP data so as to ensure that the analogues selected by the algorithm truly correspond to the most similar circulation patterns. Using anomalies could lead to analogues in different months displaying similar anomalies relative to the month they occur in, but potentially very different actual circulation patterns. As discussed in our previous answer, this leads to the bulk of the analogues for days in a given season falling in that same season.

You wrote "In other words, we are not taking an average of the persistence of configurations similar to those of the heatwave and assigning that average persistence to the heatwave itself. This is a very important point that we will make sure to clarify in the revised methods section of our manuscript." I agree that this is important and might cause a lot of confusion, because clearly what "persistence" means for the average atmospheric scientist is very different from your definition. In particular, persistent patterns in the "traditional sense" may not be classified as very persistent with your dynamical systems approach. This also relates to my earlier comment about heatwaves and persistence, and this is why you should be careful when writing that heatwaves in Western Europe are not associated with much atmospheric persistence (e.g., "We thus argue that atmospheric persistence is not a necessary requirement for summertime heatwaves") What you argue, if I understood well, is that the states of the circulation that tend to occur during heatwaves are not persistent from the attractor perspective. However, this does not imply that during a heatwave, the circulation does not tend to remain stuck in a small part of the attractor (and Hoffmann et al's metric would capture this case). Is that correct? If yes, the point needs to be made clearly in the manuscript.

Thank you for highlighting this point. Based also on the review comments we would, if invited to submit a revised manuscript, endeavor to clarify our interpretations, and word them carefully so as not to confuse the audience, in particular taking care to highlight when we consider different perspectives on persistence. As far as we understand, Hoffmann et. Al.'s metric is based only on the similarity between 10 consecutive maps, rather than drawing on analogues from a longer time series. The algorithm we use to compute theta (the extremal index) is computed using the time between clusters and lengths of clusters of similar maps (i.e. those maps within the 5% threshold to class a map as an analogue of our chosen day). Thus, our computation has elements which are quite similar to Hoffman et. Al.'s metric, with the key difference that we calculate our statistics based on the properties of a much larger data set. In this sense, our method could be viewed as a generalisation of Hoffmann et. Al.'s metric, with the added advantage that our metric can be applied forwards in time i.e. we can quantify for how many days we would expect today's circulation pattern to remain similar to itself. Our persistence metric does indeed provide an estimate of how many time steps a trajectory is expected to stay within an epsilon-sphere of it's starting point in phase space. Thus, it is directly a measure of how long a trajectory 'sticks' in a given phase-space region. Consequently, higher persistence indeed means that the circulation remains stuck in a small part of phase space, whilst

lower persistence means that it doesn't. We do note that here we can only make comments on the entire structure of the circulation pattern in a given geographical domain, and not about specific circulation features within the domain.

For future research, it would be nice to use Z500 (with the seasonality taken out) instead of SLP. For summer heat extremes, SLP is not quite as relevant, especially in the lower half of your domain. Z500 would be a better proxy for atmospheric circulation. Heatwaves may indeed be associated with heat lows while Z500 exhibits a pronounced ridge. The SLP analogues might thus not be very physically meaningful since low SLP at other times could be associated with cyclonic activity.

Thank you for this suggestion, it is a very important point and we refer you to Comment E and our response in the discussion with Reviewer 2.

One last point (motivated by personal curiosity, no need to change the manuscript): analogues are selected based on some low percentile of L2-distance across the whole trajectory. So, the number of analogues is the same for all time steps, but the average distance between the analogues and the target value is not fixed. In particular for some rare states, it could end up being high. Or is the threshold reasonably constant with time?

You are correct in stating that some days will have analogues which have a smaller L2 distance than others. We fit the generalized pareto distribution to the negative log of distance, as detailed in the methods section, to ensure that what we call "analogues" satisfy the mathematical assumptions our estimation of theta rests on, and that our overall statistics are not affected by poor analogues. One could hypothesise that this may eventually become an issue with a changing climate, however, based on the above we do not believe that trends in SLP are strong enough to affect our analysis.

---

## Author Response (AR1)

Dear Reviewer 1,

Thank you for your helpful comments. We have provided our responses below in blue.

The authors present an analysis of atmospheric circulation persistence with a focus on summer heat waves and winter warm spells in Europe. Their method is based on atmospheric circulation analogues that are used to estimate the persistence of a atmospheric circulation configuration. The study is overall very interesting and the approach appears to be promising. In the current version there is a lack of clarity in the interpretation of results and the conclusions drawn from the analysis.

We thank the reviewer for their valuable time and helpful comments, and detail after each point the changes we have made. We note that Fig. 6 has been updated as an error in the scaling factor for the plot was found during the process of preparing the revised manuscript, this has not changed the qualitative pattern or conclusions associated with that figure.

Major comments:

In the current state, the manuscript lacks some clarity on the interpretation of the main findings. Some passages indicate, that the presented analysis that is based on a dynamical systems viewpoint contradicts main findings coming from the atmospheric blocking community (line 158-160, line 180-183, line 196-197). In the discussion the authors explain the methodological differences between their approach and the blocking approach (line 197-...). I would say that the main difference is the definition of "persistence". The authors use a definition of persistence that analyses the atmospheric circulation over a larger region. They can therefore quantify persistence for any day in the observations and compare the persistence of heat-wave days to other days. When analyzing the persistence of blocking there is a focus on a specific atmospheric circulation pattern and the persistence of blocking is not analyzed relative to the persistence of other flow patterns. It seems as if two approaches that are useful for different research questions are compared to each other which makes some of the interpretations of the paper misleading.

We thank the reviewer for highlighting that this section of the manuscript requires further clarification, and have edited the text aiming to improve the clarity of this section. In response to other review comments we have expanded the analysis to include persistence computed using Z500 (Fig. 3), in addition to the previously included persistence computed using SLP. In light of this additional analysis we have now refined our line of argument, noting that the results are partially consistent with previous literature from the dynamical systems field, which highlighted zonal flow as being persistent at the surface level. Furthermore, the qualitative findings of our analysis of persistence calculated using Z500 are in line with the conventional view of heatwaves being linked to persistent configurations. We note that despite the qualitative agreement during summer, the magnitude of the theta anomalies are quite

small, thus our argument that anomalous atmospheric persistence is not a necessary requirement for heatwaves is not qualitatively altered.

We do indeed use a different definition of persistence, and do agree in part that different definitions of persistence may have more or less applicability in various situations. We agree that the persistence of blocking not being analysed with respect to other regimes is true for blocking algorithms e.g. Davini et al. (2012), however, this is not true when blocking is considered as one of the four North Atlantic weather regimes (Vautard 1990). In the latter case, the persistence of blocking is directly compared to that of the other three regimes. In our revised manuscript we specifically highlight the difference in definitions and clarify when we refer to specific atmospheric features and when to the persistence of the regional circulation. We want to emphasize that it was never our intention to mislead the reader, and we are committed to being transparent and providing a clear and accurate representation of our research findings.

I would suggest to describe the research question more precisely and frame the interpretation of the results and the discussion along this research question. Is the research question "Is the atmospheric circulation observed during heat waves more persistent than the average persistence?" or is it "Does longer persistence of a atmospheric circulation pattern that favors heat waves lead to more intense heat waves?" or is the research question "would it be more appropriate to describe the persistence of heat waves with a dynamical systems approach".

Thank you for your feedback. Based on your first suggestion, we have revised our manuscript to place a stronger emphasis on a clear research question. Specifically, we explore whether the circulation patterns observed during warm temperature extremes are related to circulation patterns that display above-average persistence. In doing so, we explain why we have chosen to use a dynamical systems approach, as suggested in your third comment. We want to clarify that our intention is not to focus on how atmospheric persistence may modulate the intensity of individual heatwaves. Rather, our primary objective is to investigate the connection between heatwave circulation patterns and above-average persistent circulation patterns.

If the main focus of the paper is a comparison with statements from the blocking literature I would also recommend to explain these statements in a bit more detail in the introduction to allow for more clarity in the discussion.

We have added more detail in the introduction according to the reviewer's suggestion. Whilst one aspect of this manuscript is to discuss different understandings of persistence, we also aim to explain how the dynamical systems approach works to readers from a more conventional atmospheric dynamics background. We wish to highlight the differences between these approaches and familiarize a broader audience with the technique we propose, so that it may be applied in research questions where it could be useful, and not remain on the mathematical or theoretical fringes of the climate science community.

One example of an interpretation that I would question:

As the authors explain, the most persistent atmospheric flow is zonal flow (see line 71-72). If summertime heat waves occur when the zonal flow is blocked, one would expect, that the atmospheric circulation during heat waves is not anomalously persistent. To me everything seems to be as expected so far. Therefore, I would write the sentence in line 157-158 differently. To me this seems to be a misunderstanding: It might be true, that more persistent blocking leads to more severe heat waves. And this can be true irrespective of whether zonal flow is generally more persistent than blocking. I therefore also disagree with line 180-182.

In our revised manuscript we have adjusted lines 157-158 of the original manuscript to reflect that the work presented in the manuscript to that point does not necessarily contrast the existing literature on blocked flow patterns when considered from the perspective of the variability of the entire North-Atlantic European domain. Furthermore, with the inclusion of persistence calculated using Z500, our results now appear somewhat more consistent with the conventional view of heatwaves being linked to persistent circulation patters. We would like to highlight that this manuscript is only considering the occurrence of heatwaves, rather than how individual heatwave severity may be modulated by persistence. Consequently, we have revised lines 180-182 of the original manuscript to reflect this refining of the argument. Furthermore, we have revised lines 42-44 of the original manuscript, as upon reading the review comments it has become apparent that we have inadvertently placed weight on the link between the persistence of blocks and the intensity of heatwaves, as opposed to only their occurrence.

Advection analysis and figure 4:

Looking at figure 4 it seems as if the advection that is analyzed here shows rather small scale features. Do these small scale features really represent the large scale flow that is shown in figure 1? Due to this (potential?) inconsistency I do not find the lines 160-166 convincing. In my view, more analysis would be needed to really interpret the role of warm air advection.

Based also on other review comments, we have significantly revised the section on temperature advection and have investigated two alternate metrics for warm temperature advection. Specifically, we now use the advection of potential temperature by 10m winds, and temperature advection at 500hPa (Figs. 4 and 5 respectively in the updated manuscript). We note that these figures have been updated since our responses due to an error in our code. Ultimately, the revised figures do not change our qualitative conclusion of the original manuscript that winter time warm spells appear to be associated with warm temperature advection (at surface levels) in all regions except Russia. We approach Russian warm spells with caution because the potential temperature advection signal is small and noisy, and the significance stippling also appears to be noisy. There is a comparatively weak signal at the surface level during summertime heatwaves. We instead see a signal at mid levels, which we interpret as suggesting that warm air is being transported higher up in the atmosphere before it descends in the vicinity of the identified heatwaves.

I would assume that the analysis is sensitive to the domain over which the atmospheric circulation is analyzed. A justification or an explanation of the choice of the Europe wide domain is lacking in section 2.1. There is one sensitivity test with a shifted domain which is great, but the interpretation of this sensitivity analysis is lacking in the main part of the manuscript. Furthermore, I think it would be more interesting to test the sensitivity to the size of the domain.

In our revised manuscript we have updated our domain to closely follow that of Messori et. Al. (2017) (except with a reduced westerly extent) as we agree that the reviewer is correct that a justification of the domain was lacking. Furthermore, we have included sensitivity tests for a smaller and larger domain, along with a west-shifted domain in the Appendix. As discussed in the sensitivity analysis in the appendix of the revised manuscript, we do see some sensitivity to the domain (and definitions). Despite this, we find that the sensitivity tests do not alter our qualitative conclusions that anomalous atmospheric persistence is not a necessary requirement for heatwaves.

Minor comments:

L11: exact?
Exact is meant as a verb in this sentence, and has been left as is in the original manuscript

L129: Is it relevant that the reader understands the method of Süveges? If yes, please explain it in more detail. If no, I'm not sure if you need to compare it to another method (Ferro and Segers) which is not expleained either?
While the details of the estimator of theta are not directly relevant to our results, we do agree that it is important to ensure the self-contained reproducibility of our article. Based on this, and other review comments, we have significantly expanded our methods section (Section 2.3), paying particular attention to adding more detail on the method of Süveges (2007) on lines 129-132 of the original manuscript, and providing some intuition to the process of calculating theta.

L130: Please write the package name here (I assume it is not "Robin")
We have amended line 130 of the original manuscript to "CDSK by Robin (2020)", thus now including the package name.

L184-185: This formulation could lead to a misinterpretation of the results (see my major comments above). The presented analysis does not study the link between more persistent anti-cyclonic configurations and the intensity and persistence of heat-waves.
Thank you for highlighting this and we agree with your second sentence, thus we have clarified the manuscript by removing mention of heatwave intensity, as this is not the message we wish to convey.

Figure A1: This sensitivity test is helpful and important. Would it also be possible to do

a sensitivity test where the domain over which atmospheric circulation is analyzed is smaller, for example only the Mediterranean?

Following one of the previous comments by the Reviewer, we have run additional sensitivity tests on the domain size which can be found in the Appendix. We find that whilst they show some sensitivity in certain regions, they do not change the overall conclusions of the manuscript. Concerning shrinking the domain to a regional scale, this is in theory possible but may lead to noisy results due to the small number of gridboxes which would then be used to determine analogues. It would likely work for a higher resolution dataset, but would then require a separate analysis for each heatwave domain. We would leave this sort of analysis for future work.

References:

Davini, P., Cagnazzo, C., Gualdi, S., and Navarra, A.: Bidimensional diagnostics, variability, and trends of northern hemisphere blocking, Journal of Climate, 25, 6496–6509, https://doi.org/10.1175/JCLI-D-12-00032.1, 2012

R. Vautard, "Multiple Weather Regimes over the North Atlantic: Analysis of Precursors and Successors-Mon," Weather Reviews, Vol. 118, 1990, pp. 2056-2081. http://dx.doi.org/10.1175/1520-0493(1990)118

Messori, G., Caballero, R., and Faranda, D.: A dynamical systems approach to studying midlatitude weather extremes, Geophysical Research Letters, 44, 3346–3354, https://doi.org/10.1002/2017GL072879, 2017

Dear Reviewer 2,

Thank you for your valuable suggestions and feedback on our manuscript. We provide our responses below in blue.

Review of paper

**The counter-intuitive link between European heatwaves and atmospheric persistence**

by Emma Holmberg et al.

submitted to *Earth System Dynamics*

Holmberg et al. examine the persistence of the European-wide SLP field during summer heat waves and winter warm spells using a previously developed measure of persistence that is based on wellestablished concepts in dynamical systems theory. The authors find that winter warm spells are associated with increased persistence (compared to climatological persistence values derived from all days) while for summer heat waves persistence anomalies are found to be moderately positive during heat waves in Scandinavia and even negative during heat waves in other regions. The authors identify in particular the absence of positive persistence anomalies for summer heat waves as "counter intuitive". The study is overall well written and the figures are, for the most part, clear. Also, I appreciate that authors' attempt to leverage concepts from dynamical systems theory to put the notion of "circulation persistence" onto more solid theoretical foundations. In that sense, I would like to encourage the authors to continue exploring their avenue of research.

However, despite the good intention and interesting overarching research goal of this study, there are a considerable number of issues that significantly compromise the value of this paper in its current form. I therefore suggest this study undergoes major revisions before it can be re-evaluated again. Some of my major comments below relate to the presentation of the methodology and I expect the authors will be able to address those rather easily. However, I also see more fundamental issues related to the authors choice of meteorological analysis techniques as well as to their interpretation of their findings.

We thank the Reviewer for their time and helpful suggestions, and have addressed below how we have edited the manuscript. We note that Fig. 6 has been updated as an error in the scaling factor for the plot was found during the process of preparing the revised manuscript, this has not changed the qualitative pattern or conclusions associated with that figure.

**Major comments**

A) The description of how "persistence" is defined and computed in Section 2.3 is not detailed enough to convey a physical intuition of what the authors exactly mean by persistence. I appreciate that the authors employ a rigorous and mathematical definition of persistence and I am aware that previous studies already introduced this definition of persistence. However, in some instances the authors challenge extremely well accepted reasoning within the atmospheric dynamics community (Blocking patterns during European heat waves are persistent flow features) and thus they should provide sufficient detail and explanations such that peers from this and other fields (i.e., scientists outside dynamical systems research community) can understand what exactly is meant by "persistence" without first consulting other papers. For instance, why exactly is the unit of theta days$^{-1}$? The frequency of what is it exactly? Moreover, is its inverse something like an e-folding time of some quantity? Or on lines 127–128: Why exactly is a generalized Pareto distribution fitted to the negative log of the analogue distances? Or on lines 129–131: What is the method of Süveges (2007)? And after all: How does a measure of clustering in extreme value theory (L69–70) inform about "persistence"? Note that I am not questioning your approach, based on the current Section 2.3 it just appears overly opaque to me.

Thank you for your comments, and detailing exactly which sections need further clarification. The revised manuscript now contains a significantly more detailed methods section, with the aim of providing a more self-contained explanation of the methodology. Specific answers to these questions, which have been included in the revised manuscript, are as follows:

The unit days$^{-1}$ is used because 1/theta is a measure of the expected number timesteps a given atmospheric state (or "map") remains qualitatively similar to itself, or in other words the expected number of timesteps for a cluster of similar maps. A cluster here is simply a continuous succession of similar maps, namely a succession of several days with a similar atmospheric state. An estimate of cluster length is therefore an estimate of persistence. We use daily data, and our theta is thus in units of days$^{-1}$, giving a persistence time in days. Theta is not derived as inverse e-folding time, and this specific response has not been included in the revised manuscript.

We fit a generalized pareto distribution (GPD) to the negative log of the distances because the mathematical theories we use are asymptotic theories which hold only in the limit of a sufficiently large data set, thus we do this to ensure that we have a sufficient sample size and that the analogues have a sufficiently well behaved distribution in time. Furthermore, the theory requires that the negative log of the distances converge to a certain family of distributions, of which the GPD is one, and convenient to work with. In the revised manuscript, we clarified that the method of Süveges (2007) pertains to the algorithm used to estimate theta, and reference has been made to the specific equation used to estimate theta. This equation is relatively complex, thus we have refrained from including it in the manuscript to improve the flow of the text.

B) Related to A): On line 125 you state that you choose the 5% of days with the smallest distances as analogues to any day of interest. However, presumably not all of these

analogues occurred during heat waves/warm spells. It is unclear to me how that fact affects the interpretation of your results. I read in your reply to the Community Comment by Dr. Alexandre Tuel that your methodology characterizes "local properties of the attractor". I don't fully understand the exact meaning of that statement, but it nevertheless appears plausible to me that a certain large-scale circulation (defined by its European-wide SLP pattern) can be persistent in some cases, but much less so in others. That is, it is quite plausible that heat waves occur in a unusually persistent manifestations of a given largescale circulation pattern, that not always exhibits this level of persistence.

We believe that there is a misunderstanding of what "local" means, which we have made every effort to clarify in the revised manuscript. When we estimate the persistence of a given atmospheric pattern, this is not the same as the persistence of its analogues. Say that we have a heatwave on e.g. the 5[th] July 2020, and one of its analogues happens to be the 20[th] August 2018, which was not a heatwave day. The persistence of the atmospheric pattern of the 5[th] July 2020 will not be the same as that of the 20[th] August 2018. Or better said: it could in theory be the same and one could build a synthetic dataset where this is the case, but in practice for "real" data it will not be. That is because the calculation of theta rests on the distribution in time of the closest 5% of analogues of each day, and the 5% of closest analogues of the 5[th] July 2020 will not be the same as the 5% of closest analogues of the 20[th] August 2018, even if the two are analogues of each other. Our method enables a heatwave to be attributed to an unusually persistent configuration compared to its analogues. The term 'local' in this context indicates that the measure of persistence is specific to a single map, which refers to a day in our dataset. We have explicitly clarified in the introduction that this approach does not categorise circulation patterns like a regime based approach, and that the persistence of each circulation pattern corresponding to a given timestep is calculated for that specific time step based on the analogues of that specific time step.

For instance, previous studies have shown that soil moisture anomalies associated with heat waves can have an "anchoring effect" on the associated anticyclonic circulation (e.g., Martius et al., 2021), which only affects the persistence of a given circulation pattern when a heat wave and the associated soil moisture anomaly occurs, but not in situations in which a similar circulation pattern occurs without a heat wave. How do the authors ensure that the persistence they quantify from sets of days including both heat wave and non-heat wave days is representative for heat wave days only? In case this comment simply results from a misunderstanding of your approach, I strongly recommend rewording the description of your approach.

As suggested by the Reviewer, we have now placed further emphasis on the role of non-atmospheric drivers of heatwaves in our introduction and discussion sections in the revised manuscript. Concerning the methodological part of the question, we refer to our above answer, namely the fact that our method assigns different persistence values to a given day versus its analogues. That is, it allows for the fact that one map in a set of similar maps can be more persistent than the others. To clarify this point, we show below (Figure 1) a box-and-whiskers plot of theta (calculated using SLP) for

a heatwave day over Germany (the cross) and for its 5% of closest analogues. As can be seen, this specific heatwave day has an anomalously low theta (high persistence) compared to its analogues. Other heatwave days may instead have an average or unusually low persistence relative to their analogues. We will endeavor to clarify this aspect of our methodology in the manuscript in order to reduce future misunderstandings. In this respect, we trust that the expanded explanation of the methodology we have introduced in response to the Reviewer's comment A will be of help.

[Figure]

Figure 1: Box plot of θ [days$^{-1}$] anomalies calculated for all analogues of September 12th, 2002. The red cross denotes the theta anomaly for September 12th, 2002.

C) A key finding of this study is that during summer heat waves in many parts of Europe, the circulation is not anomalously persistent and the authors claim that this is at odds to the well accepted notion that European summer heat waves (at least in Central and Northern Europe) often occur in association with "persistent blocking" (e.g., lines 157– 158, or 180– 182). However, a contradiction is not particularly apparent to me here. It is well possible that zonal flows (which often lead to wet and thus not anomalously hot summer conditions across central Europe) are more persistent by the author's metric than blocked flows. Nevertheless it may still be true that severe summer heat waves occur preferentially during the most persistent blocking episodes (e.g., lines 42–44 and references cited there). I therefore suggest that the authors more clearly frame their research question and better explain what exactly is counter-intuitive about their findings.

We agree with the Reviewer's point. In the revised manuscript we have clarified that our approach gives a single number quantifying the atmospheric persistence of a specific map. As mentioned in our reply to comment B, when considering a given heatwave associated with a blocked-like configuration, we are not providing the persistence of all blocked configurations, we are providing the persistence of that one specific blocked configuration associated with the heatwave we have identified. Thus, the Reviewer is correct that our results do not contradict the notion of a link between blocking persistence and heatwave persistence. Furthermore, in our revised manuscript we now also consider persistence computed on Z500, which shows a weak but significant link between heatwaves and persistent configurations. Consequently, we have updated the title and now exclude the term 'counter intuitive'. We agree that we need to make the distinction between looking at the persistence of individual heatwaves and looking at the average persistence of blocked versus zonal flows, and have removed lines 42-44 of the original manuscript. Furthermore, we have clarified our research question, specifying that we investigate whether circulation patterns associated with warm temperature extremes are related to circulation patterns displaying above average persistence.

D) I believe Fig. 4 and the quantity B does not help to understand the contribution of largescale warm air advection to heat waves/warm spells, for two reasons: Firstly, it identifies essentially just regions of gradients in the surface elevation. The vector **B** is computed from SLP (which everywhere is indicating the pressure *at sea-level*), while two-meter temperature is dependent on the height of the terrain and thus large horizontal (i.e., terrain following) gradients of two-meter temperature are first and foremost indicating steep terrain. Second, the implicit assumption motivating the definition of **B** is presumably that by the geostrophic balance the gradient of SLP is indicative of the surface winds. However, especially near the surface the geostrophic balance is often quite substantially disturbed by surface drag and the near-surface flow is not as close to geostrophic as the upper-level flow. Therefore, I believe the authors should repeat their analysis and either consider horizontal temperature advection (i.e., $-v\nabla T$) on a near-surface pressure level explicitly or, alternatively, consider the advection of potential temperature by the 10m winds.

We thank the Reviewer for this helpful suggestion and now consider the advection of potential temperature by the 10m winds, and temperature advection at 500hPa (Figs.

4 and 5 respectively in the revised manuscript). We note that these figures have been corrected since the responses were uploaded due to an error in the code. We find that these metrics produce a clearer signal and, as the Reviewer discusses, are not subject to the same issues caused by orographic effects. Ultimately, the updated figures do not change the qualitative conclusions of the original manuscript. Namely, that wintertime warm spells appear to be associated with warm temperature advection in all regions except Russia. There is a comparatively weak signal during summertime heatwaves at surface level, whilst a more pronounced signal is visible at the mid level in the vicinity of the heatwave region.

E) The choice of SLP as variable to quantify the persistence of the large-scale circulation appears peculiar and suboptimal to me, in particular in the context of this study, which often makes reference to atmospheric blocking. Firstly, it is well known that during heat waves so-called heat lows can develop and thus the evolution (and hence potentially also the persistence) of the mid- to upper-tropospheric flow may differ substantially from the evolution of the near-surface flow. Secondly, atmospheric blocks often feature rapidly amplifying disturbances on their upstream side, which feed into the blocks and thereby enhance their persistence (Shutts, 1983; Pfahl et al., 2015). In the SLP field these upstream disturbances manifest themselves as extratropical cyclones (i.e., with strong signals in the SLP field). Thus, one does not a priori expect a "persistent" European-wide SLP pattern during persistent atmospheric blocks. Thirdly, it is anyways quite a stretch to argue about the persistence of blocking flows based on SLP, as blocking is predominantly a manifestation of the upper-level flow (Kautz et al., 2022; Woollings et al., 2018). The authors could easily exclude these issues, e.g., by performing their persistence quantification with (de-trended) Z500 fields or, even better, isentropic PV fields rather than SLP fields, as these quantities much more directly inform about the large-scale circulation.

Thank you for this valuable comment. Indeed, your arguments on the dynamical relevance of SLP highlighted to us that this aspect of the manuscript required further, detailed analysis. Our original motivation for not using Z500 was that trends in the variable one computes the analogues on may lead to bias results (since one risks finding analogues concentrated in the years around the reference day, due to the trend). Detrending a variable prior to analysis may itself be problematic, since the trend is typically subtracted at each gridpoint, leading to complex distortions of the system's phase-space and analogue distances. We have since analysed this in more detail, as shown in the appendix, concluding that the distribution of analogues in time is sufficient for our statistical assumptions, even without detrending.

Consequently, we now include persistence calculations computed separately for both SLP and Z500 (Figs. 2 and 3 of the updated manuscript), as we have since seen that these analyses produced qualitatively different results. The study now considers both persistence and advection at the surface and mid levels, as these two parts of the analysis both appear to contribute information on warm temperature extremes and their mechanisms.

**Minor comments**

1. L5: I do not think your study indeed "reconciles" the dynamical systems and traditional views on persistence. It rather demonstrates that your approach leads to results that diverge from traditional views on persistence.
   We have reworded this, adding that we attempt to contribute a piece of literature which helps clarify this approach in terms of the applicability of the dynamical systems framework to further research questions.

2. L16–17: The authors should clarify what exactly they mean with the word "extreme". Do they mean "rare", e.g., as on line 20 where the authors refer to "Uncommonly high temperatures", i.e., "extreme" in the sense of a large return period? Or do the authors

perhaps mean "extreme" in the sense of "hot enough to cause impact"? If the authors mean "extreme" in the sense of "rare" then the sentence on lines 16–17 is contradictory. If extremely hot summers are defined via their rareness then, by definition, they cannot become more likely. Rather, the temperature corresponding to a certain return period will increase, i.e., events of equal extremeness will become more intense.

Here our definition of extreme is based on a percentile threshold calculated for a given time period. We highlight that if one defines the threshold for hot extremes based on a fixed value, in the future one will have more hot extremes. As the Reviewer points out, it is natural to assume that recomputing the value of the same fixed percentile for the new temperature distribution would result in events of equal extremeness becoming more intense.

3. L43–44: The authors could explain here in more detail how the "current understanding of the link between blocks and summertime heat waves" works physically. How exactly are long-lasting blocks supposed to increase the odds of intense heat waves? Perhaps mention here the adiabatic warming in subsiding air, the clear-sky conditions, drying of soils (due to the persistent local weather within the block), etc.

Thank you for your suggestions, in our revised manuscript we have added detail to the physical explanation of the link between blocking and heatwaves, incorporating your first two suggestions. We have removed the sentence on lines 42-44 of the original manuscript as we believe we have inadvertently placed emphasis on heatwave intensity as opposed to occurrence by including this sentence.

4. L57–60: The authors refer to a series of papers that developed the Quasi-resonant amplification (QRA) hypothesis. The theoretical basis of this hypothesis has been severely challenged by a number of recent publications (e.g., Wirth and Polster, 2021; Wirth, 2020). So far the authors of the QRA papers have not been able to respond to or address this critique. Therefore, if the authors of this paper decide to refer to the QRA papers they should at least indicate that the validity of the reasoning in these QRA papers is severely contested.

Thank you for highlighting this, we have edited the paper to exclude any mention of the QRA hypothesis.

5. L69–71: Here you introduce the concept of persistence from a dynamical systems perspective. I'd find it very helpful if you could add here what "persistence" means in that context, i.e., if you could convey as physical intuition for what that term means in the dynamical systems context and, perhaps also to what extent it is comparable to the persistence of an individual blocking anticyclone identified with the feature-based perspective.

We have edited lines 69-71 from the original manuscript stating that we interpret persistence as how many days in a row the atmosphere is expected to remain similar to itself, adding that this could be thought of as the number of days in a row where there is a similar structure of high and low pressure systems driving the atmospheric circulation pattern. We note that this is a mathematical definition at its core, so the explanation will likely always remain somewhat technical.

6. L89: Is it really 1978–2018 or 1979–2018? Did you use the ERA5 back extension? If so, why only back to 1978?
   This typo has been corrected from 1978 to 1979.

7. L91: Maybe 12 UTC instead of "noon"?
   This has been removed as we now consider daily averages of SSHF.

8. L93: Something is wrong with the longitudes of your domain.
   Thank you for spotting this, we have edited the manuscript accordingly.

9. L94: Is "daily climatology" the same as "calendar day climatology"? If so, consider using the latter term.
   We are using a calendar day climatology and have clarified this in the revised manuscript.

10. L97: What exactly is "the local distribution"? Do you take any forced trend into account when identifying your events of interest? If not, why is that choice justified in this case?
    We have amended line 97 of the original manuscript to include that the local distribution refers to the distribution for a given grid box. We do not take forced trends in temperature into account. Whilst there is certainly a trend, our analysis should not be sensitive to how the heatwaves are distributed in time, since we look for atmospheric analogues of each heatwave over the full time period considered. What could cause issues is if the variable we are computing the analogues on has a strong trend, although does not appear to be the case for our data set.

11. L109–114 and Figs 1, 2, 4 and 5: Given that several of the authors are excellent statisticians I am surprised that you did not apply the False-Discovery-Rate test of Benjamini and Hochberg, (1995) as detailed in Wilks, (2016). Why did you not follow the procedure suggested by Wilks, (2016) even though you clearly perform multiple test simultaneously?
    The Reviewer is correct in pointing this out. We apologise for missing this important step and have included this in the updated analysis.

12. 12. L123: Did you include any latitude weighting when computing the Euclidean distances?
    We have updated our figures to include latitudinal weighting for the computation of the Euclidean distances during the computation of theta, our persistence metric.

13. Figures 1, 2, 4 and 5: The stippling indicating significance is difficult to see. Can you make it better visible? Also, please add the boxes indicating the regions to Figs. 4 and 5, so that the reader always knows what the areas of interest are.

We have increased the saturation of the stippling and added the regions as suggested.

14. Figure 1: I'd be curious to see an analogous figure to Fig. 1 but displaying the Z500 anomaly variance. This would show where (in space) the circulation is (or is not) variable (i.e., not persistent) during your events.

We have included this figure in the Appendix (Fig. A11). This figure should however be interpreted with care as it provides a local as opposed to domain-wide view of persistence.

15. L154–155: I wonder to what extent your European wide circulation persistence compares with the persistence of the "local" weather during your events (e.g, measured by the average/median duration of hot spells during your events or by the T2m variance during your events). If you could indeed bring together these two views on persistence then I think the word "reconciling" in your abstract would be justified.

This is a good suggestion but we believe it hides considerable complexity. Indeed, it is similar to the question of how the persistence of e.g. a block is related to the persistence of a coinciding heatwave, something which is currently somewhat of a knowledge gap. Indeed, we were unable to find studies explicitly relating duration of blocks with duration of the associated heatwaves (while many studies implicitly look at this by imposing a minimum duration to both detect blocking and heatwaves).

16. Figure 3: The axis labels are difficult to read. Please enlarge them.

We have amended these.

17. L165–166: The orographic signatures in Fig. 4 are just a consequence of the definition of B. I do not think we learn anything from these signals about the importance of downslope winds for hot extremes.

We have significantly revised this section following from other review comments as well, and consider the advection of potential temperature by 10m winds and temperature advection at 500hPa, as discussed in response to comment D.

18. L171–172: I do not understand this sentence. How exactly is "Russia an exception"?

We have edited the manuscript arguing that Russia appears to have a less clear case, noting the weak anomalies over land.

19. L201–203: The authors refer to "other atmospheric structures". What exactly is meant here?

We have added that this could be, for example, low pressure systems.

20. L225–228: I find this statement on the importance of persistence for local (i.e., diabatic) and remote drivers (i.e., advection of air from climatologically warmer regions) of heat waves misleading and suggest to reword this statement. Firstly, in particular for heat waves driven by radiative effects the persistence of (local?) weather (i.e., dry conditions, clear skies) is well known to be important because the amplification of land-atmosphere feedbacks, i.e., soil drying and associated increase in sensible at the expense of latent heat fluxes, take time (e.g., Miralles et al., 2019).

We have reworded this to clarify that this is a key point we wish to make, namely differentiating between local conditions and domain-scale persistence, which corresponds to the large-scale structure of the atmosphere.

21. Furthermore it is unclear why warm air advection would require persistence. Even if the circulation changes over time, long-range transport of air from

climatologically warm regions may occur. Moreover, at least at synoptic scales, near-surface warm air advection tends to be largest ahead of cold fronts which I would not characterize as persistent weather situations. I understand that the authors here refer to the persistence of "the large-scale atmospheric configuration", but as it is written now the statement (i.e., the authors interpretation of their results) is at odds to existing literature and, in my opinion, not sufficiently well substantiated by their results. The issue might be resolved if a distinction could be made between the persistence in the large-scale circulation pattern and the persistence of local weather.

We have endeavored to revise the manuscript to highlight where we make a distinction between local and large-scale persistence. In our interpretation we actually make a point that the UK does not show a persistent signal during wintertime warm spells, despite showing a signal for temperature advection. We hypothesise that a persistent signal may be expected for regions where the advection of warm air is expected to come from a more specific direction, as this would likely have a stronger requirement for the large-scale circulation pattern to remain similar. The Reviewer is accurate in pointing out that our manuscript does not utilize a Lagrangian-style tracking algorithm, and our examination of advection is founded on composites. We have elaborated on this aspect further in the revised discussion section.

22. L237–239: The "visual appraisal" does not suffice to claim that "a blocking algorithm would detect a blocked flow persisting for several days". Firstly, the large family of blocking identification algorithms derived from the original approach of Tibaldi and Molteni (1990) usually consider gradient reversals in the absolute (not anomaly) Z500 fields, which is not apparent from your Fig. 6. Secondly, the "visual appraisal" in Fig. 6 does not convince me that, e.g., the 5-day persistence criterion for negative upper-level PV anomalies in the Schwierz et al. (2004) algorithm would be fulfilled. I suggest to tone down the interpretation of Fig. 6 in this regard.

We have toned down the interpretation accordingly, however we would like to note that the first quote of your comment has dropped the word "suggests", which in itself tones the statement down somewhat.

**References**

Benjamini, Y. and Hochberg, Y.: Controlling the False Discovery Rate: A practical and powerful approach to multiple testing, J. R. Stat. Soc., 57, 289–300, https://doi.org/10.1111/j.25176161.1995.tb02031.x, 1995.

Kautz, L.-A., Martius, O., Pfahl, S., Pinto, J. G., Ramos, A. M., Sousa, P. M., and Woollings, T.: Atmospheric blocking and weather extremes over the Euro-Atlantic sector – a review, Weather Clim. Dyn., 3, 305–336, https://doi.org/10.5194/WCD-3-305-2022, 2022.

Martius, O., Wehrli, K., and Rohrer, M.: Local and Remote Atmospheric Responses to Soil Moisture Anomalies in Australia, J. Clim., 34, 9115–9131, https://doi.org/10.1175/JCLI-D-210130.1, 2021.

Masato, G., Hoskins, B. J., and Woollings, T.: Winter and summer Northern Hemisphere blocking in CMIP5 models, J. Clim., 26, 7044–7059, https://doi.org/10.1175/JCLI-D-12-00466.1, 2013.

Miralles, D. G., Gentine, P., Seneviratne, S. I., and Teuling, A. J.: Land–atmospheric feedbacks during droughts and heatwaves: state of the science and current challenges, Ann. N. Y. Acad.

Sci., 1436, 19–35, https://doi.org/10.1111/NYAS.13912, 2019.

Pfahl, S., Schwierz, C., Croci-Maspoli, M., Grams, C. M., and Wernli, H.: Importance of latent heat release in ascending air streams for atmospheric blocking, Nat. Geosci., 8, 610–614, https://doi.org/10.1038/ngeo2487, 2015.

Scherrer, S. C., Croci-Maspoli, M., Schwierz, C., and Appenzeller, C.: Two-dimensional indices of atmospheric blocking and their statistical relationship with winter climate patterns in the EuroAtlantic region, Int. J. Climatol., 26, 233–249, https://doi.org/10.1002/joc.1250, 2006. Schwierz, C., Croci-Maspoli, M., and Davies, H. C.: Perspicacious indicators of atmospheric blocking, Geophys. Res. Lett., 31, L06125, https://doi.org/10.1029/2003GL019341, 2004.

Shutts, G. J.: The propagation of eddies in diffluent jetstreams: Eddy vorticity forcing of

"blocking" flow fields, Q. J. R. Meteorol. Soc., 109, 737–761, https://doi.org/10.1002/qj.49710946204, 1983.

Tibaldi, S. and Molteni, F.: On the operational predictability of blocking, Tellus A, 42, 343–365, https://doi.org/10.1034/J.1600-0870.1990.T01-2-00003.X, 1990.

Wilks, D. S.: "The stippling shows statistically significant grid points": How research results are routinely overstated and overinterpreted, and what to do about it, Bull. Am. Meteorol. Soc., 97, 2263–2273, https://doi.org/10.1175/BAMS-D-15-00267.1, 2016.

Wirth, V.: Waveguidability of idealized midlatitude jets and the limitations of ray tracing theory, Weather Clim. Dyn., 1, 111–125, https://doi.org/10.5194/WCD-1-111-2020, 2020.

Wirth, V. and Polster, C.: The Problem of Diagnosing Jet Waveguidability in the Presence of Large-Amplitude Eddies, J. Atmos. Sci., 78, 3137–3151, https://doi.org/10.1175/JAS-D-200292.1, 2021.

Woollings, T., Barriopedro, D., Methven, J., Son, S.-W., Martius, O., Harvey, B., Sillmann, J., Lupo, A. R., and Seneviratne, S.: Blocking and its response to climate change, Curr. Clim. Chang. Reports, 4, 287–300, https://doi.org/10.1007/s40641-018-0108-z, 2018.

Süveges, M.: Likelihood estimation of the extremal index, Extremes, 10, 41–55, https://doi.org/10.1007/s10687-007-0034-2, 2007

Dear Reviewer 3,

Thank you for your helpful comments. We provide our responses below in blue.

This paper examines the links between heat extremes and a metric of atmospheric persistence from dynamical systems theory. Using this metric, it is argued that there is no evidence of a link between heat extremes and anomalously persistent circulation patterns. Some assessment of the role of different terms in driving temperature anomalies is also provided and it is argued that there is an important role for zonal temperature advection in driving wintertime heat extremes, while in the summertime temperature advection does not play an important role. I can see that it's useful to quantify atmospheric persistence by this dynamical systems measure, although I have some questions about the methodology. My primary concern is about the interpretation and the conclusions when using this measure. I think there are two potential interpretations. One is that persistent circulation patterns are not closely linked to heat extremes, which is the interpretation the authors have provided. The other is that this dynamical systems metric is actually not a very good measure of persistent circulation patterns. I'm not totally convinced by the authors argument that this second interpretation can be ruled out, so my more major suggestion is that either a clearer demonstration of this being a preferred metric for atmospheric circulation persistence should be provided, or the discussion should be changed to be a bit more balanced on whether this metric is accurately measuring persistence of atmospheric circulation anomalies.

We thank the Reviewer for their valuable time and helpful comments, and detail below the edits we have made. We note that Fig. 6 has been updated as an error in the scaling factor for the plot was found during the process of preparing the revised manuscript, this has not changed the qualitative pattern or conclusions associated with that figure.

General comments:

As mentioned above, my primary concern is whether the interpretation that "atmospheric persistence is not a necessary requirement for summertime heatwaves" (l9) is actually correct, or whether an alternative interpretation is that this dynamical systems theory metric of atmospheric persistence does not adequately capture persistent atmospheric circulation anomalies. I feel like the demonstration in Figure 6 supports that this metric might not be very good at capturing persistent atmospheric circulation anomalies. It is clear that Figure 6b and d demonstrate a persistent block, as indeed the authors state. But it seems like it could be argued that this dynamical systems metric is, therefore, just not a very good metric for persistent atmospheric flow anomalies. I'd question whether it can really be used to argue that "highly persistent configurations are not a necessary criterion for European summertime heatwaves to occur" (l242) when clearly there is a case in Figure 6 that does have a persistant atmospheric flow configuration but is considered not persistent by this metric. I'm left a bit confused about what argument is exactly being made. If I read the paper in a cursory manner, I might think that the conclusion is that you don't need persistent

atmospheric circulation patterns to produce heatwaves, but if readers think more about it and pay attention to Figure 6, it seems the conclusion should actually be that this particular dynamical systems metric of persistence doesn't really capture persistence in atmospheric flow patterns of relevance to heatwaves. I suggest the authors either need to make a clearer case for why this metric is preferable to those based on persistent flow regimes or alter the wording in places to make clear that actually this dynamical systems metric of persistence doesn't do that good a job of picking out persistant blocking highs or other flow regimes that are relevant for heatwaves. I think either conclusions is worthy of publication, but I'm confused about which one is being drawn. Another example of a confusing conclusion is lines 247-249 where it's stated that "our results appear to contrast the conventional view of heatwaves being associated with very persistent blocked configurations" when above, in reference to Fig 6, it's stated that a blocking algorithm would detect a block in the low persistence case for several days (l238). It seems, then, that this metric is not a very good metric of blocking, so how can it be used to contrast the conventional view of very persistent blocked configurations being connected to heatwaves?

Thank you for this detailed comment. We would like to argue that the choice of a 'good' metric contains a certain amount of perspective. We believe that this is a very important discussion to be had, and possibly the crux of bringing the mathematical and dynamics perspectives of persistence together so as to be reconciled and mutually understood. We argue that a 'good' metric requires a sound, non-subjective definition, and have significantly revised our manuscript so as to clarify this. As part of this discussion, we argue that the reviewer's evaluation that Fig. 6 of the original manuscript "does have a persistent atmospheric configuration" is a subjective statement, while our chosen distance metric allows to make a quantitative statement. In the study we have updated our examples to select cases where the SLP and Z500 persistence is anomalously high or low, however, the patterns remain similar. We very carefully state that in Fig. A12b of the revised manuscript a blocking algorithm would likely detect a blocked flow pattern, not that there **is** a persistent block. Our whole argument is that the block in this figure should **not** be considered as persistent, at least from a dynamical systems persistence perspective. This is less clear when considering persistence using Z500, however, we maintain our argument that the anomalies seen would, on average, correspond to relatively small anomalies in persistence (on the order of hours). We are not arguing that blocking algorithms are wrong, but rather that they do not necessarily match an objective and quantitative definition of continental-scale atmospheric persistence. It seems to us that the two conceptualisations of persistence are useful for different research questions, which we clarify in our revised discussion section.

(2) One aspect of the methodology that I wondered about was is it easier to have higher persistence when there are less anomalies overall in the spatial field. I'm imagining that if you have a really large amplitude anomaly but that moves slightly, the spatial Euclidian distance between days that have a relatively small movement may end up being a lot larger than the Euclidian distance between days where there's much less going on in the spatial field. If so, then maybe this isn't a particularly good measure of the persistence of relevance to heat extremes. Persistence of nothing much going on over the region wouldn't be very

meaningful, whereas having a persistant blocking high that stays around for a long time in the region, even if it moves slightly, would likely be more impactful for heat extremes. Or perhaps there's something in the methodology that prevents this from happening. I recommendt this be discussed or assessed.

This method is indeed dependent on the distance metric chosen. However, note that a given day will not necessarily be more persistent simply because it has closer analogues – as in your example of a day with a large atmospheric feature versus a day with very small anomalies over the whole domain. The estimate of theta is based on the 5% of closest analogues of each day, and the "closeness" of these analogues to the day itself does not determine the persistence of that day. Rather, it is the distribution of the analogues in time that is important. We have clarified this very important point in the final paragraph of Section 2.3 in the revised manuscript.

Minor comments by line number:

l94: I don't think "345-45W" is correct. Maybe 15W-45E?

Thank you, this has been corrected.

Figure 1 caption: I think this could be a bit clearer if it stated "warm spells during winter (top) and heatwaves during summer (bottom)". (It took me a while to notice the winter and summer in the titles).

Thank you, this has been corrected.

l160: In the discussion of the role of advection starting here and referring to Figure 4, "the role of warm air advection toward the region of interest during warm spells" is not entirely obvious to me in a causal sense. Couldn't there also potentially be a role for the temperature anomaly itself being produced by some other cause actually leading to temperature advection anomalies. I feel like the reds next to blues in this figure may be indicative of that i.e., you get some warm anomaly set up and if the zonal flow is westerly then you end up with cold advection to the west of the warm anomaly and warm advection to the east. Even if that's not the case, the dominant role of warm advection isn't totally clear to me from the figure since it's very noisy. Is the intention that readers should be paying attention to the larger spatial scales, as oposed to the small scale noise? If so, maybe some filtering to retain only larger spatial scales could be performed?

Based also on other review comments, we have significantly revised this figure and use an alternate metric, in order to clarify the arguments around warm temperature advection. Specifically, we use the advection of potential temperature by 10m winds, and temperature advection at 500hPa (Figs. 4 and 5 in the revised manuscript). We note that these figures have been corrected since the responses were uploaded due to an error in the code. Ultimately, the updated figures result in no change to the qualitative conclusion of the original manuscript. Namely, that wintertime warmspells appear to be associated with warm temperature advection at surface levels in all regions except Russia. There is a comparatively weak signal during summertime heatwaves at surface level, whilst a more pronounced signal is visible at the mid level in the vicinity of the heatwave region.

l165-166: You mention the potential role of advection over topography for engendering large temperature anomalies here. But couldn't this also potentially be an artefact of using SLP? SLP will involve some extrapolation below the surface making an assumption about the lapse rate, I think. So SLP will be an approximation over topography and will be affected by the temperature at the surface, so is it possible that this could be playing a role here?

In light of both this comment and others, as mentioned above, we now use the potential temperature advection by 10m winds as a metric instead, which should not be subject to this same possible issue.

---

## Referee Report (RR1)

Second review of paper

**The link between European warm temperature extremes and atmospheric persistence**

by Emma Holmberg et al.

submitted to *Earth System Dynamics*

I thank the authors for the thoughtful and thorough revision of their paper. Many of my initial concerns have been addressed: (a) the introduction to/description of the persistence measure θ is much clearer (the authors' response to my first review also helped me greatly in that regard). (b) The study gained substance by including the analyses of Z500 persistence (I find in particular the new Fig. 2 very interesting). (c) The temperature advection analysis is now also somewhat clearer, even though I still don't find it entirely convincing (see comments 7–11 below).

Nevertheless, I believe that this paper now meets the quality standards of ESD and I therefore recommend this paper for publication after minor revision. Note that I still have a considerable number of minor comments remaining, which, however, almost exclusively pertain to the presentation of the material and ideas, rather than the content itself. I list those comments below and invite the authors to address them when producing their final version of this manuscript.

**Minor comments**

1. Abstract: I find several sentences too vague for meaningfully conveying the key results of this study to a first-time reader. Hereafter, I point to several very specific wordings that I have difficulties to fully and clearly understand what they mean. Moreover, I provide suggestions for how they could be written more clearly. I certainly do not expect that you adopt (all of) them exactly but please reword these sentences to increase their clarity.
   a. Line 5: "… no clear persistence signal in the …" -> " … no statistically significant persistence anomaly in the …".
   b. Lines 6–7: "…, while the surface signal is very weak." -> "…, while there are no significant persistence anomalies of the surface circulation pattern."
   c. Line 7: "… suggesting that other radiative and dynamical processes, as well as local effects," -> Please specify what these processes and effects could be! Do you mean maybe "… suggesting that sensible heating or adiabatic warming …"?
2. L54–55: I agree with this statement for temperature anomalies built up by diabatic heating or adiabatic warming, but not for temperature anomalies built up by advection. Very strong temperature advection often occurs in association with fronts and these are themselves clearly not persistent circulation patterns and they can surely also occur within short-lived large-scale circulation patterns. Therefore, please reword or delete this statement.
3. L62–63: The study of Miralles et al., (2014) goes in that direction. They showed that persistent anticyclonic conditions are needed to facilitate the accumulation of exceptional

heat during recent mega heat waves. Consider whether or not that reference is appropriate here.

4. L130: I find the symbol $T_{potential}$ for potential temperature rather confusing. Firstly, in all other symbols you use the subscript to indicate the level (e.g., "surf", or "500"). Secondly, the standard symbol for potential temperature is $\theta$, which, unfortunately, is already taken by your measure of persistence. Please nevertheless try to come up with a less confusing notation. Also, I find it strange to read the words "potential temperature advection" as part of a numbered equation. Please give this quantity its own symbol or introduce the quantity in an in-text equation, e.g., "The surface potential temperature advection, $-\boldsymbol{v}_{surf} \times \nabla T_{surf}^{potential}$, where ..."

5. L142–143: Please specify in which figures you have applied the FDR test.

6. L178–180: I only see a "canonical blocked configuration" in panels (g) and (j), panels (h) and (i) just show zonally oriented wave train/dipole patterns.

7. Figure 4: Please adjust the spacing of the color scale so that not only the temperature advection from onshore/offshore flows is highlighted.

8. Figures 4 & 5: Perhaps specify in the methods section how exactly you computed the gradients (centered differences?). The noisiness of the resulting advection fields can quite strongly depend on the way how gradients are computed.

9. Line 214: "potential temperature advection" or "temperature advection"?

10. Line 215: I don't think the term "dominant" is justified here, because you have not stated over "what" exactly, advection should dominate and because you have not yet quantified the importance of these other factors. Use "significant" instead, wherever the results are indeed statistically significant?

11. L226: You allude here to a "strong advective signal", but I just can't quite see such a "strong" positive signal in Fig. 4a–f, except over coastal seas. This is somehow puzzling, as we are looking at data from winter, during which the oceans are surely warmer than the land. So how could a positive temperature advection occur over coastal seas? Also, over the land-regions in your boxes the positive values are modest at best. Therefore I'd highly appreciate if (a) you could explain why the largest positive temperature advection values are found over coastal seas (and not coastal land regions) and (b) where exactly you see this "strong advective signal".

12. L249–251: I couldn't quite follow here. I agree that during "blocking onset" I would indeed expect low persistence of the large-scale flow by your metric. However, I would also not expect any relevant heat waves during that phase in the life-cycle of blocks. Rather I'd expect heat waves when blocks attain their maximum intensity or even slightly later. For that later phase, however, I'd indeed expect a rather persistent (upper-level) large-scale flow.

13. L279–281: What do you mean here exactly with "new dynamical behaviors"? To me this reads like "new synoptic flow configurations will lead to heat waves", but this would clearly not be substantiated by your results.

14. L290–291: Please be more specific about what "other mechanisms" could be.

15. L299: delete "likely", as descending motion will undoubtably lead to adiabatic warming.
16. Figure 8: Units are wrong in the color bar labels (Pa instead of hPa?).
17. Figure A11: Please increase the range of the color scale to avoid saturation in most parts of these plots. Also, the units in the color bar title do not match the units in the caption.
18. Figure A12: Please increase the range of the color scale to avoid saturation.

**References**

Miralles, D. G., Teuling, A. J., Van Heerwaarden, C. C., and De Arellano, J. V. G.: Mega-heatwave temperatures due to combined soil desiccation and atmospheric heat accumulation, Nat. Geosci., 7, 345–349, https://doi.org/10.1038/ngeo2141, 2014.

---

## Author Response (AR2)

Dear Editor,

We have revised the manuscript "The link between European warm temperature extremes and atmospheric persistence" according to the reviewers' suggestions. We provide a detailed response in blue to each of the comments below. We would like to thank all Reviewers and the Editor for the time they have invested in supporting the improvement of the manuscript thus far, and hope that this revised version is now suitable for publication in ESD.

Best regards,

Emma Holmberg,

On behalf of all co-authors

Anonymous Referee #2

Second review of paper

The link between European warm temperature extremes and atmospheric persistence

by Emma Holmberg et al.

submitted to Earth System Dynamics

I thank the authors for the thoughtful and thorough revision of their paper. Many of my initial concerns have been addressed: (a) the introduction to/description of the persistence measure θ is much clearer (the authors' response to my first review also helped me greatly in that regard). (b) The study gained substance by including the analyses of Z500 persistence (I find in particular the new Fig. 2 very interesting). (c) The temperature advection analysis is now also somewhat clearer, even though I still don't find it entirely convincing (see comments 7–11 below).

Nevertheless, I believe that this paper now meets the quality standards of ESD and I therefore recommend this paper for publication after minor revision. Note that I still have a considerable number of minor comments remaining, which, however, almost exclusively pertain to the presentation of the material and ideas, rather than the content itself. I list those comments below and invite the authors to address them when producing their final version of this manuscript.

We thank the reviewer for their time and helpful comments. Below we detail our specific answers to each review comment. We have also changed way we visualise the significance stippling in our figures to make it more visible, however, no qualitative changes have been performed unless otherwise specified.

Minor comments

1. Abstract: I find several sentences too vague for meaningfully conveying the key results of this study to a first-time reader. Hereafter, I point to several very specific wordings that I have difficulties to fully and clearly understand what they mean. Moreover, I provide suggestions for how they could be written more clearly. I certainly do not expect that you adopt (all of) them exactly but please reword these sentences to increase their clarity.

a. Line 5: "… no clear persistence signal in the …" -> " … no statistically significant persistence anomaly in the …".

b. Lines 6–7: "…, while the surface signal is very weak." -> "…, while there are no significant persistence anomalies of the surface circulation pattern."

c. Line 7: "… suggesting that other radiative and dynamical processes, as well as local effects," -> Please specify what these processes and effects could be! Do you mean maybe "… suggesting that sensible heating or adiabatic warming …"?

Thank you for your suggestions, we have amended the abstract accordingly. On ll. 6-7 we however specify "few" Instead of "no", as two regions do show a statistically significant signal.

2. L54–55: I agree with this statement for temperature anomalies built up by diabatic heating or adiabatic warming, but not for temperature anomalies built up by advection. Very strong temperature advection often occurs in association with fronts and these are themselves clearly not persistent circulation patterns and they can surely also occur within short-lived large-scale circulation patterns. Therefore, please reword or delete this statement.

We agree with your statement that fronts are associated with very strong temperature advection, yet that they are not persistent circulation patterns. In our context we have a temporal requirement for heatwaves/ warm spells, thus we have clarified in line 54-55 of the original manuscript that we are considering large and long-lived temperature anomalies (in our specific definition, large positive temperature anomalies over a period of at least 5 days).

3. L62–63: The study of Miralles et al., (2014) goes in that direction. They showed that persistent anticyclonic conditions are needed to facilitate the accumulation of exceptional heat during recent mega heat waves. Consider whether or not that reference is appropriate here.

We agree that this is a prime example of a study suggesting as much, although we chose to err on the side of caution and not directly cite this paper. As you mention, this specific study focuses on mega-heatwaves and the build up of extremely high temperatures. As we only consider occurrence and not the intensity of heatwaves, we did not want to cite literature in this specific passage of text which either explicitly or implicitly focuses on intensity, so as to not mislead the reader.

4. L130: I find the symbol $T_{potential}$ for potential temperature rather confusing. Firstly, in all other symbols you use the subscript to indicate the level (e.g., "surf", or "500"). Secondly, the standard symbol for potential temperature is $\theta$, which, unfortunately, is already taken by your measure of persistence. Please nevertheless try to come up with a less confusing notation. Also, I find it strange to read the words "potential temperature advection" as part of a numbered equation. Please give this quantity its own symbol or introduce the quantity in an in-text equation, e.g., "The surface potential temperature advection, $-v_{surf} \times \nabla T_{surf}^{potential}$, where …"

Thank you for raising this issue, we have changed potential temperature to have potential in superscript, with subscript denoting the level e.g. surf, 500hPa.

5. L142–143: Please specify in which figures you have applied the FDR test.

We have now specified that this was applied to all figures with significance testing.

6. L178–180: I only see a "canonical blocked configuration" in panels (g) and (j), panels (h) and (i) just show zonally oriented wave train/dipole patterns.

Thank you, we have edited the text accordingly.

7. Figure 4: Please adjust the spacing of the color scale so that not only the temperature advection from onshore/offshore flows is highlighted.

For ease of comparison with other figures we see a benefit in maintaining an equal spacing colour scheme. Furthermore, with the updated significance stippling the colours are in general more visible.

8. Figures 4 & 5: Perhaps specify in the methods section how exactly you computed the gradients (centered differences?). The noisiness of the resulting advection fields can quite strongly depend on the way how gradients are computed.

We have specified in the methods section, directly after defining the equations for temperature advection, that we have used the function gradient from the python package NumPy.

9. Line 214: "potential temperature advection" or "temperature advection"?

Thank you, this has been corrected.

10. Line 215: I don't think the term "dominant" is justified here, because you have not stated over "what" exactly, advection should dominate and because you have not yet quantified the importance of these other factors. Use "significant" instead, wherever the results are indeed statistically significant?

We agree with your point and have replaced dominant with significant on line 215 of the original manuscript.

11. L226: You allude here to a "strong advective signal", but I just can't quite see such a "strong" positive signal in Fig. 4a–f, except over coastal seas. This is somehow puzzling, as we are looking at data from winter, during which the oceans are surely warmer than the land. So how could a positive temperature advection occur over coastal seas? Also, over the land-regions in your boxes the positive values are modest at best. Therefore I'd highly appreciate if

(a) you could explain why the largest positive temperature advection values are found over coastal seas (and not coastal land regions) and

We have performed some additional analysis to investigate the larger positive anomalies in coastal sea grid boxes as compared to the coastal land gridboxes. Fig. 1 shows composites of sea surface temperatures during warm temperature extremes, with arrows corresponding to the wind direction overlaid. We focus on panels a, b and d as these appear to have the strongest signal for warm potential temperature advection over coastal sea gridboxes. We note that here the warmest sea surface temperatures during winter in the vicinity of the warm spell region do not occur directly at the coast line. Thus, one would expect a positive signal for warm temperature advection over this cooler coastal water. Furthermore, in the Baltic sea one would typically expect the presence of sea ice, which would have a colder skin temperature than open water. The weaker signal for positive temperature advection anomalies over coastal land gridboxes could be because we are looking at potential temperature advection at the surface, which could be subject interaction effects between the land surface and the atmosphere, in particular strong radiative cooling during winter. Furthermore, advection over land is expected to be smaller because surface winds over land are much smaller than over the ocean due to the greater surface friction over land. Consequently, we also computed temperature advection at 850hPa in an attempt to capture warm temperature advection anomalies closer to the surface, but not so close as to be impacted by land surface interaction effects. These figures can be seen in Fig. 2. We appear to see an orographic signal in Fig. 2, as well as Fig. 3. Given that we consider potential temperature at the surface, we suggest that this

orographic signal could be due to diabatic effects- namely that when warm maritime air reaches the coast and is advected over topography, rain is induced, which diabatically warms the air on the coastal side of orography.

[Figure]

Figure 1: Sea surface temperature (K) during warm spell/ heatwave days in (a,g) Scandinavia, (b, h) Germany, (c, i) Russia, (d, j) British Isles, (e, k) Iberia, (f, l) Mediterranean, during winter (a–f) and summer (g–l). Arrows corresponding to the surface wind direction have been overlaid.

[Figure]

Figure 2: Temperature advection at 850 hPa (K/day) anomaly during warm spell/ heatwave days in (a,g) Scandinavia, (b, h) Germany, (c, i) Russia, (d, j) British Isles, (e, k) Iberia, (f, l) Mediterranean, during winter (a–f) and summer (g–l). Statistical significance is assessed as described in Section 2 of the manuscript and shown with stippling.

**Potential Temperature Advection Anomaly Warm Temperature Extremes All Days**

[Figure]

Figure 3: Surface potential temperature advection (K/day) anomaly during warm spell/ heatwave days in (a,g) Scandinavia, (b, h) Germany, (c, i) Russia, (d, j) British Isles, (e, k) Iberia, (f, l) Mediterranean, during winter (a–f) and summer (g–l). Statistical significance is assessed as described in Section 2 of the manuscript and shown with stippling.

(b) where exactly you see this "strong advective signal".

We agree with the reviewer and have revised "strong" to "moderate".

12. L249–251: I couldn't quite follow here. I agree that during "blocking onset" I would indeed expect low persistence of the large-scale flow by your metric. However, I would also not expect any relevant heat waves during that phase in the life-cycle of blocks. Rather I'd expect heat waves when blocks attain their maximum intensity or even slightly later. For that later phase, however, I'd indeed expect a rather persistent (upper-level) large-scale flow.

A key point of the Lucarini and Gritsun (2020) paper is that they find the final phase of the blocks (which one would expect to be associated with heatwaves) to also be structurally unstable. We have edited line 251 of the original manuscript to clarify that we are referring to dynamical systems persistence in this context, which does not always have a 1:1 correlation with what is expected from classical large-scale dynamics.

13. L279–281: What do you mean here exactly with "new dynamical behaviors"? To me this reads like "new synoptic flow configurations will lead to heat waves", but this would clearly not be substantiated by your results.

Here we would like to draw attention to the fact that the method of analogues in the form presented in this manuscript assumes a stationary climate. While we have shown that the last few decades are sufficiently close to stationary (from an atmospheric analogues' perspective) that the method may be applied, this may no longer necessarily be valid in the future. We agree that our results do not substantiate the statement "new synoptic flow configurations will lead to heat waves" and this was not the message we intended to convey. We merely wish to draw attention to the fact that the assumptions made in our study may not be applicable to future climates. We have reworded the original statement to highlight this.

14. L290–291: Please be more specific about what "other mechanisms" could be.

We have amended the manuscript to suggest that adiabatic mechanisms such as subsidence could play a role.

15. L299: delete "likely", as descending motion will undoubtably lead to adiabatic warming.

Thank you, this has been corrected.

16. Figure 8: Units are wrong in the color bar labels (Pa instead of hPa?).

Thank you, this has also been corrected.

17. Figure A11: Please increase the range of the color scale to avoid saturation in most parts of these plots. Also, the units in the color bar title do not match the units in the caption.

Thank you, this has been amended.

18. Figure A12: Please increase the range of the color scale to avoid saturation.

Thank you, this has also been amended.

References

Miralles, D. G., Teuling, A. J., Van Heerwaarden, C. C., and De Arellano, J. V. G.: Megaheatwave temperatures due to combined soil desiccation and atmospheric heat accumulation, Nat. Geosci., 7, 345–349, https://doi.org/10.1038/ngeo2141, 2014.

Anonymous Referee #3

The link between European warm temperature extremes and atmospheric persistence, by Holmberg et al

The authors have adequately addressed my comments from the first round. A few technical suggestions...

We thank the reviewer for their time and are pleased to hear that the reviewer is satisfied with the revisions in response to their comments from the first round. Below we detail our specific answers to each review comment. We have also changed way we visualise the significance stippling in our figures to make it more visible, however, no qualitative changes have been performed unless otherwise specified.

l45: "were for example used" --> "were, for example, used"

This has been corrected, thank you.

l49: instead of "simulate" here, do you mean "investigate"? I'm not really sure what is being simulated using the SLP field.

This has been amended, thank you.

l131: "where where" --> "where"

Thank you for spotting this, the repeated where has been deleted.

l189: It might be worth reminding readers here that negative theta anomalies mean greater persistence. I had to go back to the methods section to remember the sign convention.

Thank you for this comment, we have included the suggested reminder in the revised manuscript.

---

## Author Response (AR3)

Dear Editor,

We are delighted to hear your decision and have edited the manuscript as detailed below in blue. We note one further technical correction which has been performed in the revision of this manuscript. Namely, upon reading the in-house guidelines for figure titles we realised we had capitalised all words in figure titles as opposed to just the first word and proper nouns. This has now been corrected, and no other changes have been made to any figure. We thank you for your guidance, support and the time you have invested throughout this review process.

Best regards,

Emma Holmberg on behalf of all co-authors

On line 145 your using the symbol Theta, but Theta is first defined on line 154. All symbols have to be defined at first usage.

Thank you, this mention of theta was removed.

On line 146 you mention multiple hypothesis tests and Wilks. I am assuming that you are using the False Discovery Rate; if that is the case then you should state this and not call this just a correction.

Thank you, we have amended the manuscript accordingly.